# CHARACTERIZING THE SPECTRUM OF THE NTK VIA A POWER SERIES EXPANSION

†* **Michael Murray,** †* **Hui Jin,** †* **Benjamin Bowman,** †‡§ **Guido Montufar**
† Department of Mathematics, UCLA, CA, USA
‡ Department of Statistics, UCLA, CA, USA
§ Max Planck Institute for Mathematics in the Sciences, Leipzig, Germany
`[mmurray,huijin,benbowman314,montufar]@math.ucla.edu`
*Equal contribution

## ABSTRACT

Under mild conditions on the network initialization we derive a power series expansion for the Neural Tangent Kernel (NTK) of arbitrarily deep feedforward networks in the infinite width limit. We provide expressions for the coefficients of this power series which depend on both the Hermite coefficients of the activation function as well as the depth of the network. We observe faster decay of the Hermite coefficients leads to faster decay in the NTK coefficients and explore the role of depth. Using this series, first we relate the effective rank of the NTK to the effective rank of the input-data Gram. Second, for data drawn uniformly on the sphere we study the eigenvalues of the NTK, analyzing the impact of the choice of activation function. Finally, for generic data and activation functions with sufficiently fast Hermite coefficient decay, we derive an asymptotic upper bound on the spectrum of the NTK.

## 1 INTRODUCTION

Neural networks currently dominate modern artificial intelligence, however, despite their empirical success establishing a principled theoretical foundation for them remains an active challenge. The key difficulties are that neural networks induce nonconvex optimization objectives (Sontag & Sussmann, 1989) and typically operate in an overparameterized regime which precludes classical statistical learning theory (Anthony & Bartlett, 2002). The persistent success of overparameterized models tuned via non-convex optimization suggests that the relationship between the parameterization, optimization, and generalization is more sophisticated than that which can be addressed using classical theory.

A recent breakthrough on understanding the success of overparameterized networks was established through the Neural Tangent Kernel (NTK) (Jacot et al., 2018). In the infinite width limit the optimization dynamics are described entirely by the NTK and the parameterization behaves like a linear model (Lee et al., 2019). In this regime explicit guarantees for the optimization and generalization can be obtained (Du et al., 2019a;b; Arora et al., 2019a; Allen-Zhu et al., 2019; Zou et al., 2020). While one must be judicious when extrapolating insights from the NTK to finite width networks (Lee et al., 2020), the NTK remains one of the most promising avenues for understanding deep learning on a principled basis.

The spectrum of the NTK is fundamental to both the optimization and generalization of wide networks. In particular, bounding the smallest eigenvalue of the NTK Gram matrix is a staple technique for establishing convergence guarantees for the optimization (Du et al., 2019a;b; Oymak & Soltanolkotabi, 2020). Furthermore, the full spectrum of the NTK Gram matrix governs the dynamics of the empirical risk (Arora et al., 2019b), and the eigenvalues of the associated integral operator characterize the dynamics of the generalization error outside the training set (Bowman & Montufar, 2022; Bowman & Montúfar, 2022). Moreover, the decay rate of the generalization error for Gaussian process regression using the NTK can be characterized by the decay rate of the spectrum (Caponnetto & De Vito, 2007; Cui et al., 2021; Jin et al., 2022).

The importance of the spectrum of the NTK has led to a variety of efforts to characterize its structure via random matrix theory and other tools (Yang & Salman, 2019; Fan & Wang, 2020). There is a broader body of work studying the closely related Conjugate Kernel, Fisher Information Matrix, and Hessian (Poole et al., 2016; Pennington & Worah, 2017; 2018; Louart et al., 2018; Karakida et al., 2020). These results often require complex random matrix theory or operate in a regime where the input dimension is sent to infinity. By contrast, using a just a power series expansion we are able to characterize a variety of attributes of the spectrum for fixed input dimension and recover key results from prior work.

## 1.1 CONTRIBUTIONS

In Theorem 3.1 we derive coefficients for the power series expansion of the NTK under unit variance initialization, see Assumption 2. Consequently we are able to derive insights into the NTK spectrum, notably concerning the outlier eigenvalues as well as the asymptotic decay.

- In Theorem 4.1 and Observation 4.2 we demonstrate that the largest eigenvalue $\lambda_1(\mathbf{K})$ of the NTK takes up an $\Omega(1)$ proportion of the trace and that there are $O(1)$ outlier eigenvalues of the same order as $\lambda_1(\mathbf{K})$.
- In Theorem 4.3 and Theorem 4.5 we show that the effective rank $Tr(\mathbf{K})/\lambda_1(\mathbf{K})$ of the NTK is upper bounded by a constant multiple of the effective rank $Tr(\mathbf{X}\mathbf{X}^T)/\lambda_1(\mathbf{X}\mathbf{X}^T)$ of the input data Gram matrix for both infinite and finite width networks.
- In Corollary 4.7 and Theorem 4.8 we characterize the asymptotic behavior of the NTK spectrum for both uniform and nonuniform data distributions on the sphere.

## 1.2 RELATED WORK

**Neural Tangent Kernel (NTK):** the NTK was introduced by Jacot et al. (2018), who demonstrated that in the infinite width limit neural network optimization is described via a kernel gradient descent. As a consequence, when the network is polynomially wide in the number of samples, global convergence guarantees for gradient descent can be obtained (Du et al., 2019a;b; Allen-Zhu et al., 2019; Zou & Gu, 2019; Lee et al., 2019; Zou et al., 2020; Oymak & Soltanolkotabi, 2020; Nguyen & Mondelli, 2020; Nguyen, 2021). Furthermore, the connection between infinite width networks and Gaussian processes, which traces back to Neal (1996), has been reinvigorated in light of the NTK. Recent investigations include Lee et al. (2018); de G. Matthews et al. (2018); Novak et al. (2019).

**Analysis of NTK Spectrum:** theoretical analysis of the NTK spectrum via random matrix theory was investigated by Yang & Salman (2019); Fan & Wang (2020) in the high dimensional limit. Velikanov & Yarotsky (2021) demonstrated that for ReLU networks the spectrum of the NTK integral operator asymptotically follows a power law, which is consistent with our results for the uniform data distribution. Basri et al. (2019) calculated the NTK spectrum for shallow ReLU networks under the uniform distribution, which was then expanded to the nonuniform case by Basri et al. (2020). Geifman et al. (2022) analyzed the spectrum of the conjugate kernel and NTK for convolutional networks with ReLU activations whose pixels are uniformly distributed on the sphere. Geifman et al. (2020); Bietti & Bach (2021); Chen & Xu (2021) analyzed the reproducing kernel Hilbert spaces of the NTK for ReLU networks and the Laplace kernel via the decay rate of the spectrum of the kernel. In contrast to previous works, we are able to address the spectrum in the finite dimensional setting and characterize the impact of different activation functions on it.

**Hermite Expansion:** Daniely et al. (2016) used Hermite expansion to the study the expressivity of the Conjugate Kernel. Simon et al. (2022) used this technique to demonstrate that any dot product kernel can be realized by the NTK or Conjugate Kernel of a shallow, zero bias network. Oymak & Soltanolkotabi (2020) use Hermite expansion to study the NTK and establish a quantitative bound on the smallest eigenvalue for shallow networks. This approach was incorporated by Nguyen & Mondelli (2020) to handle convergence for deep networks, with sharp bounds on the smallest NTK eigenvalue for deep ReLU networks provided by Nguyen et al. (2021). The Hermite approach was utilized by Panigrahi et al. (2020) to analyze the smallest NTK eigenvalue of shallow networks under various activations. Finally, in a concurrent work Han et al. (2022) use Hermite expansions to develop a principled and efficient polynomial based approximation algorithm for the NTK and CNTK. In contrast to the aforementioned works, here we employ the Hermite expansion to charac-

terize both the outlier and asymptotic portions of the spectrum for both shallow and deep networks under general activations.

## 2 PRELIMINARIES

For our notation, lower case letters, e.g., $x, y$, denote scalars, lower case bold characters, e.g., $\mathbf{x}, \mathbf{y}$ are for vectors, and upper case bold characters, e.g., $\mathbf{X}, \mathbf{Y}$, are for matrices. For natural numbers $k_1, k_2 \in \mathbb{N}$ we let $[k_1] = \{1, \ldots, k_1\}$ and $[k_2, k_1] = \{k_2, \ldots, k_1\}$. If $k_2 > k_1$ then $[k_2, k_1]$ is the empty set. We use $\|\cdot\|_p$ to denote the $p$-norm of the matrix or vector in question and as default use $\|\cdot\|$ as the operator or 2-norm respectively. We use $\mathbf{1}_{m \times n} \in \mathbb{R}^{m \times n}$ to denote the matrix with all entries equal to one. We define $\delta_{p=c}$ to take the value 1 if $p = c$ and be zero otherwise. We will frequently overload scalar functions $\phi : \mathbb{R} \to \mathbb{R}$ by applying them elementwise to vectors and matrices. The entry in the $i$th row and $j$th column of a matrix we access using the notation $[\mathbf{X}]_{ij}$. The Hadamard or entrywise product of two matrices $\mathbf{X}, \mathbf{Y} \in \mathbb{R}^{m \times n}$ we denote $\mathbf{X} \odot \mathbf{Y}$ as is standard. The $p$th Hadamard power we denote $\mathbf{X}^{\odot p}$ and define it as the Hadamard product of $\mathbf{X}$ with itself $p$ times,

$$\mathbf{X}^{\odot p} := \mathbf{X} \odot \mathbf{X} \odot \cdots \odot \mathbf{X}.$$

Given a Hermitian or symmetric matrix $\mathbf{X} \in \mathbb{R}^{n \times n}$, we adopt the convention that $\lambda_i(\mathbf{X})$ denotes the $i$th largest eigenvalue,

$$\lambda_1(\mathbf{X}) \geq \lambda_2(\mathbf{X}) \geq \cdots \geq \lambda_n(\mathbf{X}).$$

Finally, for a square matrix $\mathbf{X} \in \mathbb{R}^{n \times n}$ we let $Tr(\mathbf{X}) = \sum_{i=1}^{n} [\mathbf{X}]_{ii}$ denote the trace.

### 2.1 HERMITE EXPANSION

We say that a function $f \colon \mathbb{R} \to \mathbb{R}$ is square integrable with respect to the standard Gaussian measure $\gamma(z) = \frac{1}{\sqrt{2\pi}} e^{-z^2/2}$ if $\mathbb{E}_{X \sim \mathcal{N}(0,1)}[f(X)^2] < \infty$. We denote by $L^2(\mathbb{R}, \gamma)$ the space of all such functions. The normalized probabilist's Hermite polynomials are defined as

$$h_k(x) = \frac{(-1)^k e^{x^2/2}}{\sqrt{k!}} \frac{d^k}{dx^k} e^{-x^2/2}, \quad k = 0, 1, \ldots$$

and form a complete orthonormal basis in $L^2(\mathbb{R}, \gamma)$ (O'Donnell, 2014, §11). The Hermite expansion of a function $\phi \in L^2(\mathbb{R}, \gamma)$ is given by $\phi(x) = \sum_{k=0}^{\infty} \mu_k(\phi) h_k(x)$, where $\mu_k(\phi) = \mathbb{E}_{X \sim \mathcal{N}(0,1)}[\phi(X) h_k(X)]$ is the $k$th normalized probabilist's Hermite coefficient of $\phi$.

### 2.2 NTK PARAMETRIZATION

In what follows, for $n, d \in \mathbb{N}$ let $\mathbf{X} \in \mathbb{R}^{n \times d}$ denote a matrix which stores $n$ points in $\mathbb{R}^d$ row-wise. Unless otherwise stated, we assume $d \leq n$ and denote the $i$th row of $\mathbf{X}_n$ as $\mathbf{x}_i$. In this work we consider fully-connected neural networks of the form $f^{(L+1)} \colon \mathbb{R}^d \to \mathbb{R}$ with $L \in \mathbb{N}$ hidden layers and a linear output layer. For a given input vector $\mathbf{x} \in \mathbb{R}^d$, the activation $f^{(l)}$ and preactivation $g^{(l)}$ at each layer $l \in [L+1]$ are defined via the following recurrence relations,

$$g^{(1)}(\mathbf{x}) = \gamma_w \mathbf{W}^{(1)} \mathbf{x} + \gamma_b \mathbf{b}^{(1)}, \ f^{(1)}(\mathbf{x}) = \phi\left(g^{(1)}(\mathbf{x})\right),$$

$$g^{(l)}(\mathbf{x}) = \frac{\sigma_w}{\sqrt{m_{l-1}}} \mathbf{W}^{(l)} f^{(l-1)}(\mathbf{x}) + \sigma_b \mathbf{b}^{(l)}, \ f^{(l)}(\mathbf{x}) = \phi\left(g^{(l)}(\mathbf{x})\right), \ \forall l \in [2, L], \quad (1)$$

$$g^{(L+1)}(\mathbf{x}) = \frac{\sigma_w}{\sqrt{m_L}} \mathbf{W}^{(L+1)} f^{(L)}(\mathbf{x}), \ f^{(L+1)}(\mathbf{x}) = g^{(L+1)}(\mathbf{x}).$$

The parameters $\mathbf{W}^{(l)} \in \mathbb{R}^{m_l \times m_{l-1}}$ and $\mathbf{b}^{(l)} \in \mathbb{R}^{m_l}$ are the weight matrix and bias vector at the $l$th layer respectively, $m_0 = d$, $m_{L+1} = 1$, and $\phi \colon \mathbb{R} \to \mathbb{R}$ is the activation function applied elementwise. The variables $\gamma_w, \sigma_w \in \mathbb{R}_{>0}$ and $\gamma_b, \sigma_b \in \mathbb{R}_{\geq 0}$ correspond to weight and bias hyperparameters respectively. Let $\theta_l \in \mathbb{R}^p$ denote a vector storing the network parameters $(\mathbf{W}^{(h)}, \mathbf{b}^{(h)})_{h=1}^{l}$ up to and including the $l$th layer. The Neural Tangent Kernel (Jacot et al., 2018) $\tilde{\Theta}^{(l)} \colon \mathbb{R}^d \times \mathbb{R}^d \to \mathbb{R}$ associated with $f^{(l)}$ at layer $l \in [L+1]$ is defined as

$$\tilde{\Theta}^{(l)}(\mathbf{x}, \mathbf{y}) := \langle \nabla_{\theta_l} f^{(l)}(\mathbf{x}), \nabla_{\theta_l} f^{(l)}(\mathbf{y}) \rangle. \quad (2)$$

We will mostly study the NTK under the following standard assumptions.

**Assumption 1.** *NTK initialization.*

1. *At initialization all network parameters are distributed as $\mathcal{N}(0, 1)$ and are mutually independent.*

2. *The activation function satisfies $\phi \in L^2(\mathbb{R}, \gamma)$, is differentiable almost everywhere and its derivative, which we denote $\phi'$, also satisfies $\phi' \in L^2(\mathbb{R}, \gamma)$.*

3. *The widths are sent to infinity in sequence, $m_1 \to \infty, m_2 \to \infty, \ldots, m_L \to \infty$.*

Under Assumption 1, for any $l \in [L+1]$, $\tilde{\Theta}^{(l)}(\mathbf{x}, \mathbf{y})$ converges in probability to a deterministic limit $\Theta^{(l)} \colon \mathbb{R}^d \times \mathbb{R}^d \to \mathbb{R}$ (Jacot et al., 2018) and the network behaves like a kernelized linear predictor during training; see, e.g., Arora et al. (2019b); Lee et al. (2019); Woodworth et al. (2020). Given access to the rows $(\mathbf{x}_i)_{i=1}^n$ of $\mathbf{X}$ the NTK matrix at layer $l \in [L+1]$, which we denote $\mathbf{K}_l$, is the $n \times n$ matrix with entries defined as

$$[\mathbf{K}_l]_{ij} = \frac{1}{n}\Theta^{(l)}(\mathbf{x}_i, \mathbf{x}_j), \ \forall (i, j) \in [n] \times [n]. \tag{3}$$

## 3 EXPRESSING THE NTK AS A POWER SERIES

The following assumption allows us to study a power series for the NTK of deep networks and with general activation functions. We remark that power series for the NTK of deep networks with positive homogeneous activation functions, namely ReLU, have been studied in prior works Han et al. (2022); Chen & Xu (2021); Bietti & Bach (2021); Geifman et al. (2022). We further remark that while these works focus on the asymptotics of the NTK spectrum we also study the large eigenvalues.

**Assumption 2.** *The hyperparameters of the network satisfy $\gamma_w^2 + \gamma_b^2 = 1$, $\sigma_w^2 \mathbb{E}_{Z \sim \mathcal{N}(0,1)}[\phi(Z)^2] \leq 1$ and $\sigma_b^2 = 1 - \sigma_w^2 \mathbb{E}_{Z \sim \mathcal{N}(0,1)}[\phi(Z)^2]$. The data is normalized so that $\|\mathbf{x}_i\| = 1$ for all $i \in [n]$.*

Recall under Assumption 1 that the preactivations of the network are centered Gaussian processes (Neal, 1996; Lee et al., 2018). Assumption 2 ensures the preactivation of each neuron has unit variance and thus is reminiscent of the LeCun et al. (2012), Glorot & Bengio (2010) and He et al. (2015) initializations, which are designed to avoid vanishing and exploding gradients. We refer the reader to Appendix A.3 for a thorough discussion. Under Assumption 2 we will show it is possible to write the NTK not only as a dot-product kernel but also as an analytic power series on $[-1, 1]$ and derive expressions for the coefficients. In order to state this result recall, given a function $f \in L^2(\mathbb{R}, \gamma)$, that the $p$th normalized probabilist's Hermite coefficient of $f$ is denoted $\mu_p(f)$, we refer the reader to Appendix A.4 for an overview of the Hermite polynomials and their properties. Furthermore, letting $\bar{a} = (a_j)_{j=0}^\infty$ denote a sequence of real numbers, then for any $p, k \in \mathbb{Z}_{\geq 0}$ we define

$$F(p, k, \bar{a}) = \begin{cases} 1, & k = 0 \text{ and } p = 0, \\ 0, & k = 0 \text{ and } p \geq 1, \\ \sum_{(j_i) \in \mathcal{J}(p,k)} \prod_{i=1}^k a_{j_i}, & k \geq 1 \text{ and } p \geq 0, \end{cases} \tag{4}$$

where

$$\mathcal{J}(p, k) := \left\{ (j_i)_{i \in [k]} \ : \ j_i \geq 0 \ \forall i \in [k], \ \sum_{i=1}^k j_i = p \right\} \quad \text{for all } p \in \mathbb{Z}_{\geq 0}, k \in \mathbb{N}.$$

Here $\mathcal{J}(p, k)$ is the set of all $k$-tuples of nonnegative integers which sum to $p$ and $F(p, k, \bar{a})$ is therefore the sum of all ordered products of $k$ elements of $\bar{a}$ whose indices sum to $p$. We are now ready to state the key result of this section, Theorem 3.1, whose proof is provided in Appendix B.1.

**Theorem 3.1.** *Under Assumptions 1 and 2, for all $l \in [L+1]$*

$$n\mathbf{K}_l = \sum_{p=0}^\infty \kappa_{p,l} \left( \mathbf{X}\mathbf{X}^T \right)^{\odot p}. \tag{5}$$

*The series for each entry $n[\mathbf{K}_l]_{ij}$ converges absolutely and the coefficients $\kappa_{p,l}$ are nonnegative and can be evaluated using the recurrence relationships*

$$\kappa_{p,l} = \begin{cases} \delta_{p=0}\gamma_b^2 + \delta_{p=1}\gamma_w^2, & l = 1, \\ \alpha_{p,l} + \sum_{q=0}^p \kappa_{q,l-1}v_{p-q,l}, & l \in [2, L+1], \end{cases} \tag{6}$$

*where*

$$\alpha_{p,l} = \begin{cases} \sigma_w^2 \mu_p^2(\phi) + \delta_{p=0}\sigma_b^2, & l = 2, \\ \sum_{k=0}^{\infty} \alpha_{k,2} F(p, k, \bar{\alpha}_{l-1}), & l \geq 3, \end{cases} \tag{7}$$

*and*

$$\upsilon_{p,l} = \begin{cases} \sigma_w^2 \mu_p^2(\phi'), & l = 2, \\ \sum_{k=0}^{\infty} \upsilon_{k,2} F(p, k, \bar{\alpha}_{l-1}), & l \geq 3, \end{cases} \tag{8}$$

*are likewise nonnegative for all $p \in \mathbb{Z}_{\geq 0}$ and $l \in [2, L+1]$.*

As already remarked, power series for the NTK have been studied in previous works, however, to the best of our knowledge Theorem 3.1 is the first to explicitly express the coefficients at a layer in terms of the coefficients of previous layers. To compute the coefficients of the NTK as per Theorem 3.1, the Hermite coefficients of both $\phi$ and $\phi'$ are required. Under Assumption 3 below, which has minimal impact on the generality of our results, this calculation can be simplified. In short, under Assumption 3 $\upsilon_{p,2} = (p+1)\alpha_{p+1,2}$ and therefore only the Hermite coefficients of $\phi$ are required. We refer the reader to Lemma B.3 in Appendix B.2 for further details.

**Assumption 3.** *The activation function $\phi \colon \mathbb{R} \to \mathbb{R}$ is absolutely continuous on $[-a, a]$ for all $a > 0$, differentiable almost everywhere, and is polynomially bounded, i.e., $|\phi(x)| = \mathcal{O}(|x|^\beta)$ for some $\beta > 0$. Further, the derivative $\phi' \colon \mathbb{R} \to \mathbb{R}$ satisfies $\phi' \in L^2(\mathbb{R}, \gamma)$.*

We remark that ReLU, Tanh, Sigmoid, Softplus and many other commonly used activation functions satisfy Assumption 3. In order to understand the relationship between the Hermite coefficients of the activation function and the coefficients of the NTK, we first consider the simple two-layer case with $L = 1$ hidden layers. From Theorem 3.1

$$\kappa_{p,2} = \sigma_w^2(1 + \gamma_w^2 p)\mu_p^2(\phi) + \sigma_w^2 \gamma_b^2(1 + p)\mu_{p+1}^2(\phi) + \delta_{p=0}\sigma_b^2. \tag{9}$$

As per Table 1, a general trend we observe across all activation functions is that the first few coefficients account for the large majority of the total NTK coefficient series.

Table 1: Percentage of $\sum_{p=0}^{\infty} \kappa_{p,2}$ accounted for by the first $T + 1$ NTK coefficients assuming $\gamma_w^2 = 1$, $\gamma_b^2 = 0$, $\sigma_w^2 = 1$ and $\sigma_b^2 = 1 - \mathbb{E}[\phi(Z)^2]$.

| $T =$ | 0 | 1 | 2 | 3 | 4 | 5 |
|---|---|---|---|---|---|---|
| ReLU | 43.944 | 77.277 | 93.192 | 93.192 | 95.403 | 95.403 |
| Tanh | 41.362 | 91.468 | 91.468 | 97.487 | 97.487 | 99.090 |
| Sigmoid | 91.557 | 99.729 | 99.729 | 99.977 | 99.977 | 99.997 |
| Gaussian | 95.834 | 95.834 | 98.729 | 98.729 | 99.634 | 99.634 |

However, the asymptotic rate of decay of the NTK coefficients varies significantly by activation function, due to the varying behavior of their tails. In Lemma 3.2 we choose ReLU, Tanh and Gaussian as prototypical examples of activations functions with growing, constant, and decaying tails respectively, and analyze the corresponding NTK coefficients in the two layer setting. For typographical ease we denote the zero mean Gaussian density function with variance $\sigma^2$ as $\omega_\sigma(z) := (1/\sqrt{2\pi\sigma^2})\exp\left(-z^2/(2\sigma^2)\right)$.

**Lemma 3.2.** *Under Assumptions 1 and 2,*

1. *if $\phi(z) = ReLU(z)$, then $\kappa_{p,2} = \delta_{(\gamma_b > 0) \cup (p \text{ even})}\Theta(p^{-3/2})$,*

2. *if $\phi(z) = Tanh(z)$, then $\kappa_{p,2} = \mathcal{O}\left(\exp\left(-\frac{\pi\sqrt{p-1}}{2}\right)\right)$,*

3. *if $\phi(z) = \omega_\sigma(z)$, then $\kappa_{p,2} = \delta_{(\gamma_b > 0) \cup (p \text{ even})}\Theta(p^{1/2}(\sigma^2 + 1)^{-p})$.*

The trend we observe from Lemma 3.2 is that activation functions whose Hermite coefficients decay quickly, such as $\omega_\sigma$, result in a faster decay of the NTK coefficients. We remark that analyzing the rates of decay for $l \geq 3$ is challenging due to the calculation of $F(p, k, \bar{\alpha}_{l-1})$ (4). In Appendix B.4 we provide preliminary results in this direction, upper bounding, in a very specific setting, the decay of the NTK coefficients for depths $l \geq 2$. Finally, we briefly pause here to highlight the potential for

using a truncation of (5) in order to perform efficient numerical approximation of the infinite width NTK. We remark that this idea is also addressed in a concurrent work by Han et al. (2022), albeit under a somewhat different set of assumptions [1]. As per our observations thus far that the coefficients of the NTK power series (5) typically decay quite rapidly, one might consider approximating $\Theta^{(l)}$ by computing just the first few terms in each series of (5). Figure 2 in Appendix B.3 displays the absolute error between the truncated ReLU NTK and the analytical expression for the ReLU NTK, which is also defined in Appendix B.3. Letting $\rho$ denote the input correlation then the key takeaway is that while for $|\rho|$ close to one the approximation is poor, for $|\rho| < 0.5$, which is arguably more realistic for real-world data, with just 50 coefficients machine level precision can be achieved. We refer the interested reader to Appendix B.3 for a proper discussion.

## 4 ANALYZING THE SPECTRUM OF THE NTK VIA ITS POWER SERIES

In this section, we consider a general kernel matrix power series of the form $n\mathbf{K} = \sum_{p=0}^{\infty} c_p (\mathbf{X}\mathbf{X}^T)^{\odot p}$ where $\{c_p\}_{p=0}^{\infty}$ are coefficients and $\mathbf{X}$ is the data matrix. According to Theorem 3.1, the coefficients of the NTK power series (5) are always nonnegative, thus we only consider the case where $c_p$ are nonnegative. We will also consider the kernel function power series, which we denote as $K(x_1, x_2) = \sum_{p=0}^{\infty} c_p \langle x_1, x_2 \rangle^p$. Later on we will analyze the spectrum of kernel matrix $\mathbf{K}$ and kernel function $K$.

### 4.1 ANALYSIS OF THE UPPER SPECTRUM AND EFFECTIVE RANK

In this section we analyze the upper part of the spectrum of the NTK, corresponding to the large eigenvalues, using the power series given in Theorem 3.1. Our first result concerns the *effective rank* (Huang et al., 2022) of the NTK. Given a positive semidefinite matrix $\mathbf{A} \in \mathbb{R}^{n \times n}$ we define the effective rank of $\mathbf{A}$ to be

$$\text{eff}(\mathbf{A}) = \frac{Tr(\mathbf{A})}{\lambda_1(\mathbf{A})}.$$

The effective rank quantifies how many eigenvalues are on the order of the largest eigenvalue. This follows from the Markov-like inequality

$$|\{p : \lambda_p(\mathbf{A}) \geq c\lambda_1(\mathbf{A})\}| \leq c^{-1}\text{eff}(\mathbf{A}) \tag{10}$$

and the eigenvalue bound

$$\frac{\lambda_p(\mathbf{A})}{\lambda_1(\mathbf{A})} \leq \frac{\text{eff}(\mathbf{A})}{p}.$$

Our first result is that the effective rank of the NTK can be bounded in terms of a ratio involving the power series coefficients. As we are assuming the data is normalized so that $\|\mathbf{x}_i\| = 1$ for all $i \in [n]$, then observe by the linearity of the trace

$$Tr(n\mathbf{K}) = \sum_{p=0}^{\infty} c_p Tr((\mathbf{X}\mathbf{X}^T)^{\odot p}) = n\sum_{p=0}^{\infty} c_p,$$

where we have used the fact that $Tr((\mathbf{X}\mathbf{X}^T)^{\odot p}) = n$ for all $p \in \mathbb{N}$. On the other hand,

$$\lambda_1(n\mathbf{K}) \geq \lambda_1(c_0(\mathbf{X}\mathbf{X}^T)^0) = \lambda_1(c_0\mathbf{1}_{n \times n}) = nc_0.$$

Combining these two results we get the following theorem.

**Theorem 4.1.** *Assume that we have a kernel Gram matrix $\mathbf{K}$ of the form $n\mathbf{K} = \sum_{p=0}^{\infty} c_p(\mathbf{X}\mathbf{X}^T)^{\odot p}$ where $c_0 \neq 0$. Furthermore, assume the input data $\mathbf{x}_i$ are normalized so that $\|\mathbf{x}_i\| = 1$ for all $i \in [n]$. Then*

$$\text{eff}(\mathbf{K}) \leq \frac{\sum_{p=0}^{\infty} c_p}{c_0}.$$

---

[1] In particular, in Han et al. (2022) the authors focus on homogeneous activation functions and allow the data to lie off the sphere. By contrast, we require the data to lie on the sphere but can handle non-homogeneous activation functions in the deep setting.

By Theorem 3.1 $c_0 \neq 0$ provided the network has biases or the activation function has nonzero Gaussian expectation (i.e., $\mu_0(\phi) \neq 0$). Thus we have that the effective rank of $\mathbf{K}$ is bounded by an $O(1)$ quantity. In the case of ReLU for example, as evidenced by Table 1, the effective rank will be roughly 2.3 for a shallow network. By contrast, a well-conditioned matrix would have an effective rank that is $\Omega(n)$. Combining Theorem 4.1 and the Markov-type bound (10) we make the following important observation.

**Observation 4.2.** *The largest eigenvalue $\lambda_1(\mathbf{K})$ of the NTK takes up an $\Omega(1)$ fraction of the entire trace and there are $O(1)$ eigenvalues on the same order of magnitude as $\lambda_1(\mathbf{K})$, where the $O(1)$ and $\Omega(1)$ notation are with respect to the parameter $n$.*

While the constant term $c_0 \mathbf{1}_{n \times n}$ in the kernel leads to a significant outlier in the spectrum of $\mathbf{K}$, it is rather uninformative beyond this. What interests us is how the structure of the data $\mathbf{X}$ manifests in the spectrum of the kernel matrix $\mathbf{K}$. For this reason we will examine the centered kernel matrix $\widetilde{\mathbf{K}} := \mathbf{K} - \frac{c_0}{n} \mathbf{1}_{n \times n}$. By a very similar argument as before we get the following result.

**Theorem 4.3.** *Assume that we have a kernel Gram matrix $\mathbf{K}$ of the form $n\mathbf{K} = \sum_{p=0}^{\infty} c_p (\mathbf{X}\mathbf{X}^T)^{\odot p}$ where $c_1 \neq 0$. Furthermore, assume the input data $\mathbf{x}_i$ are normalized so that $\|\mathbf{x}_i\| = 1$ for all $i \in [n]$. Then the centered kernel $\widetilde{\mathbf{K}} := \mathbf{K} - \frac{c_0}{n} \mathbf{1}_{n \times n}$ satisfies*

$$\mathrm{eff}(\widetilde{\mathbf{K}}) \leq \mathrm{eff}(\mathbf{X}\mathbf{X}^T) \frac{\sum_{p=1}^{\infty} c_p}{c_1}.$$

Thus we have that the effective rank of the centered kernel $\widetilde{\mathbf{K}}$ is upper bounded by a constant multiple of the effective rank of the input data Gram $\mathbf{X}\mathbf{X}^T$. Furthermore, we can take the ratio $\frac{\sum_{p=1}^{\infty} c_p}{c_1}$ as a measure of how much the NTK inherits the behavior of the linear kernel $\mathbf{X}\mathbf{X}^T$: in particular, if the input data gram has low effective rank and this ratio is moderate then we may conclude that the centered NTK must also have low effective rank. Again from Table 1, in the shallow setting we see that this ratio tends to be small for many of the common activations, for example, for ReLU it is roughly 1.3. To summarize then from Theorem 4.3 we make the important observation.

**Observation 4.4.** *Whenever the input data are approximately low rank, the centered kernel matrix $\widetilde{\mathbf{K}} = \mathbf{K} - \frac{c_0}{n} \mathbf{1}_{n \times n}$ is also approximately low rank.*

It turns out that this phenomenon also holds for finite-width networks at initialization. Consider the shallow model

$$\sum_{\ell=1}^{m} a_\ell \phi(\langle \mathbf{w}_\ell, \mathbf{x} \rangle),$$

where $\mathbf{x} \in \mathbb{R}^d$ and $\mathbf{w}_\ell \in \mathbb{R}^d$, $a_\ell \in \mathbb{R}$ for all $\ell \in [m]$. The following theorem demonstrates that when the width $m$ is linear in the number of samples $n$ then $\mathrm{eff}(\mathbf{K})$ is upper bounded by a constant multiple of $\mathrm{eff}(\mathbf{X}\mathbf{X}^T)$.

**Theorem 4.5.** *Assume $\phi(x) = ReLU(x)$ and $n \geq d$. Fix $\epsilon > 0$ small. Suppose that $\mathbf{w}_1, \ldots, \mathbf{w}_m \sim N(0, \nu_1^2 I_d)$ i.i.d. and $a_1, \ldots, a_m \sim N(0, \nu_2^2)$. Set $M = \max_{i \in [n]} \|\mathbf{x}_i\|_2$, and let*

$$\Sigma := \mathbb{E}_{\mathbf{w} \sim N(0, \nu_1^2 I)}[\phi(\mathbf{X}\mathbf{w})\phi(\mathbf{w}^T \mathbf{X}^T)].$$

*Then*

$$m = \Omega\left(\max(\lambda_1(\Sigma)^{-2}, 1) \max(n, \log(1/\epsilon))\right), \quad \nu_1 = O(1/M\sqrt{m})$$

*suffices to ensure that, with probability at least $1 - \epsilon$ over the sampling of the parameter initialization,*

$$\mathrm{eff}(\mathbf{K}) \leq C \cdot \mathrm{eff}(\mathbf{X}\mathbf{X}^T),$$

*where $C > 0$ is an absolute constant.*

Many works consider the model where the outer layer weights are fixed and have constant magnitude and only the inner layer weights are trained. This is the setting considered by Xie et al. (2017), Arora et al. (2019a), Du et al. (2019b), Oymak et al. (2019), Li et al. (2020), and Oymak & Soltanolkotabi (2020). In this setting we can reduce the dependence on the width $m$ to only be logarithmic in the number of samples $n$, and we have an accompanying lower bound. See Theorem C.5 in the Appendix C.2.3 for details.

In Figure 1 we empirically validate our theory by computing the spectrum of the NTK on both Caltech101 (Li et al., 2022) and isotropic Gaussian data for feedforward networks. We use the `functorch`[2] module in PyTorch (Paszke et al., 2019) using an algorithmic approach inspired by Novak et al. (2022). As per Theorem 4.1 and Observation 4.2, we observe all network architectures exhibit a dominant outlier eigenvalue due to the nonzero constant coefficient in the power series. Furthermore, this dominant outlier becomes more pronounced with depth, as can be observed if one carries out the calculations described in Theorem 3.1. Additionally, this outlier is most pronounced for ReLU, as the combination of its Gaussian mean plus bias term is the largest out of the activations considered here. As predicted by Theorem 4.3, Observation 4.4 and Theorem 4.5, we observe real-world data, which has a skewed spectrum and hence a low effective rank, results in the spectrum of the NTK being skewed. By contrast, isotropic Gaussian data has a flat spectrum, and as a result beyond the outlier the decay of eigenvalues of the NTK is more gradual. These observations support the claim that the NTK inherits its spectral structure from the data. We also observe that the spectrum for Tanh is closer to the linear activation relative to ReLU: intuitively this should not be surprising as close to the origin Tanh is well approximated by the identity. Our theory provides a formal explanation for this observation, indeed, the power series coefficients for Tanh networks decay quickly relative to ReLU. We provide further experimental results in Appendix C.3, including for CNNs where we observe the same trends. We note that the effective rank has implications for the generalization error. The Rademacher complexity of a kernel method (and hence the NTK model) within a parameter ball is determined by its its trace (Bartlett & Mendelson, 2002). Since for the NTK $\lambda_1(\mathbf{K}) = O(1)$, lower effective rank implies smaller trace and hence limited complexity.

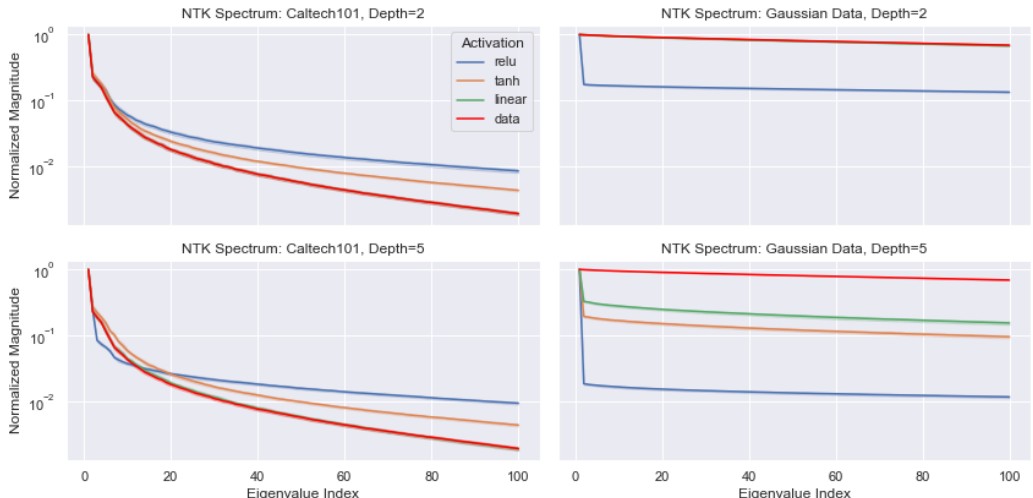

Figure 1: **(Feedforward NTK Spectrum)** We plot the normalized eigenvalues $\lambda_p/\lambda_1$ of the NTK Gram matrix $\mathbf{K}$ and the data Gram matrix $\mathbf{X}\mathbf{X}^T$ for Caltech101 and isotropic Gaussian datasets. To compute the NTK we randomly initialize feedforward networks of depths 2 and 5 with width 500. We use the standard parameterization and Pytorch's default Kaiming uniform initialization in order to better connect our results with what is used in practice. We consider a batch size of $n = 200$ and plot the first 100 eigenvalues. The thick part of each curve corresponds to the mean across 10 trials, while the transparent part corresponds to the 95% confidence interval

## 4.2 ANALYSIS OF THE LOWER SPECTRUM

In this section, we analyze the lower part of the spectrum using the power series. We first analyze the kernel function $K$ which we recall is a dot-product kernel of the form $K(x_1, x_2) = \sum_{p=0}^{\infty} c_p \langle x_1, x_2 \rangle^p$. Assuming the training data is uniformly distributed on a hypersphere it was shown by Basri et al. (2019); Bietti & Mairal (2019) that the eigenfunctions of $K$ are the spherical

---

[2]https://pytorch.org/functorch/stable/notebooks/neural_tangent_kernels.html

harmonics. Azevedo & Menegatto (2015) gave the eigenvalues of the kernel $K$ in terms of the power series coefficients.

**Theorem 4.6.** *[Azevedo & Menegatto (2015)] Let $\Gamma$ denote the gamma function. Suppose that the training data are uniformly sampled from the unit hypersphere $\mathbb{S}^d$, $d \geq 2$. If the dot-product kernel function has the expansion $K(x_1, x_2) = \sum_{p=0}^{\infty} c_p \langle x_1, x_2 \rangle^p$ where $c_p \geq 0$, then the eigenvalue of every spherical harmonic of frequency $k$ is given by*

$$\overline{\lambda_k} = \frac{\pi^{d/2}}{2^{k-1}} \sum_{\substack{p \geq k \\ p-k \text{ is even}}} c_p \frac{\Gamma(p+1)\Gamma(\frac{p-k+1}{2})}{\Gamma(p-k+1)\Gamma(\frac{p-k+1}{2}+k+d/2)}.$$

A proof of Theorem 4.6 is provided in Appendix C.4 for the reader's convenience. This theorem connects the coefficients $c_p$ of the kernel power series with the eigenvalues $\overline{\lambda_k}$ of the kernel. In particular, given a specific decay rate for the coefficients $c_p$ one may derive the decay rate of $\overline{\lambda_k}$: for example, Scetbon & Harchaoui (2021) examined the decay rate of $\overline{\lambda_k}$ if $c_p$ admits a polynomial decay or exponential decay. The following Corollary summarizes the decay rates of $\overline{\lambda_k}$ corresponding to two layer networks with different activations.

**Corollary 4.7.** *Under the same setting as in Theorem 4.6,*

1. *if $c_p = \Theta(p^{-a})$ where $a \geq 1$, then $\overline{\lambda_k} = \Theta(k^{-d-2a+2})$,*

2. *if $c_p = \delta_{(p \text{ even})}\Theta(p^{-a})$, then $\overline{\lambda_k} = \delta_{(k \text{ even})}\Theta(k^{-d-2a+2})$,*

3. *if $c_p = \mathcal{O}\left(\exp\left(-a\sqrt{p}\right)\right)$, then $\overline{\lambda_k} = \mathcal{O}\left(k^{-d+1/2}\exp\left(-a\sqrt{k}\right)\right)$,*

4. *if $c_p = \Theta(p^{1/2}a^{-p})$, then $\overline{\lambda_k} = \mathcal{O}\left(k^{-d+1}a^{-k}\right)$ and $\overline{\lambda_k} = \Omega\left(k^{-d/2+1}2^{-k}a^{-k}\right)$.*

In addition to recovering existing results for ReLU networks Basri et al. (2019); Velikanov & Yarotsky (2021); Geifman et al. (2020); Bietti & Bach (2021), Corollary 4.7 also provides the decay rates for two-layer networks with Tanh and Gaussian activations. As faster eigenvalue decay implies a smaller RKHS Corollary 4.7 shows using ReLU results in a larger RKHS relative to Tanh or Gaussian activations. Numerics for Corollary 4.7 are provided in Figure 4 in Appendix C.3. Finally, in Theorem 4.8 we relate a kernel's power series to its spectral decay for arbitrary data distributions.

**Theorem 4.8** (Informal). *Let the rows of $\mathbf{X} \in \mathbb{R}^{n \times d}$ be arbitrary points on the unit sphere. Consider the kernel matrix $n\mathbf{K} = \sum_{p=0}^{\infty} c_p \left(\mathbf{X}\mathbf{X}^T\right)^{\odot p}$ and let $r(n) \leq d$ denote the rank of $\mathbf{X}\mathbf{X}^T$. Then*

1. *if $c_p = \mathcal{O}(p^{-\alpha})$ with $\alpha > r(n) + 1$ for all $n \in \mathbb{Z}_{\geq 0}$ then $\lambda_n(\mathbf{K}) = \mathcal{O}\left(n^{-\frac{\alpha-1}{r(n)}}\right)$,*

2. *if $c_p = \mathcal{O}(e^{-\alpha\sqrt{p}})$ then $\lambda_n(\mathbf{K}) = \mathcal{O}\left(n^{\frac{1}{2r(n)}}\exp\left(-\alpha'n^{\frac{1}{2r(n)}}\right)\right)$ for any $\alpha' < \alpha 2^{-1/2r(n)}$,*

3. *if $c_p = \mathcal{O}(e^{-\alpha p})$ then $\lambda_n(\mathbf{K}) = \mathcal{O}\left(\exp\left(-\alpha'n^{\frac{1}{r(n)}}\right)\right)$ for any $\alpha' < \alpha 2^{-1/2r(n)}$.*

Although the presence of the factor $1/r(n)$ in the exponents of $n$ in these bounds is a weakness, Theorem 4.8 still illustrates how, in a highly general setting, the asymptotic decay of the coefficients of the power series ensures a certain asymptotic decay in the eigenvalues of the kernel matrix. A formal version of this result is provided in Appendix C.5 along with further discussion.

## 5  CONCLUSION

Using a power series expansion we derived a number of insights into both the outliers as well as the asymptotic decay of the spectrum of the NTK, in particular highlighting the role of the activation function. We performed our analysis without recourse to a high dimensional limit or the use of random matrix theory. Interesting avenues for future work include better analyzing the role of depth as well as characterizing the outlier eigenvalues and spectrum as a whole for networks with convolutional, residual or transformer layers.

**Acknowledgements and Disclosure of Funding:** This project has been supported by ERC Grant 757983 and NSF CAREER Grant DMS-2145630.

**Reproducibility Statement:** To ensure reproducibility, we make the code public at `https://github.com/bbowman223/data_ntk`.

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

## APPENDIX

The appendix is organized as follows.

- Appendix A gives background material on Gaussan kernels, NTK, unit variance intitialization, and Hermite polynomial expansions.

- Appendix B provides details for Section 3.

- Appendix C provides details for Section 4.

## A  BACKGROUND MATERIAL

### A.1  GAUSSIAN KERNEL

Observe by construction that the flattened collection of preactivations at the first layer $(g^{(1)}(\mathbf{x}_i))_{i=1}^n$ form a centered Gaussian process, with the covariance between the $\alpha$th and $\beta$th neuron being described by

$$\Sigma_{\alpha\beta}^{(1)}(\mathbf{x}_i, \mathbf{x}_j) := \mathbb{E}[g_\alpha^{(1)}(\mathbf{x}_i)g_\beta^{(1)}(\mathbf{x}_j)] = \delta_{\alpha=\beta}\left(\gamma_w^2 \mathbf{x}_i^T \mathbf{x}_j + \gamma_b^2\right).$$

Under the Assumption 1, the preactivations at each layer $l \in [L+1]$ converge also in distribution to centered Gaussian processes (Neal, 1996; Lee et al., 2018). We remark that the sequential width limit condition of Assumption 1 is not necessary for this behavior, for example the same result can be derived in the setting where the widths of the network are sent to infinity simultaneously under certain conditions on the activation function (de G. Matthews et al., 2018). However, as our interests lie in analyzing the limit rather than the conditions for convergence to said limit, for simplicity we consider only the sequential width limit. As per Lee et al. (2018, Eq. 4), the covariance between the preactivations of the $\alpha$th and $\beta$th neurons at layer $l \geq 2$ for any input pair $\mathbf{x}, \mathbf{y} \in \mathbb{R}$ are described by the following kernel,

$$\begin{aligned}\Sigma_{\alpha\beta}^{(l)}(\mathbf{x}, \mathbf{y}) &:= \mathbb{E}[g_\alpha^{(l)}(\mathbf{x})g_\beta^{(l)}(\mathbf{y})] \\ &= \delta_{\alpha=\beta}\left(\sigma_w^2 \mathbb{E}_{g^{(l-1)} \sim \mathcal{GP}(0, \Sigma^{l-1})}[\phi(g_\alpha^{(l-1)}(\mathbf{x}))\phi(g_\beta^{(l-1)}(\mathbf{y}))] + \sigma_b^2\right).\end{aligned}$$

We refer to this kernel as the Gaussian kernel. As each neuron is identically distributed and the covariance between pairs of neurons is 0 unless $\alpha = \beta$, moving forward we drop the subscript and discuss only the covariance between the preactivations of an arbitrary neuron given two inputs. As per the discussion by Lee et al. (2018, Section 2.3), the expectations involved in the computation of these Gaussian kernels can be computed with respect to a bivariate Gaussian distribution, whose covariance matrix has three distinct entries: the variance of a preactivation of $\mathbf{x}$ at the previous layer, $\Sigma^{(l-1)}(\mathbf{x}, \mathbf{x})$, the variance of a preactivation of $\mathbf{y}$ at the previous layer, $\Sigma^{(l)}(\mathbf{y}, \mathbf{y})$, and the covariance between preactivations of $\mathbf{x}$ and $\mathbf{y}$, $\Sigma^{(l-1)}(\mathbf{x}, \mathbf{y})$. Therefore the Gaussian kernel, or covariance function, and its derivative, which we will require later for our analysis of the NTK, can be computed via the the following recurrence relations, see for instance (Lee et al., 2018; Jacot et al., 2018; Arora et al., 2019b; Nguyen et al., 2021),

$$\begin{aligned}\Sigma^{(1)}(\mathbf{x}, \mathbf{y}) &= \gamma_w^2 \mathbf{x}^T \mathbf{x} + \gamma_b^2, \\ \mathbf{A}^{(l)}(\mathbf{x}, \mathbf{y}) &= \begin{bmatrix} \Sigma^{(l-1)}(\mathbf{x}, \mathbf{x}) & \Sigma^{(l-1)}(\mathbf{x}, \mathbf{y}) \\ \Sigma^{(l-1)}(\mathbf{y}, \mathbf{x}) & \Sigma^{(l-1)}(\mathbf{x}, \mathbf{x}) \end{bmatrix} \\ \Sigma^{(l)}(\mathbf{x}, \mathbf{y}) &= \sigma_w^2 \mathbb{E}_{(B_1, B_2) \sim \mathcal{N}(0, \mathbf{A}^{(l)}(\mathbf{x}, \mathbf{y}))}[\phi(B_1)\phi(B_2)] + \sigma_b^2, \\ \dot{\Sigma}^{(l)}(\mathbf{x}, \mathbf{y}) &= \sigma_w^2 \mathbb{E}_{(B_1, B_2) \sim \mathcal{N}(0, \mathbf{A}^{(l)}(\mathbf{x}, \mathbf{y}))}[\phi'(B_1)\phi'(B_2)].\end{aligned} \quad (11)$$

### A.2  NEURAL TANGENT KERNEL (NTK)

As discussed in the Section 1, under Assumption 1 $\tilde{\Theta}^{(l)}$ converges in probability to a deterministic limit, which we denote $\Theta^{(l)}$. This deterministic limit kernel can be expressed in terms of the Gaussian kernels and their derivatives from Section A.1 via the following recurrence relationships (Jacot

et al., 2018, Theorem 1),

$$
\begin{aligned}
\Theta^{(1)}(\mathbf{x}, \mathbf{y}) &= \Sigma^{(1)}(\mathbf{x}, \mathbf{y}), \\
\Theta^{(l)}(\mathbf{x}, \mathbf{y}) &= \Theta^{(l-1)}(\mathbf{x}, \mathbf{y})\dot{\Sigma}^{(l)}(\mathbf{x}, \mathbf{y}) + \Sigma^{(l)}(\mathbf{x}, \mathbf{y}) \\
&= \Sigma^{(l)}(\mathbf{x}, \mathbf{y}) + \sum_{h=1}^{l-1} \Sigma^{(h)}(\mathbf{x}, \mathbf{y}) \left( \prod_{h'=h+1}^{l} \dot{\Sigma}^{(h')}(\mathbf{x}, \mathbf{y}) \right) \ \forall l \in [2, L+1].
\end{aligned}
\tag{12}
$$

A useful expression for the NTK matrix, which is a straightforward extension and generalization of Nguyen et al. (2021, Lemma 3.1), is provided in Lemma A.1 below.

**Lemma A.1.** *(Based on Nguyen et al. 2021, Lemma 3.1) Under Assumption 1, a sequence of positive semidefinite matrices $(\mathbf{G}_l)_{l=1}^{L+1}$ in $\mathbb{R}^{n \times n}$, and the related sequence $(\dot{\mathbf{G}}_l)_{l=2}^{L+1}$ also in $\mathbb{R}^{n \times n}$, can be constructed via the following recurrence relationships,*

$$
\begin{aligned}
\mathbf{G}_1 &= \gamma_w^2 \mathbf{X}\mathbf{X}^T + \gamma_b^2 \boldsymbol{I}_{n \times n}, \\
\mathbf{G}_2 &= \sigma_w^2 \mathbb{E}_{\mathbf{w} \sim \mathcal{N}(\boldsymbol{0}, \boldsymbol{I}_d)}[\phi(\mathbf{X}\mathbf{w})\phi(\mathbf{X}\mathbf{w})^T] + \sigma_b^2 \boldsymbol{I}_{n \times n}, \\
\dot{\mathbf{G}}_2 &= \sigma_w^2 \mathbb{E}_{\mathbf{w} \sim \mathcal{N}(\boldsymbol{0}, \boldsymbol{I}_n)}[\phi'(\mathbf{X}\mathbf{w})\phi'(\mathbf{X}\mathbf{w})^T], \\
\mathbf{G}_l &= \sigma_w^2 \mathbb{E}_{\mathbf{w} \sim \mathcal{N}(\boldsymbol{0}, \boldsymbol{I}_n)}[\phi(\sqrt{\mathbf{G}_{l-1}}\mathbf{w})\phi(\sqrt{\mathbf{G}_{l-1}}\mathbf{w})^T] + \sigma_b^2 \boldsymbol{I}_{n \times n}, \ l \in [3, L+1], \\
\dot{\mathbf{G}}_l &= \sigma_w^2 \mathbb{E}_{\mathbf{w} \sim \mathcal{N}(\boldsymbol{0}, \boldsymbol{I}_n)}[\phi'(\sqrt{\mathbf{G}_{l-1}}\mathbf{w})\phi'(\sqrt{\mathbf{G}_{l-1}}\mathbf{w})^T], \ l \in [3, L+1].
\end{aligned}
\tag{13}
$$

*The sequence of NTK matrices $(\mathbf{K}_l)_{l=1}^{L+1}$ can in turn be written using the following recurrence relationship,*

$$
\begin{aligned}
n\mathbf{K}_1 &= \mathbf{G}_1, \\
n\mathbf{K}_l &= \mathbf{G}_l + n\mathbf{K}_{l-1} \odot \dot{\mathbf{G}}_l \\
&= \mathbf{G}_l + \sum_{i=1}^{l-1} \left( \mathbf{G}_i \odot \left( \odot_{j=i+1}^{l} \dot{\mathbf{G}}_j \right) \right).
\end{aligned}
\tag{14}
$$

*Proof.* For the sequence $(\mathbf{G}_l)_{l=1}^{L+1}$ it suffices to prove for any $i, j \in [n]$ and $l \in [L+1]$ that

$$
[\mathbf{G}_l]_{i,j} = \Sigma^{(l)}(\mathbf{x}_i, \mathbf{x}_j)
$$

and $\mathbf{G}_l$ is positive semi-definite. We proceed by induction, considering the base case $l = 1$ and comparing (13) with (11) then it is evident that

$$
[\mathbf{G}_1]_{i,j} = \Sigma^{(1)}(\mathbf{x}_i, \mathbf{x}_j).
$$

In addition, $\mathbf{G}_1$ is also clearly positive semi-definite as for any $\mathbf{u} \in \mathbb{R}^n$

$$
\mathbf{u}^T \mathbf{G}_1 \mathbf{u} = \gamma_w^2 \left\| \mathbf{X}^T \mathbf{u} \right\|^2 + \gamma_b^2 \left\| \mathbf{1}_n^T \mathbf{u} \right\|^2 \geq 0.
$$

We now assume the induction hypothesis is true for $\mathbf{G}_{l-1}$. We will need to distinguish slightly between two cases, $l = 2$ and $l \in [3, L+1]$. The proof of the induction step in either case is identical. To this end, and for notational ease, let $\mathbf{V} = \mathbf{X}$, $\mathbf{w} \sim \mathcal{N}(0, \mathbf{I}_d)$ when $l = 2$, and $\mathbf{V} = \sqrt{\mathbf{G}_{l-1}}$, $\mathbf{w} \sim \mathcal{N}(0, \mathbf{I}_n)$ for $l \in [3, L+1]$. In either case we let $\mathbf{v}_i$ denote the $i$th row of $\mathbf{V}$. For any $i, j \in [n]$

$$
[\mathbf{G}_l]_{ij} = \sigma_w^2 \mathbb{E}_{\mathbf{w}}[\phi(\mathbf{v}_i^T \mathbf{w})\phi(\mathbf{v}_j^T \mathbf{w})] + \sigma_b^2.
$$

Now let $B_1 = \mathbf{v}_i^T \mathbf{w}$, $B_2 = \mathbf{v}_j^T \mathbf{w}$ and observe for any $\alpha_1, \alpha_2 \in \mathbb{R}$ that $\alpha_1 B_1 + \alpha_2 B_2 = \sum_k^n (\alpha_1 v_{ik} + \alpha_2 v_{jk})w_k \sim \mathcal{N}(0, \|\alpha_1 \mathbf{v}_i + \alpha_2 \mathbf{v}_j\|^2)$. Therefore the joint distribution of $(B_1, B_2)$ is a mean 0 bivariate normal distribution. Denoting the covariance matrix of this distribution as $\tilde{\mathbf{A}} \in \mathbb{R}^{2 \times 2}$, then $[\mathbf{G}_l]_{ij}$ can be expressed as

$$
[\mathbf{G}_l]_{ij} = \sigma_w^2 \mathbb{E}_{(B_1, B_2) \sim \tilde{\mathbf{A}}}[\phi(B_1)\phi(B_2)] + \sigma_b^2.
$$

To prove $[\mathbf{G}_l]_{i,j} = \Sigma^{(l)}$ it therefore suffices to show that $\tilde{\mathbf{A}} = \mathbf{A}^{(l)}$ as per (11). This follows by the induction hypothesis as

$$\mathbb{E}[B_1^2] = \mathbf{v}_i^T \mathbf{v}_i = [\mathbf{G}_{l-1}]_{ii} = \Sigma^{(l-1)}(\mathbf{x}_i, \mathbf{x}_i),$$
$$\mathbb{E}[B_2^2] = \mathbf{v}_j^T \mathbf{v}_j = [\mathbf{G}_{l-1}]_{jj} = \Sigma^{(l-1)}(\mathbf{x}_j, \mathbf{x}_j),$$
$$\mathbb{E}[B_1 B_2] = \mathbf{v}_i^T \mathbf{v}_j = [\mathbf{G}_{l-1}]_{ij} = \Sigma^{(l-1)}(\mathbf{x}_i, \mathbf{x}_j).$$

Finally, $\mathbf{G}_l$ is positive semi-definite as long as $\mathbb{E}_{\mathbf{w}}[\phi(\mathbf{V}\mathbf{w})\phi(\mathbf{V}\mathbf{w})^T]$ is positive semi-definite. Let $M(\mathbf{w}) = \phi(\mathbf{V}\mathbf{w}) \in \mathbb{R}^{n \times n}$ and observe for any $\mathbf{w}$ that $M(\mathbf{w})M(\mathbf{w})^T$ is positive semi-definite. Therefore $\mathbb{E}_{\mathbf{w}}[M(\mathbf{w})M(\mathbf{w})^T]$ must also be positive semi-definite. Thus the inductive step is complete and we may conclude for $l \in [L+1]$ that

$$[\mathbf{G}_l]_{i,j} = \Sigma^{(l)}(\mathbf{x}_i, \mathbf{x}_j). \tag{15}$$

For the proof of the expression for the sequence $(\dot{\mathbf{G}}_l)_{l=2}^{L+1}$ it suffices to prove for any $i, j \in [n]$ and $l \in [L+1]$ that

$$[\dot{\mathbf{G}}_l]_{i,j} = \dot{\Sigma}^{(l)}(\mathbf{x}_i, \mathbf{x}_j).$$

By comparing (13) with (11) this follows immediately from (15). Therefore with (13) proven (14) follows from (12). $\qquad\square$

## A.3 Unit variance initialization

The initialization scheme for a neural network, particularly a deep neural network, needs to be designed with some care in order to avoid either vanishing or exploding gradients during training Glorot & Bengio (2010); He et al. (2015); Mishkin & Matas (2016); LeCun et al. (2012). Some of the most popular initialization strategies used in practice today, in particular LeCun et al. (2012) and Glorot & Bengio (2010) initialization, first model the preactivations of the network as Gaussian random variables and then select the network hyperparameters in order that the variance of these idealized preactivations is fixed at one. Under Assumption 1 this idealized model on the preactivations is actually realized and if we additionally assume the conditions of Assumption 2 hold then likewise the variance of the preactivations at every layer will be fixed at one. To this end, and as in Poole et al. (2016); Murray et al. (2022), consider the function $V : \mathbb{R}_{\geq 0} \to \mathbb{R}_{\geq 0}$ defined as

$$V(q) = \sigma_w^2 \mathbb{E}_{Z \sim \mathcal{N}(0,1)} \left[ \phi\left(\sqrt{q}Z\right)^2 \right] + \sigma_b^2. \tag{16}$$

Noting that $V$ is another expression for $\Sigma^{(l)}(\mathbf{x}, \mathbf{x})$, derived via a change of variables as per Poole et al. (2016), the sequence of variances $(\Sigma^{(l)}(\mathbf{x}, \mathbf{x}))_{l=2}^L$ can therefore be generated as follows,

$$\Sigma^{(l)}(\mathbf{x}, \mathbf{x}) = V(\Sigma^{(l-1)}(\mathbf{x}, \mathbf{x})). \tag{17}$$

The linear correlation $\rho^{(l)} : \mathbb{R}^d \times \mathbb{R}^d \to [-1, 1]$ between the preactivations of two inputs $\mathbf{x}, \mathbf{y} \in \mathbb{R}^d$ we define as

$$\rho^{(l)}(\mathbf{x}, \mathbf{y}) = \frac{\Sigma^{(l)}(\mathbf{x}, \mathbf{y})}{\sqrt{\Sigma^{(l)}(\mathbf{x}, \mathbf{x})\Sigma^{(l)}(\mathbf{y}, \mathbf{y})}}. \tag{18}$$

Assuming $\Sigma^{(l)}(\mathbf{x}, \mathbf{x}) = \Sigma^{(l)}(\mathbf{y}, \mathbf{y}) = 1$ for all $l \in [L+1]$, then $\rho^{(l)}(\mathbf{x}, \mathbf{y}) = \Sigma^{(l)}(\mathbf{x}, \mathbf{y})$. Again as in Murray et al. (2022) and analogous to (16), with $Z_1, Z_2 \sim \mathcal{N}(0, 1)$ independent, $U_1 := Z_1$, $U_2(\rho) := (\rho Z_1 + \sqrt{1 - \rho^2} Z_2)$ [3] we define the correlation function $R : [-1, 1] \to [-1, 1]$ as

$$R(\rho) = \sigma_w^2 \mathbb{E}[\phi(U_1)\phi(U_2(\rho))] + \sigma_b^2. \tag{19}$$

Noting under these assumptions that $R$ is equivalent to $\Sigma^{(l)}(\mathbf{x}, \mathbf{y})$, the sequence of correlations $(\rho^{(l)}(\mathbf{x}, \mathbf{y}))_{l=2}^L$ can thus be generated as

$$\rho^{(l)}(\mathbf{x}, \mathbf{y}) = R(\rho^{(l-1)}(\mathbf{x}, \mathbf{y})).$$

As observed in Poole et al. (2016); Schoenholz et al. (2017), $R(1) = V(1) = 1$, hence $\rho = 1$ is a fixed point of $R$. We remark that as all preactivations are distributed as $\mathcal{N}(0, 1)$, then a correlation

---

[3] We remark that $U_1, U_2$ are dependent and identically distributed as $U_1, U_2 \sim \mathcal{N}(0, 1)$.

of one between preactivations implies they are equal. The stability of the fixed point $\rho = 1$ is of particular significance in the context of initializing deep neural networks successfully. Under mild conditions on the activation function one can compute the derivative of $R$, see e.g., Poole et al. (2016); Schoenholz et al. (2017); Murray et al. (2022), as follows,

$$R'(\rho) = \sigma_w^2 \mathbb{E}[\phi'(U_1)\phi'(U_2(\rho))]. \tag{20}$$

Observe that the expression for $\dot{\Sigma}^{(l)}$ and $R'$ are equivalent via a change of variables (Poole et al., 2016), and therefore the sequence of correlation derivatives may be computed as

$$\dot{\Sigma}^{(l)}(\mathbf{x}, \mathbf{y}) = R'(\rho^{(l)}(\mathbf{x}, \mathbf{y})).$$

With the relevant background material now in place we are in a position to prove Lemma A.2.

**Lemma A.2.** *Under Assumptions 1 and 2 and defining $\chi = \sigma_w^2 \mathbb{E}_{Z \sim \mathcal{N}(0,1)}[\phi'(Z)^2] \in \mathbb{R}_{>0}$, then for all $i, j \in [n]$, $l \in [L+1]$*

- $[\mathbf{G}_{n,l}]_{ij} \in [-1, 1]$ *and* $[\mathbf{G}_{n,l}]_{ii} = 1$,

- $[\dot{\mathbf{G}}_{n,l}]_{ij} \in [-\chi, \chi]$ *and* $[\dot{\mathbf{G}}_{n,l}]_{ii} = \chi$.

*Furthermore, the NTK is a dot product kernel, meaning $\Theta(\mathbf{x}_i, \mathbf{x}_j)$ can be written as a function of the inner product between the two inputs, $\Theta(\mathbf{x}_i^T \mathbf{x}_j)$.*

*Proof.* Recall from Lemma A.1 and its proof that for any $l \in [L+1]$, $i, j \in [n]$ $[\mathbf{G}_{n,l}]_{ij} = \Sigma^{(l)}(\mathbf{x}_i, \mathbf{x}_j)$ and $[\dot{\mathbf{G}}_{n,l}]_{ij} = \dot{\Sigma}^{(l)}(\mathbf{x}_i, \mathbf{x}_j)$. We first prove by induction $\Sigma^{(l)}(\mathbf{x}_i, \mathbf{x}_i) = 1$ for all $l \in [L+1]$. The base case $l = 1$ follows as

$$\Sigma^{(1)}(\mathbf{x}, \mathbf{x}) = \gamma_w^2 \mathbf{x}^T \mathbf{x} + \gamma_b^2 = \gamma_w^2 + \gamma_b^2 = 1.$$

Assume the induction hypothesis is true for layer $l - 1$. With $Z \sim \mathcal{N}(0, 1)$, then from (16) and (17)

$$\begin{aligned}
\Sigma^{(l)}(\mathbf{x}, \mathbf{x}) &= V(\Sigma^{(l-1)}(\mathbf{x}, \mathbf{x})) \\
&= \sigma_w^2 \mathbb{E}\left[\phi^2\left(\sqrt{\Sigma^{(l-1)}(\mathbf{x}, \mathbf{x})} Z\right)\right] + \sigma_b^2 \\
&= \sigma_w^2 \mathbb{E}\left[\phi^2(Z)\right] + \sigma_b^2 \\
&= 1,
\end{aligned}$$

thus the inductive step is complete. As an immediate consequence it follows that $[\mathbf{G}_l]_{ii} = 1$. Also, for any $i, j \in [n]$ and $l \in [L+1]$,

$$\Sigma^{(l)}(\mathbf{x}_i, \mathbf{x}_j) = \rho^{(l)}(\mathbf{x}_i, \mathbf{x}_j) = R(\rho^{(l-1)}(\mathbf{x}_i, \mathbf{x}_j)) = R(...R(R(\mathbf{x}_i^T \mathbf{x}_j))).$$

Thus we can consider $\Sigma^{(l)}$ as a univariate function of the input correlation $\Sigma : [-1, 1] \to [-1, 1]$ and also conclude that $[\mathbf{G}_l]_{ij} \in [-1, 1]$. Furthermore,

$$\dot{\Sigma}^{(l)}(\mathbf{x}_i, \mathbf{x}_j) = R'(\rho^{(l)}(\mathbf{x}_i, \mathbf{x}_j)) = R'(R(...R(R(\mathbf{x}_i^T \mathbf{x}_j)))),$$

which likewise implies $\dot{\Sigma}$ is a dot product kernel. Recall now the random variables introduced to define $R$: $Z_1, Z_2 \sim \mathcal{N}(0, 1)$ are independent and $U_1 = Z_1$, $U_2 = (\rho Z_1 + \sqrt{1 - \rho^2} Z_2)$. Observe $U_1, U_2$ are dependent but identically distributed as $U_1, U_2 \sim \mathcal{N}(0, 1)$. For any $\rho \in [-1, 1]$ then applying the Cauchy-Schwarz inequality gives

$$|R'(\rho)|^2 = \sigma_w^4 \left|\mathbb{E}[\phi'(U_1)\phi'(U_2)]\right|^2 \leq \sigma_w^4 \mathbb{E}[\phi'(U_1)^2]\mathbb{E}[\phi'(U_2)^2] = \sigma_w^4 \mathbb{E}[\phi'(U_1)^2]^2 = |R'(1)|^2.$$

As a result, under the assumptions of the lemma $\dot{\Sigma}^{(l)} : [-1, 1] \to [-\chi, \chi]$ and $\dot{\Sigma}^{(l)}(\mathbf{x}_i, \mathbf{x}_i) = \chi$. From this it immediately follows that $[\dot{\mathbf{G}}_l]_{ij} \in [-\chi, \chi]$ and $[\dot{\mathbf{G}}_l]_{ii} = \chi$ as claimed. Finally, as $\Sigma : [-1, 1] \to [-1, 1]$ and $\dot{\Sigma} : [-1, 1] \to [-\chi, \chi]$ are dot product kernels, then from (12) the NTK must also be a dot product kernel and furthermore a univariate function of the pairwise correlation of its input arguments. $\square$

The following corollary, which follows immediately from Lemma A.2 and (14), characterizes the trace of the NTK matrix in terms of the trace of the input gram.

**Corollary A.3.** *Under the same conditions as Lemma A.2, suppose $\phi$ and $\sigma_w^2$ are chosen such that $\chi = 1$. Then*

$$Tr(\mathbf{K}_{n,l}) = l. \tag{21}$$

## A.4 HERMITE EXPANSIONS

We say that a function $f : \mathbb{R} \to \mathbb{R}$ is square integrable w.r.t. the standard Gaussian measure $\gamma = e^{-x^2/2}/\sqrt{2\pi}$ if $\mathbb{E}_{x \sim \mathcal{N}(0,1)}[f(x)^2] < \infty$. We denote by $L^2(\mathbb{R}, \gamma)$ the space of all such functions. The probabilist's Hermite polynomials are given by

$$H_k(x) = (-1)^k e^{x^2/2} \frac{d^k}{dx^k} e^{-x^2/2}, \quad k = 0, 1, \dots.$$

The first three Hermite polynomials are $H_0(x) = 1$, $H_1(x) = x$, $H_2(x) = (x^2 - 1)$. Let $h_k(x) = \frac{H_k(x)}{\sqrt{k!}}$ denote the normalized probabilist's Hermite polynomials. The normalized Hermite polynomials form a complete orthonormal basis in $L^2(\mathbb{R}, \gamma)$ (O'Donnell, 2014, §11): in all that follows, whenever we reference the Hermite polynomials, we will be referring to the normalized Hermite polynomials. The Hermite expansion of a function $\phi \in L^2(\mathbb{R}, \gamma)$ is given by

$$\phi(x) = \sum_{k=0}^{\infty} \mu_k(\phi) h_k(x), \tag{22}$$

where

$$\mu_k(\phi) = \mathbb{E}_{X \sim \mathcal{N}(0,1)}[\phi(X) h_k(X)] \tag{23}$$

is the $k$th normalized probabilist's Hermite coefficient of $\phi$. In what follows we shall make use of the following identities.

$$\forall k \geq 1, \; h_k'(x) = \sqrt{k} h_{k-1}(x), \tag{24}$$

$$\forall k \geq 1, \; x h_k(x) = \sqrt{k+1} h_{k+1}(x) + \sqrt{k} h_{k-1}(x). \tag{25}$$

$$h_k(0) = \begin{cases} 0, & \text{if } k \text{ is odd} \\ \frac{1}{\sqrt{k!}}(-1)^{\frac{k}{2}}(k-1)!! & \text{if } k \text{ is even} \end{cases},$$

$$\text{where } k!! = \begin{cases} 1, & k \leq 0 \\ k \cdot (k-2) \cdots 5 \cdot 3 \cdot 1, & k > 0 \text{ odd} \\ k \cdot (k-2) \cdots 6 \cdot 4 \cdot 2, & k > 0 \text{ even} \end{cases}. \tag{26}$$

We also remark that the more commonly encountered physicist's Hermite polynomials, which we denote $\tilde{H}_k$, are related to the normalized probablist's polynomials as follows,

$$h_k(z) = \frac{2^{-k/2} \tilde{H}_k(z/\sqrt{2})}{\sqrt{k!}}.$$

The Hermite expansion of the activation function deployed will play a key role in determining the coefficients of the NTK power series. In particular, the Hermite coefficients of ReLU are as follows.

**Lemma A.4.** *Daniely et al. (2016) For $\phi(z) = \max\{0, z\}$ the Hermite coefficients are given by*

$$\mu_k(\phi) = \begin{cases} 1/\sqrt{2\pi}, & k = 0, \\ 1/2, & k = 1, \\ (k-3)!!/\sqrt{2\pi k!}, & k \text{ even and } k \geq 2, \\ 0, & k \text{ odd and } k > 3. \end{cases} \tag{27}$$

## B EXPRESSING THE NTK AS A POWER SERIES

### B.1 DERIVING A POWER SERIES FOR THE NTK

We will require the following minor adaptation of Nguyen & Mondelli (2020, Lemma D.2). We remark this result was first stated for ReLU and Softplus activations in the work of Oymak & Soltanolkotabi (2020, Lemma H.2).

**Lemma B.1.** *For arbitrary $n, d \in \mathbb{N}$, let $\mathbf{A} \in \mathbb{R}^{n \times d}$. For $i \in [n]$, we denote the ith row of $\mathbf{A}$ as $\mathbf{a}_i$, and further assume that $\|\mathbf{a}_i\| = 1$. Let $\phi : \mathbb{R} \to \mathbb{R}$ satisfy $\phi \in L^2(\mathbb{R}, \gamma)$ and define*

$$\mathbf{M} = \mathbb{E}_{\mathbf{w} \sim \mathcal{N}(0, I_n)}[\phi(\mathbf{A}\mathbf{w})\phi(\mathbf{A}\mathbf{w})^T] \in \mathbb{R}^{n \times n}.$$

*Then the matrix series*

$$\mathbf{S}_K = \sum_{k=0}^{K} \mu_k^2(\phi) \left(\mathbf{A}\mathbf{A}^T\right)^{\odot k}$$

*converges uniformly to $\mathbf{M}$ as $K \to \infty$.*

The proof of Lemma B.1 follows exactly as in (Nguyen & Mondelli, 2020, Lemma D.2), and is in fact slightly simpler due to the fact we assume the rows of $\mathbf{A}$ are unit length and $\mathbf{w} \sim \mathcal{N}(0, \mathbf{I}_d)$ instead of $\sqrt{d}$ and $\mathbf{w} \sim \mathcal{N}(0, \frac{1}{d}\mathbf{I}_d)$ respectively. For the ease of the reader, we now recall the following definitions, which are also stated in Section 3. Letting $\bar{\alpha}_l := (\alpha_{p,l})_{p=0}^{\infty}$ denote a sequence of real coefficients, then

$$F(p, k, \bar{\alpha}_l) := \begin{cases} 1 & k = 0 \text{ and } p = 0, \\ 0 & k = 0 \text{ and } p \geq 1, \\ \sum_{(j_i) \in \mathcal{J}(p,k)} \prod_{i=1}^{k} \alpha_{j_i, l} & k \geq 1 \text{ and } p \geq 0, \end{cases} \tag{28}$$

where

$$\mathcal{J}(p, k) := \{(j_i)_{i \in [k]} \ : \ j_i \geq 0 \ \forall i \in [k], \ \sum_{i=1}^{k} j_i = p\}$$

for all $p \in \mathbb{Z}_{\geq 0}$, $k \in \mathbb{Z}_{\geq 1}$.

We are now ready to derive power series for elements of $(\mathbf{G}_l))_{l=1}^{L+1}$ and $(\dot{\mathbf{G}}_l))_{l=2}^{L+1}$.

**Lemma B.2.** *Under Assumptions 1 and 2, for all $l \in [2, L+1]$*

$$\mathbf{G}_l = \sum_{k=0}^{\infty} \alpha_{k,l}(\mathbf{X}\mathbf{X}^T)^{\odot k}, \tag{29}$$

*where the series for each element $[\mathbf{G}_l]_{ij}$ converges absolutely and the coefficients $\alpha_{p,l}$ are nonnegative. The coefficients of the series (29) for all $p \in \mathbb{Z}_{\geq 0}$ can be expressed via the following recurrence relationship,*

$$\alpha_{p,l} = \begin{cases} \sigma_w^2 \mu_p^2(\phi) + \delta_{p=0}\sigma_b^2, & l = 2, \\ \sum_{k=0}^{\infty} \alpha_{k,2} F(p, k, \bar{\alpha}_{l-1}), & l \geq 3. \end{cases} \tag{30}$$

*Furthermore,*

$$\dot{\mathbf{G}}_l = \sum_{k=0}^{\infty} \upsilon_{k,l}(\mathbf{X}\mathbf{X}^T)^{\odot k}, \tag{31}$$

*where likewise the series for each entry $[\dot{\mathbf{G}}_l]_{ij}$ converges absolutely and the coefficients $\upsilon_{p,l}$ for all $p \in \mathbb{Z}_{\geq 0}$ are nonnegative and can be expressed via the following recurrence relationship,*

$$\upsilon_{p,l} = \begin{cases} \sigma_w^2 \mu_p^2(\phi'), & l = 2, \\ \sum_{k=0}^{\infty} \upsilon_{k,2} F(p, k, \bar{\alpha}_{l-1}), & l \geq 3. \end{cases} \tag{32}$$

*Proof.* We start by proving (29) and (30). Proceeding by induction, consider the base case $l = 2$. From Lemma A.1

$$\mathbf{G}_2 = \sigma_w^2 \mathbb{E}_{\mathbf{w} \sim \mathcal{N}(\mathbf{0}, \mathbf{I}_d)}[\phi(\mathbf{X}\mathbf{w})\phi(\mathbf{X}\mathbf{w})^T] + \sigma_b^2 \mathbf{1}_{n \times n}.$$

By the assumptions of the lemma, the conditions of Lemma B.1 are satisfied and therefore

$$\mathbf{G}_2 = \sigma_w^2 \sum_{k=0}^{\infty} \mu_k^2(\phi) \left(\mathbf{X}\mathbf{X}^T\right)^{\odot k} + \sigma_b^2 \mathbf{1}_{n \times n}$$

$$= \alpha_{0,2} \mathbf{1}_{n \times n} + \sum_{k=1}^{\infty} \alpha_{k,2} \left(\mathbf{X}\mathbf{X}^T\right)^{\odot k}.$$

Observe the coefficients $(\alpha_{k,2})_{k \in \mathbb{Z}_{\geq 0}}$ are nonnegative. Therefore, for any $i, j \in [n]$ using Lemma A.2 the series for $[\mathbf{G}_l]_{ij}$ satisfies

$$\sum_{k=0}^{\infty} |\alpha_{k,2}| \left| \langle \mathbf{x}_i, \mathbf{x}_j \rangle^k \right| \leq \sum_{k=0}^{\infty} \alpha_{k,2} \langle \mathbf{x}_i, \mathbf{x}_i \rangle^k = [\mathbf{G}_l]_{ii} = 1 \tag{33}$$

and so must be absolutely convergent. With the base case proved we proceed to assume the inductive hypothesis holds for arbitrary $\mathbf{G}_l$ with $l \in [2, L]$. Observe

$$\mathbf{G}_{l+1} = \sigma_w^2 \mathbb{E}_{\mathbf{w} \sim \mathcal{N}(\mathbf{0}, \mathbf{I}_n)}[\phi(\mathbf{A}\mathbf{w})\phi(\mathbf{A}\mathbf{w})^T] + \sigma_b^2 \mathbf{1}_{n \times n},$$

where $\mathbf{A}$ is a matrix square root of $\mathbf{G}_l$, meaning $\mathbf{G}_l = \mathbf{A}\mathbf{A}$. Recall from Lemma A.1 that $\mathbf{G}_l$ is also symmetric and positive semi-definite, therefore we may additionally assume, without loss of generality, that $\mathbf{A} \in \mathbb{R}^{n \times n}$ is symmetric, which conveniently implies $\mathbf{G}_{n,l} = \mathbf{A}\mathbf{A}^T$. Under the assumptions of the lemma the conditions for Lemma A.2 are satisfied and as a result $[\mathbf{G}_{n,l}]_{ii} = \|\mathbf{a}_i\| = 1$ for all $i \in [n]$, where we recall $\mathbf{a}_i$ denotes the $i$th row of $\mathbf{A}$. Therefore we may again apply Lemma A.1,

$$\mathbf{G}_{l+1} = \sigma_w^2 \sum_{k=0}^{\infty} \mu_k^2(\phi) \left( \mathbf{A}\mathbf{A}^T \right)^{\odot k} + \sigma_b^2 \mathbf{1}_{n \times n}$$

$$= (\sigma_w^2 \mu_0^2(\phi) + \sigma_b^2) \mathbf{1}_{n \times n} + \sigma_w^2 \sum_{k=1}^{\infty} \mu_k^2(\phi) \left( \mathbf{G}_{n,l} \right)^{\odot k}$$

$$= (\sigma_w^2 \mu_0^2(\phi) + \sigma_b^2) \mathbf{1}_{n \times n} + \sigma_w^2 \sum_{k=1}^{\infty} \mu_k^2(\phi) \left( \sum_{m=0}^{\infty} \alpha_{m,l} (\mathbf{X}\mathbf{X}^T)^{\odot m} \right)^{\odot k},$$

where the final equality follows from the inductive hypothesis. For any pair of indices $i, j \in [n]$

$$[\mathbf{G}_{l+1}]_{ij} = (\sigma_w^2 \mu_0^2(\phi) + \sigma_b^2) + \sigma_w^2 \sum_{k=1}^{\infty} \mu_k^2(\phi) \left( \sum_{m=0}^{\infty} \alpha_{m,l} \langle \mathbf{x}_i, \mathbf{x}_j \rangle^m \right)^k.$$

By the induction hypothesis, for any $i, j \in [n]$ the series $\sum_{m=0}^{\infty} \alpha_{m,l} \langle \mathbf{x}_i, \mathbf{x}_j \rangle^m$ is absolutely convergent. Therefore, from the Cauchy product of power series and for any $k \in \mathbb{Z}_{\geq 0}$ we have

$$\left( \sum_{m=0}^{\infty} \alpha_{m,l} \langle \mathbf{x}_i, \mathbf{x}_j \rangle^m \right)^k = \sum_{p=0}^{\infty} F(p, k, \bar{\alpha}_l) \langle \mathbf{x}_i, \mathbf{x}_j \rangle^p, \tag{34}$$

where $F(p, k, \bar{\alpha}_l)$ is defined in (4). By definition, $F(p, k, \bar{\alpha}_l)$ is a sum of products of positive coefficients, and therefore $|F(p, k, \bar{\alpha}_l)| = F(p, k, \bar{\alpha}_l)$. In addition, recall again by Assumption 2 and Lemma A.2 that $[\mathbf{G}_l]_{ii} = 1$. As a result, for any $k \in \mathbb{Z}_{\geq 0}$, as $|\langle \mathbf{x}_i, \mathbf{x}_j \rangle| \leq 1$

$$\sum_{p=0}^{\infty} |F(p, k, \bar{\alpha}_l) \langle \mathbf{x}_i, \mathbf{x}_j \rangle^p| \leq \left( \sum_{m=0}^{\infty} \alpha_{m,l} \right)^k = [\mathbf{G}_{n,l}]_{ii} = 1 \tag{35}$$

and therefore the series $\sum_{p=0}^{\infty} F(p, k, \bar{\alpha}_l) \langle \mathbf{x}_i, \mathbf{x}_j \rangle^p$ converges absolutely. Recalling from the proof of the base case that the series $\sum_{p=1}^{\infty} \alpha_{p,2}$ is absolutely convergent and has only nonnegative elements, we may therefore interchange the order of summation in the following,

$$[\mathbf{G}_{l+1}]_{ij} = (\sigma_w^2 \mu_0^2(\phi) + \sigma_b^2) + \sigma_w^2 \sum_{k=1}^{\infty} \mu_k^2(\phi) \left( \sum_{p=0}^{\infty} F(p, k, \bar{\alpha}_l) \langle \mathbf{x}_i, \mathbf{x}_j \rangle^p \right)$$

$$= \alpha_{0,2} + \sum_{k=1}^{\infty} \alpha_{k,2} \left( \sum_{p=0}^{\infty} F(p, k, \bar{\alpha}_l) \langle \mathbf{x}_i, \mathbf{x}_j \rangle^p \right)$$

$$= \alpha_{0,2} + \sum_{p=0}^{\infty} \left( \sum_{k=1}^{\infty} \alpha_{k,2} F(p, k, \bar{\alpha}_l) \right) \langle \mathbf{x}_i, \mathbf{x}_j \rangle^p.$$

Recalling the definition of $F(p, k, l)$ in (4), in particular $F(0, 0, \bar{\alpha}_l) = 1$ and $F(p, 0, \bar{\alpha}_l) = 0$ for $p \in \mathbb{Z}_{\geq 1}$, then

$$
\begin{aligned}
[\mathbf{G}_{l+1}]_{ij} &= \left( \alpha_{0,2} + \sum_{k=1}^{\infty} \alpha_{k,2} F(0, k, \bar{\alpha}_l) \right) \langle \mathbf{x}_i, \mathbf{x}_j \rangle^0 + \sum_{p=1}^{\infty} \left( \sum_{k=1}^{\infty} \alpha_{k,2} F(p, k, \bar{\alpha}_l) \right) \langle \mathbf{x}_i, \mathbf{x}_j \rangle^p \\
&= \left( \sum_{k=0}^{\infty} \alpha_{k,2} F(0, k, \bar{\alpha}_l) \right) \langle \mathbf{x}_i, \mathbf{x}_j \rangle^0 + \sum_{p=1}^{\infty} \left( \sum_{k=0}^{\infty} \alpha_{k,2} F(p, k, \bar{\alpha}_l) \right) \langle \mathbf{x}_i, \mathbf{x}_j \rangle^p \\
&= \sum_{p=0}^{\infty} \left( \sum_{k=0}^{\infty} \alpha_{k,2} F(p, k, \bar{\alpha}_l) \right) \langle \mathbf{x}_i, \mathbf{x}_j \rangle^p \\
&= \sum_{p=0}^{\infty} \alpha_{p,l+1} \langle \mathbf{x}_i, \mathbf{x}_j \rangle^p.
\end{aligned}
$$

As the indices $i, j \in [n]$ were arbitrary we conclude that

$$
\mathbf{G}_{l+1} = \sum_{p=0}^{\infty} \alpha_{p,l+1} \left( \mathbf{X} \mathbf{X}^T \right)^{\odot p}
$$

as claimed. In addition, by inspection and using the induction hypothesis it is clear that the coefficients $(\alpha_{p,l+1})_{p=0}^{\infty}$ are nonnegative. Therefore, by an argument identical to (33), the series for each entry of $[\mathbf{G}_{l+1}]_{ij}$ is absolutely convergent. This concludes the proof of (29) and (30).

We now turn our attention to proving the (31) and (32). Under the assumptions of the lemma the conditions for Lemmas A.1 and B.1 are satisfied and therefore for the base case $l = 2$

$$
\begin{aligned}
\dot{\mathbf{G}}_2 &= \sigma_w^2 \mathbb{E}_{\mathbf{w} \sim \mathcal{N}(\mathbf{0}, \mathbf{I}_n)} [\phi'(\mathbf{X}\mathbf{w}) \phi'(\mathbf{X}\mathbf{w})^T] \\
&= \sigma_w^2 \sum_{k=0}^{\infty} \mu_k^2(\phi') \left( \mathbf{X}\mathbf{X}^T \right)^{\odot k} \\
&= \sum_{k=0}^{\infty} \upsilon_{k,2} \left( \mathbf{X}\mathbf{X}^T \right)^{\odot k}.
\end{aligned}
$$

By inspection the coefficients $(\upsilon_{p,2})_{p=0}^{\infty}$ are nonnegative and as a result by an argument again identical to (33) the series for each entry of $[\dot{\mathbf{G}}_2]_{ij}$ is absolutely convergent. For $l \in [2, L]$, from (29) and its proof there is a matrix $\mathbf{A} \in \mathbb{R}^{n \times n}$ such that $\mathbf{G}_l = \mathbf{A}\mathbf{A}^T$. Again applying Lemma B.1

$$
\begin{aligned}
\dot{\mathbf{G}}_{n,l+1} &= \sigma_w^2 \mathbb{E}_{\mathbf{w} \sim \mathcal{N}(\mathbf{0}, \mathbf{I}_n)} [\phi'(\mathbf{A}\mathbf{w}) \phi'(\mathbf{A}\mathbf{w})^T] \\
&= \sigma_w^2 \sum_{k=0}^{\infty} \mu_k^2(\phi') \left( \mathbf{A}\mathbf{A}^T \right)^{\odot k} \\
&= \sum_{k=0}^{\infty} \upsilon_{k,2} \left( \mathbf{G}_{n,l} \right)^{\odot k} \\
&= \sum_{k=0}^{\infty} \upsilon_{k,2} \left( \sum_{p=0}^{\infty} \alpha_{p,l} \left( \mathbf{X}\mathbf{X}^T \right)^{\odot p} \right)^{\odot k}
\end{aligned}
$$

Analyzing now an arbitrary entry $[\dot{\mathbf{G}}_{l+1}]_{ij}$, by substituting in the power series expression for $\mathbf{G}_l$ from (29) and using (34) we have

$$
\begin{aligned}
[\dot{\mathbf{G}}_{l+1}]_{ij} &= \sum_{k=0}^{\infty} \upsilon_{k,2} \left( \sum_{p=0}^{\infty} \alpha_{p,l} \langle \mathbf{x}_i, \mathbf{x}_j \rangle^p \right)^k \\
&= \sum_{k=0}^{\infty} \upsilon_{k,2} \left( \sum_{p=0}^{\infty} F(p,k,\bar{\alpha}_l) \langle \mathbf{x}_i, \mathbf{x}_j \rangle^p \right) \\
&= \sum_{p=0}^{\infty} \left( \sum_{k=0}^{\infty} \upsilon_{k,2} F(p,k,\bar{\alpha}_l) \right) \langle \mathbf{x}_i, \mathbf{x}_j \rangle^p \\
&= \sum_{p=0}^{\infty} \upsilon_{p,l+1} \langle \mathbf{x}_i, \mathbf{x}_j \rangle^p.
\end{aligned}
$$

Note that exchanging the order of summation in the third equality above is justified as for any $k \in \mathbb{Z}_{\geq 0}$ by (35) we have $\sum_{p=0}^{\infty} F(p,k,\bar{\alpha}_l)|\langle \mathbf{x}_i, \mathbf{x}_j \rangle|^p \leq 1$ and therefore $\sum_{k=0}^{\infty} \sum_{p=0}^{\infty} \upsilon_{k,2} F(p,k,\bar{\alpha}_l) \langle \mathbf{x}_i, \mathbf{x}_j \rangle^p$ converges absolutely. As the indices $i,j \in [n]$ were arbitrary we conclude that

$$
\dot{\mathbf{G}}_{l+1} = \sum_{p=0}^{\infty} \upsilon_{p,l+1} \left( \mathbf{X}\mathbf{X}^T \right)^{\odot p}
$$

as claimed. Finally, by inspection the coefficients $(\upsilon_{p,l+1})_{p=0}^{\infty}$ are nonnegative, therefore, and again by an argument identical to (33), the series for each entry of $[\dot{\mathbf{G}}_{n,l+1}]_{ij}$ is absolutely convergent. This concludes the proof. $\qquad\square$

We are now prove the key result of Section 3.

**Theorem 3.1.** *Under Assumptions 1 and 2, for all $l \in [L+1]$*

$$
n\mathbf{K}_l = \sum_{p=0}^{\infty} \kappa_{p,l} \left( \mathbf{X}\mathbf{X}^T \right)^{\odot p}. \tag{5}
$$

*The series for each entry $n[\mathbf{K}_l]_{ij}$ converges absolutely and the coefficients $\kappa_{p,l}$ are nonnegative and can be evaluated using the recurrence relationships*

$$
\kappa_{p,l} = \begin{cases} \delta_{p=0}\gamma_b^2 + \delta_{p=1}\gamma_w^2, & l = 1, \\ \alpha_{p,l} + \sum_{q=0}^{p} \kappa_{q,l-1}\upsilon_{p-q,l}, & l \in [2, L+1], \end{cases} \tag{6}
$$

*where*

$$
\alpha_{p,l} = \begin{cases} \sigma_w^2 \mu_p^2(\phi) + \delta_{p=0}\sigma_b^2, & l = 2, \\ \sum_{k=0}^{\infty} \alpha_{k,2} F(p,k,\bar{\alpha}_{l-1}), & l \geq 3, \end{cases} \tag{7}
$$

*and*

$$
\upsilon_{p,l} = \begin{cases} \sigma_w^2 \mu_p^2(\phi'), & l = 2, \\ \sum_{k=0}^{\infty} \upsilon_{k,2} F(p,k,\bar{\alpha}_{l-1}), & l \geq 3, \end{cases} \tag{8}
$$

*are likewise nonnegative for all $p \in \mathbb{Z}_{\geq 0}$ and $l \in [2, L+1]$.*

*Proof.* We proceed by induction. The base case $l = 1$ follows trivially from Lemma A.1. We therefore assume the induction hypothesis holds for an arbitrary $l - 1 \in [1, L]$. From (14) and Lemma B.2

$$
\begin{aligned}
n\mathbf{K}_l &= \mathbf{G}_l + n\mathbf{K}_{l-1} \odot \dot{\mathbf{G}}_l \\
&= \left( \sum_{p=0}^{\infty} \alpha_{p,l} \left( \mathbf{X}\mathbf{X}^T \right)^{\odot p} \right) + \left( n \sum_{q=0}^{\infty} \kappa_{q,l-1} \left( \mathbf{X}\mathbf{X}^T \right)^{\odot q} \right) \odot \left( \sum_{w=0}^{\infty} \upsilon_{w,l} \left( \mathbf{X}\mathbf{X}^T \right)^{\odot w} \right).
\end{aligned}
$$

Therefore, for arbitrary $i, j \in [n]$

$$[n\mathbf{K}_l]_{ij} = \sum_{p=0}^{\infty} \alpha_{p,l} \langle \mathbf{x}_i, \mathbf{x}_j \rangle^p + \left( n \sum_{q=0}^{\infty} \kappa_{q,l-1} \langle \mathbf{x}_i, \mathbf{x}_j \rangle^q \right) \left( \sum_{w=0}^{\infty} \upsilon_{w,l} \langle \mathbf{x}_i, \mathbf{x}_j \rangle^w \right).$$

Observe $n \sum_{q=0}^{\infty} \kappa_{q,l-1} \langle \mathbf{x}_i, \mathbf{x}_j \rangle^q = \Theta^{(l-1)}(\mathbf{x}_i, \mathbf{x}_j)$ and therefore the series must converge due to the convergence of the NTK. Furthermore, $\sum_{w=0}^{\infty} \upsilon_{w,l} \langle \mathbf{x}_i, \mathbf{x}_j \rangle^w = [\dot{\mathbf{G}}_{n,l}]_{ij}$ and therefore is absolutely convergent by Lemma B.2. As a result, by Merten's Theorem the product of these two series is equal to their Cauchy product. Therefore

$$\begin{aligned} [n\mathbf{K}_l]_{ij} &= \sum_{p=0}^{\infty} \alpha_{p,l} \langle \mathbf{x}_i, \mathbf{x}_j \rangle^p + \sum_{p=0}^{\infty} \left( \sum_{q=0}^{p} \kappa_{q,l-1} \upsilon_{p-q,l} \right) \langle \mathbf{x}_i, \mathbf{x}_j \rangle^p \\ &= \sum_{p=0}^{\infty} \left( \alpha_{p,l} + \sum_{q=0}^{p} \kappa_{q,l-1} \upsilon_{p-q,l} \right) \langle \mathbf{x}_i, \mathbf{x}_j \rangle^p \\ &= \sum_{p=0}^{\infty} \kappa_{p,l} \langle \mathbf{x}_i, \mathbf{x}_j \rangle^p, \end{aligned}$$

from which the (5) immediately follows. □

## B.2 ANALYZING THE COEFFICIENTS OF THE NTK POWER SERIES

In this section we study the coefficients of the NTK power series stated in Theorem 3.1. Our first observation is that, under additional assumptions on the activation function $\phi$, the recurrence relationship (6) can be simplified in order to depend only on the Hermite expansion of $\phi$.

**Lemma B.3.** *Under Assumption 3 the Hermite coefficients of $\phi'$ satisfy*

$$\mu_k(\phi') = \sqrt{k+1}\mu_{k+1}(\phi)$$

*for all $k \in \mathbb{Z}_{\geq 0}$.*

*Proof.* Note for each $n \in \mathbb{N}$ as $\phi$ is absolutely continuous on $[-n, n]$ it is differentiable a.e. on $[-n, n]$. It follows by the countable additivity of the Lebesgue measure that $\phi$ is differentiable a.e. on $\mathbb{R}$. Furthermore, as $\phi$ is polynomially bounded we have $\phi \in L^2(\mathbb{R}, e^{-x^2/2}/\sqrt{2\pi})$. Fix $a > 0$. Since $\phi$ is absolutely continuous on $[-a, a]$ it is of bounded variation on $[-a, a]$. Also note that $h_k(x)e^{-x^2/2}$ is of bounded variation on $[-a, a]$ due to having a bounded derivative. Thus we have by Lebesgue-Stieltjes integration-by-parts (see e.g. Folland 1999, Chapter 3)

$$\int_{-a}^{a} \phi'(x) h_k(x) e^{-x^2/2} dx$$

$$= \phi(a) h_k(a) e^{-a^2/2} - \phi(-a) h_k(-a) e^{-a^2/2} + \int_{-a}^{a} \phi(x)[x h_k(x) - h_k'(x)] e^{-x^2/2} dx$$

$$= \phi(a) h_k(a) e^{-a^2/2} - \phi(-a) h_k(-a) e^{-a^2/2} + \int_{-a}^{a} \phi(x) \sqrt{k+1} h_{k+1}(x) e^{-x^2/2} dx,$$

where in the last line above we have used the fact that (24) and (25) imply that $x h_k(x) - h_k'(x) = \sqrt{k+1} h_{k+1}(x)$. Thus we have shown

$$\int_{-a}^{a} \phi'(x) h_k(x) e^{-x^2/2} dx$$

$$= \phi(a) h_k(a) e^{-a^2/2} - \phi(-a) h_k(-a) e^{-a^2/2} + \int_{-a}^{a} \phi(x) \sqrt{k+1} h_{k+1}(x) e^{-x^2/2} dx.$$

We note that since $|\phi(x) h_k(x)| = \mathcal{O}(|x|^{\beta+k})$ we have that as $a \to \infty$ the first two terms above vanish. Thus by sending $a \to \infty$ we have

$$\int_{-\infty}^{\infty} \phi'(x) h_k(x) e^{-x^2/2} dx = \int_{-\infty}^{\infty} \sqrt{k+1} \phi(x) h_{k+1}(x) e^{-x^2/2} dx.$$

After dividing by $\sqrt{2\pi}$ we get the desired result. □

In particular, under Assumption 3, and as highlighted by Corollary B.4, which follows directly from Lemmas B.2 and B.3, the NTK coefficients can be computed only using the Hermite coefficients of $\phi$.

**Corollary B.4.** *Under Assumptions 1, 2 and 3, for all $p \in \mathbb{Z}_{\geq 0}$*

$$v_{p,l} = \begin{cases} (p+1)\alpha_{p+1,2}, & l = 2, \\ \sum_{k=0}^{\infty} v_{k,2} F(p, k, \bar{\alpha}_{l-1}), & l \geq 3. \end{cases} \tag{36}$$

With these results in place we proceed to analyze the decay of the coefficients of the NTK for depth two networks. As stated in the main text, the decay of the NTK coefficients depends on the decay of the Hermite coefficients of the activation function deployed. This in turn is strongly influenced by the behavior of the tails of the activation function. To this end we roughly group activation functions into three categories: growing tails, flat or constant tails and finally decaying tails. Analyzing each of these groups in full generality is beyond the scope of this paper, we therefore instead study the behavior of ReLU, Tanh and Gaussian activation functions, being prototypical and practically used examples of each of these three groups respectively. We remark that these three activation functions satisfy Assumption 3. For typographical ease we let $\omega_\sigma(z) := (1/\sqrt{2\pi\sigma^2}) \exp\left(-z^2/(2\sigma^2)\right)$ denote the Gaussian activation function with variance $\sigma^2$.

**Lemma B.5.** *Under Assumptions 1 and 2,*

1. *if $\phi(z) = ReLU(z)$, then $\kappa_{p,2} = \delta_{(\gamma_b > 0) \cup (p\ even)} \Theta(p^{-3/2})$,*

2. *if $\phi(z) = Tanh(z)$, then $\kappa_{p,2} = \mathcal{O}\left(\exp\left(-\frac{\pi\sqrt{p-1}}{2}\right)\right)$,*

3. *if $\phi(z) = \omega_\sigma(z)$, then $\kappa_{p,2} = \delta_{(\gamma_b > 0) \cup (p\ even)} \Theta(p^{1/2}(\sigma^2 + 1)^{-p})$.*

*Proof.* Recall (9),

$$\kappa_{p,2} = \sigma_w^2 (1 + \gamma_w^2 p) \mu_p^2(\phi) + \sigma_w^2 \gamma_b^2 (1 + p) \mu_{p+1}^2(\phi) + \delta_{p=0} \sigma_b^2.$$

In order to bound $\kappa_{p,2}$ we proceed by using Lemma A.4 to bound the square of the Hermite coefficients. We start with ReLU. Note Lemma A.4 actually provides precise expressions for the Hermite coefficients of ReLU, however, these are not immediately easy to interpret. Observe from Lemma A.4 that above index $p = 2$ all odd indexed Hermite coefficients are 0. It therefore suffices to bound the even indexed terms, given by

$$\mu_p(ReLU) = \frac{1}{\sqrt{2\pi}} \frac{(p-3)!!}{\sqrt{p!}}.$$

Observe from (26) that for $p$ even

$$h_p(0) = (-1)^{p/2} \frac{(p-1)!!}{\sqrt{p!}},$$

therefore

$$\mu_p(ReLU) = \frac{1}{\sqrt{2\pi}} \frac{(p-3)!!}{\sqrt{p!}} = \frac{1}{\sqrt{2\pi}} \frac{|h_p(0)|}{p-1}.$$

Analyzing now $|h_p(0)|$,

$$\frac{(p-1)!!}{\sqrt{p!}} = \frac{\prod_{i=1}^{p/2}(2i-1)}{\sqrt{\prod_{i=1}^{p/2}(2i-1)2i}} = \sqrt{\frac{\prod_{i=1}^{p/2}(2i-1)}{\prod_{i=1}^{p/2} 2i}} = \sqrt{\frac{(p-1)!!}{p!!}}.$$

Here, the expression inside the square root is referred to in the literature as the Wallis ratio, for which the following lower and upper bounds are available Kazarinoff (1956),

$$\sqrt{\frac{1}{\pi(p+0.5)}} < \frac{(p-1)!!}{p!!} < \sqrt{\frac{1}{\pi(p+0.25)}}. \tag{37}$$

As a result

$$|h_p(0)| = \Theta(p^{-1/4})$$

and therefore

$$\mu_p(ReLU) = \begin{cases} \Theta(p^{-5/4}), & p \text{ even}, \\ 0, & p \text{ odd}. \end{cases}$$

As $(p+1)^{-3/2} = \Theta(p^{-3/2})$, then from (9)

$$\begin{aligned}
\kappa_{p,2} &= \Theta((p\mu_p^2(ReLU) + \delta_{\gamma_b>0}(p+1)\mu_{p+1}^2(ReLU))) \\
&= \Theta((\delta_{p\text{ even}}p^{-3/2} + \delta_{(p\text{ odd})\cap(\gamma_b>0)}(p+1)^{-3/2})) \\
&= \Theta\left(\delta_{(p\text{ even})\cup((p\text{ odd})\cap(\gamma_b>0))}p^{-3/2}\right) \\
&= \delta_{(p\text{ even})\cup(\gamma_b>0)}\Theta\left(p^{-3/2}\right)
\end{aligned}$$

as claimed in item *1*.

We now proceed to analyze $\phi(z) = Tanh(z)$. From Panigrahi et al. (2020, Corollary F.7.1)

$$\mu_p(Tanh') = \mathcal{O}\left(\exp\left(-\frac{\pi\sqrt{p}}{4}\right)\right).$$

As Tanh satisfies the conditions of Lemma B.3

$$\mu_p(Tanh) = p^{-1/2}\mu_{p-1}(Tanh') = \mathcal{O}\left(p^{-1/2}\exp\left(-\frac{\pi\sqrt{p-1}}{4}\right)\right).$$

Therefore the result claimed in item *2.* follows as

$$\begin{aligned}
\kappa_{p,2} &= \mathcal{O}((p\mu_p^2(Tanh) + (p+1)\mu_{p+1}^2(Tanh))) \\
&= \mathcal{O}\left(\exp\left(-\frac{\pi\sqrt{p-1}}{2}\right) + \exp\left(-\frac{\pi\sqrt{p}}{2}\right)\right) \\
&= \mathcal{O}\left(\exp\left(-\frac{\pi\sqrt{p-1}}{2}\right)\right).
\end{aligned}$$

Finally, we now consider $\phi(z) = \omega_\sigma(z)$ where $\omega_\sigma(z)$ is the density function of $\mathcal{N}(0,\sigma^2)$. Similar to ReLU, analytic expressions for the Hermite coefficients of $\omega_\sigma(z)$ are known (see e.g., Davis, 2021, Theorem 2.9),

$$\mu_p^2(\omega_\sigma) = \begin{cases} \frac{p!}{((p/2)!)^2 2^p 2\pi(\sigma^2+1)^{p+1}}, & p \text{ even}, \\ 0, & p \text{ odd}. \end{cases}$$

For $p$ even

$$(p/2)! = p!!2^{-p/2}.$$

Therefore

$$\frac{p!}{(p/2)!(p/2)!} = 2^p \frac{p!}{p!!p!!} = 2^p \frac{(p-1)!!}{p!!}.$$

As a result, for $p$ even and using (37), it follows that

$$\mu_p^2(\omega_\sigma) = \frac{(\sigma^2+1)^{-(p+1)}}{2\pi}\frac{(p-1)!!}{p!!} = \Theta(p^{-1/2}(\sigma^2+1)^{-p}).$$

Finally, since $(p+1)^{1/2}(\sigma^2+1)^{-p-1} = \Theta(p^{1/2}(\sigma^2+1)^{-p})$, then from (9)

$$\begin{aligned}
\kappa_{p,2} &= \Theta((p\mu_p^2(\omega_\sigma) + \delta_{\gamma_b>0}(p+1)\mu_{p+1}^2(\omega_\sigma))) \\
&= \Theta\left(\delta_{(p\text{ even})\cup((p\text{ odd})\cap(\gamma_b>0))}p^{1/2}(\sigma^2+1)^{-p}\right) \\
&= \delta_{(p\text{ even})\cup(\gamma_b>0)}\Theta\left(p^{1/2}(\sigma^2+1)^{-p}\right)
\end{aligned}$$

as claimed in item *3*. $\qquad\square$

### B.3 Numerical approximation via a truncated NTK power series and interpretation of Figure 2

Currently, computing the infinite width NTK requires either a) explicit evaluation of the Gaussian integrals highlighted in (13), b) numerical approximation of these same integrals such as in Lee et al. (2018), or c) approximation via a sufficiently wide yet still finite width network, see for instance Engel et al. (2022); Novak et al. (2022). These Gaussian integrals (13) can be solved solved analytically only for a minority of activation functions, notably ReLU as discussed for example by Arora et al. (2019b), while the numerical integration and finite width approximation approaches are relatively computationally expensive. The truncated NTK power series we define as analogous to (5) but with the series involved being computed only up to the $T$th element. Once the top $T$ coefficients are computed, then for any input correlation the NTK can be approximated by evaluating the corresponding finite degree $T$ polynomial.

**Definition B.6.** *For an arbitrary pair* $\mathbf{x}, \mathbf{y} \in \mathbb{S}^{d-1}$ *let* $\rho = \mathbf{x}^T \mathbf{y}$ *denote their linear correlation. Under Assumptions 1, 2 and 3, for all* $l \in [2, L+1]$ *the $T$-truncated NTK power series* $\hat{\Theta}_T^{(l)} : [-1, 1] \rightarrow \mathbb{R}$ *is defined as*

$$\Theta_T^{(l)}(\rho) = \sum_{p=0}^{T} \hat{\kappa}_{p,l} \rho^p. \tag{38}$$

*and whose coefficients are defined via the following recurrence relation,*

$$\hat{\kappa}_{p,l} = \begin{cases} \delta_{p=0} \gamma_b^2 + \delta_{p=1} \gamma_w^2, & l = 1, \\ \hat{\alpha}_{p,l} + \sum_{q=0}^{p} \hat{\kappa}_{q,l-1} \hat{v}_{p-q,l}, & l \in [2, L+1]. \end{cases} \tag{39}$$

*Here, with* $\bar{\hat{\alpha}}_{l-1} = (\hat{\alpha}_{p,l-1})_{p=0}^T$,

$$\hat{\alpha}_{p,l} := \begin{cases} \sigma_w^2 \mu_p^2(\phi) + \delta_{p=0} \sigma_b^2, & l = 2, \\ \sum_{k=0}^{T} \hat{\alpha}_{k,2} F(p, k, \bar{\hat{\alpha}}_{l-1}), & l \geq 3 \end{cases} \tag{40}$$

*and*

$$\hat{v}_{p,l} := \begin{cases} \sqrt{p+1} \hat{\alpha}_{p+1,2}, & l = 2, \\ \sum_{k=0}^{T} \sqrt{k+1} \hat{\alpha}_{p+1,2} F(p, k, \bar{\hat{\alpha}}_l), & l \geq 3. \end{cases} \tag{41}$$

In order to analyze the performance and potential of the truncated NTK for numerical approximation, we compute it for ReLU and compare it with its analytical expression Arora et al. (2019b). To recall this result, let

$$R(\rho) := \frac{\sqrt{1-\rho^2} + \rho \cdot \arcsin(\rho)}{\pi} + \frac{\rho}{2},$$

$$R'(\rho) := \frac{\arcsin(\rho)}{\pi} + \frac{1}{2}.$$

Under Assumptions 1 and 2, with $\phi(z) = ReLU(z)$, $\gamma_w^2 = 1$, $\sigma_w^2 = 2$, $\sigma_b^2 = \gamma_b^2 = 0$, $\mathbf{x}, \mathbf{y} \in \mathbb{S}^d$ and $\rho_1 := \mathbf{x}^T \mathbf{y}$, then $\Theta_1(\mathbf{x}, \mathbf{y}) = \rho$ and for all $l \in [2, L+1]$

$$\begin{aligned} \rho_l &= R(\rho_{l-1}), \\ \Theta_l(\mathbf{x}, \mathbf{y}) &= \rho_l + \rho_{l-1} R'(\rho_{l-1}). \end{aligned} \tag{42}$$

Turning our attention to Figure 2, we observe particularly for input correlations $|\rho| \approx 0.5$ and below then the truncated ReLU NTK power series achieves machine level precision. For $|\rho| \approx 1$ higher order coefficients play a more significant role. As the truncated ReLU NTK power series approximates these coefficients less well the overall approximation of the ReLU NTK is worse. We remark also that negative correlations have a smaller absolute error as odd indexed terms cancel with even index terms: we emphasize again that in Figure 2 we plot the absolute not relative error. In addition, for $L = 1$ there is symmetry in the absolute error for positive and negative correlations as $\alpha_{p,2} = 0$ for all odd $p$. One also observes that approximation accuracy goes down with depth, which is due to the error in the coefficients at the previous layer contributing to the error in the coefficients at the next, thereby resulting in an accumulation of error with depth. Also, and certainly as one might expect, a larger truncation point $T$ results in overall better approximation. Finally, as the decay in the Hermite coefficients for ReLU is relatively slow, see e.g., Table 1 and Lemma 3.2, we expect the truncated ReLU NTK power series to perform worse relative to the truncated NTK's for other activation functions.

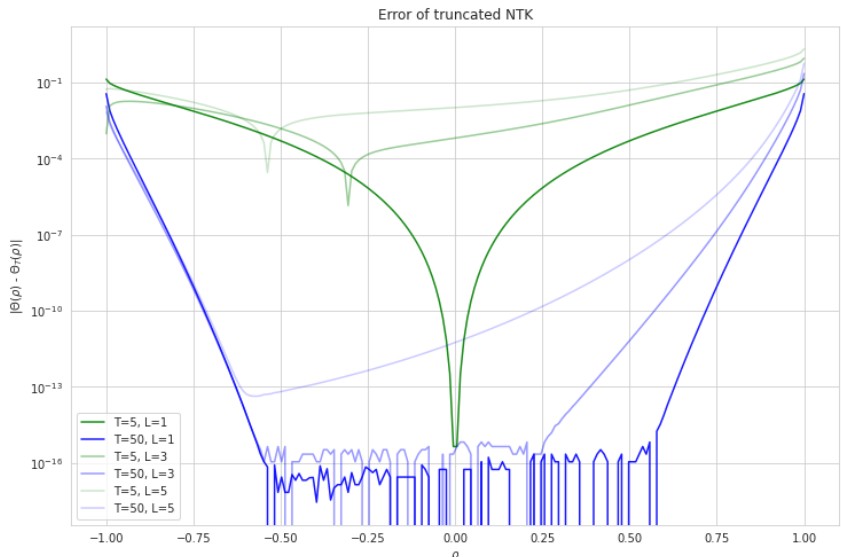

Figure 2: **(NTK Approximation via Truncation)** Absolute error between the analytical ReLU NTK and the truncated ReLU NTK power series as a function of the input correlation $\rho$ for two different values of the truncation point $T$ and three different values for the depth $L$ of the network. Although the truncated NTK achieves a uniform approximation error of only $10^{-1}$ on $[-1, 1]$, for $|\rho| \leq 0.5$, which we remark is more typical for real world data, $T = 50$ suffices for the truncated NTK to achieve machine level precision.

### B.4 CHARACTERIZING NTK POWER SERIES COEFFICIENT DECAY RATES FOR DEEP NETWORKS

In general, Theorem 3.1 does not provide a straightforward path to analyzing the decay of the NTK power series coefficients for depths greater than two. This is at least in part due to the difficulty of analyzing $F(p, k, \bar{\alpha}_{l-1})$, which recall is the sum of all ordered products of $k$ elements of $\bar{\alpha}_{l-1}$ whose indices sum to $p$, defined in (4). However, in the setting where the squares of the Hermite coefficients, and therefore the series $(\alpha_{p,2})_{p=0}^{\infty}$, decay at an exponential rate, this quantity can be characterized and therefore an analysis, at least to a certain degree, of the impact of depth conducted. Although admittedly limited in scope, we highlight that this setting is relevant for the study of Gaussian activation functions and radial basis function (RBF) networks. We will also make the additional simplifying assumption that the activation function has zero Gaussian mean (which can be obtained by centering). Unfortunately this further reduces the applicability of the following results to activation functions commonly used in practice. We leave the study of relaxing this zero bias assumption, perhaps only enforcing exponential decay asymptotically, as well as a proper exploration of other decay patterns, to future work.

The following lemma precisely describes, in the specific setting considered here, the evolution of the coefficients of the Gaussian Process kernel with depth.

**Lemma B.7.** *Let $\alpha_{0,2} = 0$ and $\alpha_{p,2} = C_2 \eta_2^{-p}$ for $p \in \mathbb{Z}_{\geq 1}$, where $C_2$ and $\eta_2$ are constants such that $\sum_{p=1}^{\infty} \alpha_{p,2} = 1$. Then for all $l \geq 2$ and $p \in \mathbb{Z}_{\geq 0}$*

$$\alpha_{p,l+1} = \begin{cases} 0, & p = 0, \\ C_{l+1}\eta_{l+1}^{-p}, & p \geq 1 \end{cases} \tag{43}$$

*where the constants $\eta_{l+1}$ and $C_{l+1}$ are defined as*

$$\eta_{l+1} = \frac{\eta_l \eta_2}{\eta_2 + C_l}, \quad C_{l+1} = \frac{C_l C_2}{\eta_2 + C_l}. \tag{44}$$

*Proof.* Observe for $l = 2$, we have that $\alpha_{0,l} = 0$ and $\alpha_{p,l} = C_l \eta_l^{-p}$ hold by assumption. Thus by induction it suffices to show that $\alpha_{0,l} = 0$ and $\alpha_{p,l} = C_l \eta_l^{-p}$ implies (43) and (44) hold. Thus

assume for some $l \geq 2$ we have that $\alpha_{0,l} = 0$ and $\alpha_{p,l} = C_l \eta_l^{-p}$. Recall the definition of $F$ from (4): as $\alpha_{0,l} = 0$ then with $p \geq 1$ and $1 \leq k \leq p$

$$F(p, k, \bar{\alpha}_l) = \sum_{(j_i) \in \mathcal{J}(p,k)} \prod_{i=1}^{k} \alpha_{j_i, l} = \sum_{(j_i) \in \mathcal{J}_+(p,k)} \prod_{i=1}^{k} \alpha_{j_i, l},$$

where

$$\mathcal{J}_+(p, k) := \left\{ (j_i)_{i \in [k]} \; : \; j_i \geq 1 \; \forall i \in [k], \; \sum_{i=1}^{k} j_i = p \right\} \quad \text{for all } p \in \mathbb{Z}_{\geq 1}, k \in [p],$$

which is the set of all $k$-tuples of *positive* (instead of non-negative) integers which sum to $p$. Substituting $\alpha_{p,l} = C_l \eta_l^{-p}$ then

$$F(p, k, \bar{\alpha}_l) = \sum_{(j_i) \in \mathcal{J}_+(p,k)} C_l^k \eta_l^{-p} = C_l^k \eta_l^{-p} |\mathcal{J}_+(p, k)| = C_l^k \eta_l^{-p} \binom{p-1}{k-1},$$

where the final equality follows from a stars and bars argument. Now observe for $k > p$ that at least one of the indices in $(j_i)_{i=1}^k$ must be 0 and therefore $\prod_{i=1}^k \alpha_{j_i,2} = 0$. As a result under the assumptions of the lemma

$$F(p, k, \bar{\alpha}_l) = \begin{cases} 1, & k = 0 \text{ and } p = 0, \\ C_l^k \eta_l^{-p} \binom{p-1}{k-1}, & k \in [p] \text{ and } p \geq 1, \\ 0, & \text{otherwise.} \end{cases} \tag{45}$$

Substituting (45) into (7) it follows that

$$\alpha_{0,l+1} = \sum_{k=0}^{\infty} \alpha_{k,2} F(0, k, \bar{\alpha}_l) = \alpha_{0,2} = 0$$

and for $p \geq 1$

$$\begin{aligned} \alpha_{p,l+1} &= \sum_{k=0}^{\infty} \alpha_{k,2} F(p, k, \bar{\alpha}_l) \\ &= C_2 \eta_l^{-p} \sum_{k=1}^{p} \left( \frac{C_l}{\eta_2} \right)^k \binom{p-1}{k-1} \\ &= \eta_l^{-p} C_l \eta_2^{-1} C_2 \sum_{h=0}^{p-1} \left( \frac{C_l}{\eta_2} \right)^h \binom{p-1}{h} \\ &= \eta_l^{-p} C_l \eta_2^{-1} C_2 \left( 1 + \frac{C_l}{\eta_2} \right)^{p-1} \\ &= \frac{C_l C_2}{\eta_2 + C_l} \left( \frac{\eta_l \eta_2}{\eta_2 + C_l} \right)^{-p} \\ &= C_{l+1} \eta_{l+1}^{-p} \end{aligned}$$

as claimed. $\qquad \square$

We now analyze the coefficients of the derivative of the Gaussian Process kernel.

**Lemma B.8.** *In addition to the assumptions of Lemma B.7, assume also that $\phi$ satisfies Assumption 3. Then $\upsilon_{p,2} = \frac{C_2}{\eta_2}(1 + p)\eta_2^{-p}$. Furthermore, for all $l \geq 2$ and $p \in \mathbb{Z}_{\geq 0}$*

$$\upsilon_{p,l+1} = \begin{cases} C_2 \eta_2^{-1}, & p = 0, \\ (V_{l+1}' + V_{l+1} p) \eta_{l+1}^{-p}, & p \geq 1, \end{cases} \tag{46}$$

*where the constants $V_{l+1}'$ and $V_{l+1}$ are defined as*

$$V_{l+1}' := \frac{2 C_2 C_l}{\eta_2(C_l + \eta_2)} - \frac{C_2 C_l^2}{\eta_2(C_l + \eta_2)^2}, \quad V_{l+1} := \frac{C_2 C_l^2}{\eta_2(C_l + \eta_2)^2} \tag{47}$$

*and $C_l$ and $\eta_l$ are defined in* (44).

*Proof.* Under Assumption 3 then for all $p \in \mathbb{Z}_{\geq 0}$ we have

$$v_{p,2} = \sigma_w^2 \mu_p^2(\phi') = \sigma_w^2 (p+1)\mu_{p+1}(\phi)^2 = (p+1)\alpha_{p+1,2} = \frac{C_2}{\eta_2}(1+p)\eta_2^{-p}.$$

For $l \geq 2$ and $p = 0$ it therefore follows that

$$v_{0,l+1} = \sum_{k=0}^{\infty}(k+1)\alpha_{k+1,2}F(0,k,\bar{\alpha}_l) = \alpha_{1,2} = C_2\eta_2^{-1}.$$

For $l \geq 2$ and $p \geq 1$ then

$$\begin{aligned}
v_{p,l+1} &= \sum_{k=0}^{\infty} v_{k,2}F(p,k,\bar{\alpha}_l) \\
&= \sum_{k=0}^{\infty}(k+1)\alpha_{k+1,2}F(p,k,\bar{\alpha}_l) \\
&= \sum_{h=1}^{\infty} hC_2\eta_2^{-h}F(p,h-1,\bar{\alpha}_l) \\
&= \frac{C_2}{C_l}\eta_l^{-p}\sum_{h=2}^{p+1} h\left(\frac{C_l}{\eta_2}\right)^h \binom{p-1}{h-2} \\
&= \frac{C_2}{C_l}\eta_l^{-p}\sum_{r=0}^{p-1}(r+2)\left(\frac{C_l}{\eta_2}\right)^{r+2}\binom{p-1}{r} \\
&= \frac{C_2 C_l}{\eta_2^2}\eta_l^{-p}\left(2\sum_{r=0}^{p-1}\left(\frac{C_l}{\eta_2}\right)^r\binom{p-1}{r} + \sum_{r=0}^{p-1} r\left(\frac{C_l}{\eta_2}\right)^r\binom{p-1}{r}\right) \\
&= \frac{C_2 C_l}{\eta_2^2}\eta_l^{-p}\left(2\left(1+\frac{C_l}{\eta_2}\right)^{p-1} + \frac{C_l}{\eta_2}(p-1)\left(1+\frac{C_l}{\eta_2}\right)^{p-2}\right) \\
&= \frac{2C_2 C_l}{\eta_2(C_l+\eta_2)}\left(\frac{\eta_l\eta_2}{\eta_2+C_l}\right)^{-p} + \frac{C_2 C_l^2}{\eta_2(C_l+\eta_2)^2}(p-1)\left(\frac{\eta_l\eta_2}{\eta_2+C_l}\right)^{-p} \\
&= \left(\frac{2C_2 C_l}{\eta_2(C_l+\eta_2)} - \frac{C_2 C_l^2}{\eta_2(C_l+\eta_2)^2}\right)\eta_{l+1}^{-p} + \left(\frac{C_2 C_l^2}{\eta_2(C_l+\eta_2)^2}\right)p\eta_{l+1}^{-p} \\
&= (V'_{l+1} + V_{l+1}p)\eta_{l+1}^{-p}
\end{aligned}$$

as claimed. $\qquad\square$

With the coefficients of both the Gaussian Process kernel and its derivative characterized, we proceed to upper bound the decay of the NTK coefficients in the specific setting outlined in Lemma B.7 and B.8.

**Lemma B.9.** *Let the data, hyperparameters and activation function $\phi$ be such that Assumptions 1, 2 and 3 are satisfied along with the conditions of of Lemma B.7. Then for any $l \geq 2$ there exist positive constants $M'_l$ and $K'_l$ such that for all $p \in \mathbb{Z}_{\geq 1}$*

$$\kappa_{p,l} \leq (M'_l + K'_l p^{2l-3})\eta_l^{-p} \tag{48}$$

*where $\eta_l$ is defined in Lemma B.7.*

*Proof.* We proceed by induction starting with the base case $l = 2$. Applying the results of Lemmas B.7 and B.8 to (6) then for $p \in \mathbb{Z}_{\geq 1}$

$$\kappa_{p,2} = ((C_2 + \gamma_b^2 C_2\eta_2^{-1}) + (\gamma_b^2 C_2\eta_2^{-1} + \gamma_w^2 C_2)p)\eta_2^{-p}. \tag{49}$$

If we define $M'_2 := C_2 + \gamma_b^2 C_2\eta_2^{-1}$ and $K'_2 := \gamma_b^2 C_2\eta_2^{-1} + \gamma_w^2 C_2$, which are clearly positive constants, then $\kappa_{p,2} = (M'_2 + K'_2 p)\eta_2^{-p}$ and so for $l = 2$ the induction hypothesis clearly holds.

We now assume the inductive hypothesis holds for some $l \geq 2$. Observe from (46), with $l \geq 2$ and $p \in \mathbb{Z}_{\geq 0}$ that

$$\upsilon_{p,l+1} \leq (A'_{l+1} + V_{l+1}p)\eta_{l+1}^{-p}. \tag{50}$$

where $A'_{l+1} := \max\{C_2\eta_2^{-1}, V'_{l+1}\}$. Substituting 50 and the inductive hypothesis inequality into (6) it follows for $p \geq 1$ that

$$\kappa_{p,l+1} \leq C_{l+1}\eta_{l+1}^{-p} + \eta_{l+1}^{-p}\sum_{q=0}^{p}(M'_l + K'_l q^{2l-3})\eta_l^{-q}(A'_{l+1} + V_{l+1}(p-q))\eta_{l+1}^q$$

$$= C_{l+1}\eta_{l+1}^{-p} + \eta_{l+1}^{-p}\sum_{q=0}^{p}(M'_l + K'_l q^{2l-3})(A'_{l+1} + V_{l+1}(p-q))\left(\frac{\eta_2}{\eta_2 + C_l}\right)^q$$

$$\leq C_{l+1}\eta_{l+1}^{-p} + \eta_{l+1}^{-p}\sum_{q=0}^{p}(M'_l + K'_l q^{2l-3})(A'_{l+1} + V_{l+1}(p-q))$$

$$\leq C_{l+1}\eta_{l+1}^{-p} + \eta_{l+1}^{-p}\sum_{q=0}^{p}(M'_l + K'_l q^{2l-3})(A'_{l+1} + V_{l+1}p)$$

$$\leq (C_{l+1} + M'_l A'_{l+1})\eta_{l+1}^{-p} + \left(M'_l V_{l+1}p + \sum_{q=1}^{p}(M'_l + K'_l q^{2l-3})(A'_{l+1} + V_{l+1}p)\right)\eta_{l+1}^{-p}$$

$$\leq (C_{l+1} + M'_l A'_{l+1})\eta_{l+1}^{-p} + \left(M'_l V_{l+1}p + p(M'_l + K'_l p^{2l-3})(A'_{l+1} + V_{l+1}p)\right)\eta_{l+1}^{-p}$$

$$\leq (C_{l+1} + M'_l A'_{l+1})\eta_{l+1}^{-p} + p\left(M'_l A'_{l+1} + 2M'_l V_{l+1}p + K'_l A'_{l+1}p^{2l-3} + K'_l V_{l+1}p^{2l-2}\right)\eta_{l+1}^{-p}$$

$$\leq \left((C_{l+1} + M'_l A'_{l+1}) + \left(M'_l A'_{l+1} + 2M'_l V_{l+1} + K'_l A'_{l+1} + K'_l V_{l+1}\right)p^{2l-1}\right)\eta_{l+1}^{-p}$$

Therefore there exist positive constants $M'_{l+1} = C_{l+1} + M'_l A'_{l+1}$ and $K'_{l+1} = M'_l A'_{l+1} + 2M'_l V_{l+1} + K'_l A'_{l+1} + K'_l V_{l+1}$ such that $\kappa_{p,l+1} \leq (M'_{l+1} + K'_{l+1}p^{2(l+1)-3})\eta_{l+1}^{-p}$ as claimed. This completes the inductive step and therefore also the proof of the lemma. $\qquad\square$

## C ANALYZING THE SPECTRUM OF THE NTK VIA ITS POWER SERIES

### C.1 EFFECTIVE RANK OF POWER SERIES KERNELS

Recall that for a positive semidefinite matrix $\mathbf{A}$ we define the *effective rank* Huang et al. (2022) via the following ratio

$$\text{eff}(\mathbf{A}) := \frac{Tr(\mathbf{A})}{\lambda_1(\mathbf{A})}.$$

We consider a kernel Gram matrix $\mathbf{K} \in \mathbb{R}^{n \times n}$ that has the following power series representation in terms of an input gram matrix $\mathbf{X}\mathbf{X}^T$

$$n\mathbf{K} = \sum_{i=0}^{\infty} c_i(\mathbf{X}\mathbf{X}^T)^{\odot i}.$$

Whenever $c_0 \neq 0$ the effective rank of $\mathbf{K}$ is $O(1)$, as displayed in the following theorem.

**Theorem 4.1.** *Assume that we have a kernel Gram matrix $\mathbf{K}$ of the form $n\mathbf{K} = \sum_{p=0}^{\infty} c_p(\mathbf{X}\mathbf{X}^T)^{\odot p}$ where $c_0 \neq 0$. Furthermore, assume the input data $\mathbf{x}_i$ are normalized so that $\|\mathbf{x}_i\| = 1$ for all $i \in [n]$. Then*

$$\text{eff}(\mathbf{K}) \leq \frac{\sum_{p=0}^{\infty} c_p}{c_0}.$$

*Proof.* By linearity of trace we have that

$$Tr(n\mathbf{K}) = \sum_{i=0}^{\infty} c_i Tr((\mathbf{X}\mathbf{X}^T)^{\odot i}) = n\sum_{i=0}^{\infty} c_i$$

where we have used the fact that $Tr((\mathbf{X}\mathbf{X}^T)^{\odot i}) = n$ for all $i \in \mathbb{N}$. On the other hand

$$\lambda_1(n\mathbf{K}) \geq \lambda_1(c_0(\mathbf{X}\mathbf{X}^T)^0) = \lambda_1(c_0 \mathbf{1}_{n \times n}) = nc_0.$$

Thus we have that

$$\text{eff}(\mathbf{K}) = \frac{Tr(\mathbf{K})}{\lambda_1(\mathbf{K})} = \frac{Tr(n\mathbf{K})}{\lambda_1(n\mathbf{K})} \leq \frac{\sum_{i=0}^{\infty} c_i}{c_0}.$$

$\square$

The above theorem demonstrates that the constant term $c_0 \mathbf{1}_{n \times n}$ in the kernel leads to a significant outlier in the spectrum of $\mathbf{K}$. However this fails to capture how the structure of the input data $\mathbf{X}$ manifests in the spectrum of $\mathbf{K}$. For this we will examine the centered kernel matrix $\widetilde{\mathbf{K}} := \mathbf{K} - \frac{c_0}{n}\mathbf{1}\mathbf{1}^T$. Using a very similar argument as before we can demonstrate that the effective rank of $\widetilde{\mathbf{K}}$ is controlled by the effective rank of the input data gram $\mathbf{X}\mathbf{X}^T$. This is formalized in the following theorem.

**Theorem 4.3.** *Assume that we have a kernel Gram matrix $\mathbf{K}$ of the form $n\mathbf{K} = \sum_{p=0}^{\infty} c_p (\mathbf{X}\mathbf{X}^T)^{\odot p}$ where $c_1 \neq 0$. Furthermore, assume the input data $\mathbf{x}_i$ are normalized so that $\|\mathbf{x}_i\| = 1$ for all $i \in [n]$. Then the centered kernel $\widetilde{\mathbf{K}} := \mathbf{K} - \frac{c_0}{n}\mathbf{1}_{n \times n}$ satisfies*

$$\text{eff}(\widetilde{\mathbf{K}}) \leq \text{eff}(\mathbf{X}\mathbf{X}^T)\frac{\sum_{p=1}^{\infty} c_p}{c_1}.$$

*Proof.* By the linearity of the trace we have that

$$Tr(n\widetilde{\mathbf{K}}) = \sum_{i=1}^{\infty} c_i Tr((\mathbf{X}\mathbf{X}^T)^{\odot i}) = Tr(\mathbf{X}\mathbf{X}^T)\sum_{i=1}^{\infty} c_i$$

where we have used the fact that $Tr((\mathbf{X}\mathbf{X}^T)^{\odot i}) = Tr(\mathbf{X}\mathbf{X}^T) = n$ for all $i \in [n]$. On the other hand we have that

$$\lambda_1(n\widetilde{\mathbf{K}}) \geq \lambda_1(c_1\mathbf{X}\mathbf{X}^T) = c_1\lambda_1(\mathbf{X}\mathbf{X}^T).$$

Thus we conclude

$$\text{eff}(\widetilde{\mathbf{K}}) = \frac{Tr(\widetilde{\mathbf{K}})}{\lambda_1(\widetilde{\mathbf{K}})} = \frac{Tr(n\widetilde{\mathbf{K}})}{\lambda_1(n\widetilde{\mathbf{K}})} \leq \frac{Tr(\mathbf{X}\mathbf{X}^T)}{\lambda_1(\mathbf{X}\mathbf{X}^T)}\frac{\sum_{i=1}^{\infty} c_i}{c_1}.$$

$\square$

## C.2 EFFECTIVE RANK OF THE NTK FOR FINITE WIDTH NETWORKS

### C.2.1 NOTATION AND DEFINITIONS

We will let $[k] := \{1, 2, \dots, k\}$. We consider a neural network

$$\sum_{\ell=1}^{m} a_\ell \phi(\langle \mathbf{w}_\ell, \mathbf{x}\rangle)$$

where $\mathbf{x} \in \mathbb{R}^d$ and $\mathbf{w}_\ell \in \mathbb{R}^d$, $a_\ell \in \mathbb{R}$ for all $\ell \in [m]$ and $\phi$ is a scalar valued activation function. The network we present here does not have any bias values in the inner-layer, however the results we will prove later apply to the nonzero bias case by replacing $\mathbf{x}$ with $[\mathbf{x}^T, 1]^T$. We let $\mathbf{W} \in \mathbb{R}^{m \times d}$ be the matrix whose $\ell$-th row is equal to $\mathbf{w}_\ell$ and $\mathbf{a} \in \mathbb{R}^m$ be the vector whose $\ell$-th entry is equal to $a_\ell$. We can then write the neural network in vector form

$$f(\mathbf{x}; \mathbf{W}, \mathbf{a}) = \mathbf{a}^T \phi(\mathbf{W}\mathbf{x})$$

where $\phi$ is understood to be applied entry-wise.

Suppose we have $n$ training data inputs $\mathbf{x}_1, \dots, \mathbf{x}_n \in \mathbb{R}^d$. We will let $\mathbf{X} \in \mathbb{R}^{n \times d}$ be the matrix whose $i$-th row is equal to $\mathbf{x}_i$. Let $\theta_{inner} = vec(\mathbf{W})$ denote the row-wise vectorization of the inner-layer weights. We consider the Jacobian of the neural networks predictions on the training data with respect to the inner layer weights:

$$\mathbf{J}_{inner}^T = \left[\frac{\partial f(\mathbf{x}_1)}{\partial \theta_{inner}}, \frac{\partial f(\mathbf{x}_2)}{\partial \theta_{inner}}, \dots, \frac{\partial f(\mathbf{x}_n)}{\partial \theta_{inner}}\right]$$

Similarly we can look at the analagous quantity for the outer layer weights

$$\mathbf{J}_{outer}^T = \left[\frac{\partial f(\mathbf{x}_1)}{\partial \mathbf{a}}, \frac{\partial f(\mathbf{x}_2)}{\partial \mathbf{a}}, \ldots, \frac{\partial f(\mathbf{x}_n)}{\partial \mathbf{a}}\right] = \phi\left(\mathbf{W}\mathbf{X}^T\right).$$

Our first observation is that the per-example gradients for the inner layer weights have a nice Kronecker product representation

$$\frac{\partial f(\mathbf{x})}{\partial \theta_{inner}} = \begin{bmatrix} a_1\phi'(\langle \mathbf{w}_1, \mathbf{x}\rangle) \\ a_2\phi'(\langle \mathbf{w}_2, \mathbf{x}\rangle) \\ \cdots \\ a_m\phi'(\langle \mathbf{w}_m, \mathbf{x}\rangle) \end{bmatrix} \otimes \mathbf{x}.$$

For convenience we will let

$$\mathbf{Y}_i := \begin{bmatrix} a_1\phi'(\langle \mathbf{w}_1, \mathbf{x}_i\rangle) \\ a_2\phi'(\langle \mathbf{w}_2, \mathbf{x}_i\rangle) \\ \cdots \\ a_m\phi'(\langle \mathbf{w}_m, \mathbf{x}_i\rangle) \end{bmatrix}.$$

where the dependence of $\mathbf{Y}_i$ on the parameters $\mathbf{W}$ and $\mathbf{a}$ is suppressed (formally $\mathbf{Y}_i = \mathbf{Y}_i(\mathbf{W}, \mathbf{a})$). This way we may write

$$\frac{\partial f(\mathbf{x}_i)}{\partial \theta_{inner}} = \mathbf{Y}_i \otimes \mathbf{x}_i.$$

We will study the NTK with respect to the inner-layer weights

$$\mathbf{K}_{inner} = \mathbf{J}_{inner}\mathbf{J}_{inner}^T$$

and the same quantity for the outer-layer weights

$$\mathbf{K}_{outer} = \mathbf{J}_{outer}\mathbf{J}_{outer}^T.$$

For a hermitian matrix $\mathbf{A}$ we will let $\lambda_i(\mathbf{A})$ denote the $i$th largest eigenvalue of $\mathbf{A}$ so that $\lambda_1(\mathbf{A}) \geq \lambda_2(\mathbf{A}) \geq \cdots \geq \lambda_n(\mathbf{A})$. Similarly for an arbitrary matrix $\mathbf{A}$ we will let $\sigma_i(\mathbf{A})$ to the $i$th largest singular value of $\mathbf{A}$. For a matrix $\mathbf{A} \in \mathbb{R}^{r \times k}$ we will let $\sigma_{min}(\mathbf{A}) = \sigma_{\min(r,k)}$.

### C.2.2 EFFECTIVE RANK

For a positive semidefinite matrix $\mathbf{A}$ we define the *effective rank* (Huang et al., 2022) of $\mathbf{A}$ to be the quantity

$$\text{eff}(\mathbf{A}) := \frac{Tr(\mathbf{A})}{\lambda_1(\mathbf{A})}.$$

The effective rank quantifies how many eigenvalues are on the order of the largest eigenvalue. We have the Markov-like inequality

$$|\{i : \lambda_i(\mathbf{A}) \geq c\lambda_1(\mathbf{A})\}| \leq c^{-1}\frac{Tr(\mathbf{A})}{\lambda_1(\mathbf{A})}$$

and the eigenvalue bound

$$\frac{\lambda_i(\mathbf{A})}{\lambda_1(\mathbf{A})} \leq \frac{1}{i}\frac{Tr(\mathbf{A})}{\lambda_1(\mathbf{A})}.$$

Let $\mathbf{A}$ and $\mathbf{B}$ be positive semidefinite matrices. Then we have

$$\frac{Tr(\mathbf{A} + \mathbf{B})}{\lambda_1(\mathbf{A} + \mathbf{B})} \leq \frac{Tr(\mathbf{A}) + Tr(\mathbf{B})}{\max(\lambda_1(\mathbf{A}), \lambda_1(\mathbf{B}))} \leq \frac{Tr(\mathbf{A})}{\lambda_1(\mathbf{A})} + \frac{Tr(\mathbf{B})}{\lambda_1(\mathbf{B})}.$$

Thus the effective rank is subadditive for positive semidefinite matrices.

We will be interested in bounding the effective rank of the NTK. Let $\mathbf{K} = \mathbf{J}\mathbf{J}^T = \mathbf{J}_{outer}\mathbf{J}_{outer}^T + \mathbf{J}_{inner}\mathbf{J}_{inner}^T = \mathbf{K}_{outer} + \mathbf{K}_{inner}$ be the NTK matrix with respect to all the network parameters. Note that by subadditivity

$$\frac{Tr(\mathbf{K})}{\lambda_1(\mathbf{K})} \leq \frac{Tr(\mathbf{K}_{outer})}{\lambda_1(\mathbf{K}_{outer})} + \frac{Tr(\mathbf{K}_{inner})}{\lambda_1(\mathbf{K}_{inner})}.$$

In this vein we will control the effective rank of $\mathbf{K}_{inner}$ and $\mathbf{K}_{outer}$ separately.

### C.2.3 EFFECTIVE RANK OF INNER-LAYER NTK

We will show that the effective rank of inner-layer NTK is bounded by a multiple of the effective rank of the data input gram $\mathbf{X}\mathbf{X}^T$. We introduce the following meta-theorem that we will use to prove various corollaries later

**Theorem C.1.** *Set* $\alpha := \sup_{\|\mathbf{b}\|=1} \left[ \min_{j \in [n]} |\langle \mathbf{Y}_j, \mathbf{b} \rangle| \right]$. *Assume* $\alpha > 0$. *Then*

$$\frac{\min_{i \in [n]} \|\mathbf{Y}_i\|_2^2 \, Tr(\mathbf{X}\mathbf{X}^T)}{\max_{i \in [n]} \|\mathbf{Y}_i\|_2^2 \, \lambda_1(\mathbf{X}\mathbf{X}^T)} \leq \frac{Tr(\mathbf{K}_{inner})}{\lambda_1(\mathbf{K}_{inner})} \leq \frac{\max_{i \in [n]} \|\mathbf{Y}_i\|_2^2 \, Tr(\mathbf{X}\mathbf{X}^T)}{\alpha^2 \, \lambda_1(\mathbf{X}\mathbf{X}^T)}$$

*Proof.* We will first prove the upper bound. We first observe that

$$Tr(\mathbf{K}_{inner}) = \sum_{i=1}^n \left\| \frac{\partial f(\mathbf{x}_i)}{\partial \theta_{inner}} \right\|_2^2 = \sum_{i=1}^n \|\mathbf{Y}_i \otimes \mathbf{x}_i\|_2^2 = \sum_{i=1}^n \|\mathbf{Y}_i\|_2^2 \|\mathbf{x}_i\|_2^2$$

$$\leq \max_{j \in [n]} \|\mathbf{Y}_j\|_2^2 \sum_{i=1}^n \|\mathbf{x}_i\|_2^2 = \max_{j \in [n]} \|\mathbf{Y}_j\|_2^2 \, Tr(\mathbf{X}\mathbf{X}^T)$$

Recall that

$$\lambda_1(\mathbf{K}_{inner}) = \lambda_1(\mathbf{J}_{inner}\mathbf{J}_{inner}^T) = \lambda_1(\mathbf{J}_{inner}^T\mathbf{J}_{inner}).$$

Well

$$\mathbf{J}_{inner}^T \mathbf{J}_{inner} = \sum_{i=1}^n \frac{\partial f(\mathbf{x}_i)}{\partial \theta_{inner}} \frac{\partial f(\mathbf{x}_i)}{\partial \theta_{inner}}^T = \sum_{i=1}^n [\mathbf{Y}_i \otimes \mathbf{x}_i][\mathbf{Y}_i \otimes \mathbf{x}_i]^T$$

$$= \sum_{i=1}^n [\mathbf{Y}_i\mathbf{Y}_i^T] \otimes [\mathbf{x}_i\mathbf{x}_i^T]$$

Well then we may use the fact that

$$\lambda_1(\mathbf{J}_{inner}^T\mathbf{J}_{inner}) = \max_{\|\mathbf{b}\|_2=1} \mathbf{b}^T \mathbf{J}_{inner}^T\mathbf{J}_{inner} \mathbf{b}$$

Let $\mathbf{b}_1 \in \mathbb{R}^m$ and $\mathbf{b}_2 \in \mathbb{R}^d$ be vectors that we will optimize later satisfying $\|\mathbf{b}_1\|_2 \|\mathbf{b}_2\|_2 = 1$. Then we have that $\|\mathbf{b}_1 \otimes \mathbf{b}_2\| = 1$ and

$$(\mathbf{b}_1 \otimes \mathbf{b}_2)^T \mathbf{J}_{inner}^T\mathbf{J}_{inner}(\mathbf{b}_1 \otimes \mathbf{b}_2) = \sum_{i=1}^n (\mathbf{b}_1 \otimes \mathbf{b}_2)^T \left( [\mathbf{Y}_i\mathbf{Y}_i^T] \otimes [\mathbf{x}_i\mathbf{x}_i^T] \right) (\mathbf{b}_1 \otimes \mathbf{b}_2)$$

$$= \sum_{i=1}^n [\mathbf{b}_1^T\mathbf{Y}_i\mathbf{Y}_i^T\mathbf{b}_1][\mathbf{b}_2^T\mathbf{x}_i\mathbf{x}_i^T\mathbf{b}_2] \geq \left[ \min_{j \in [n]} \mathbf{b}_1^T\mathbf{Y}_j\mathbf{Y}_j^T\mathbf{b}_1 \right] \sum_{i=1}^n \mathbf{b}_2^T\mathbf{x}_i\mathbf{x}_i^T\mathbf{b}_2$$

$$= \left[ \min_{j \in [n]} \mathbf{b}_1^T\mathbf{Y}_j\mathbf{Y}_j^T\mathbf{b}_1 \right] \mathbf{b}_2^T \left[ \sum_{i=1}^n \mathbf{x}_i\mathbf{x}_i^T \right] \mathbf{b}_2 = \left[ \min_{j \in [n]} \mathbf{b}_1^T\mathbf{Y}_j\mathbf{Y}_j^T\mathbf{b}_1 \right] \mathbf{b}_2^T\mathbf{X}^T\mathbf{X}\mathbf{b}_2$$

Pick $\mathbf{b}_2$ so that $\|\mathbf{b}_2\| = 1$ and

$$\mathbf{b}_2^T\mathbf{X}^T\mathbf{X}\mathbf{b}_2 = \lambda_1(\mathbf{X}^T\mathbf{X}) = \lambda_1(\mathbf{X}\mathbf{X}^T).$$

Thus for this choice of $\mathbf{b}_2$ we have

$$\lambda_1(\mathbf{J}_{inner}^T\mathbf{J}_{inner}) \geq (\mathbf{b}_1 \otimes \mathbf{b}_2)^T\mathbf{J}_{inner}^T\mathbf{J}_{inner}(\mathbf{b}_1 \otimes \mathbf{b}_2) \geq$$

$$\left[ \min_{j \in [n]} \mathbf{b}_1^T\mathbf{Y}_j\mathbf{Y}_j^T\mathbf{b}_1 \right] \mathbf{b}_2^T\mathbf{X}^T\mathbf{X}\mathbf{b}_2 = \left[ \min_{j \in [n]} \mathbf{b}_1^T\mathbf{Y}_j\mathbf{Y}_j^T\mathbf{b}_1 \right] \lambda_1(\mathbf{X}\mathbf{X}^T)$$

Now note that $\alpha^2 = \sup_{\|\mathbf{b}_1\|=1} \left[ \min_{j \in [n]} \mathbf{b}_1^T\mathbf{Y}_j\mathbf{Y}_j^T\mathbf{b}_1 \right]$. Thus by taking the sup over $\mathbf{b}_1$ in our previous bound we have

$$\lambda_1(\mathbf{K}_{inner}) = \lambda_1(\mathbf{J}_{inner}^T\mathbf{J}_{inner}) \geq \alpha^2 \lambda_1(\mathbf{X}\mathbf{X}^T).$$

Thus combined with our previous result we have

$$\frac{Tr(\mathbf{K}_{inner})}{\lambda_1(\mathbf{K}_{inner})} \leq \frac{\max_{i \in [n]} \|\mathbf{Y}_i\|_2^2}{\alpha^2} \frac{Tr(\mathbf{X}\mathbf{X}^T)}{\lambda_1(\mathbf{X}\mathbf{X}^T)}.$$

We now prove the lower bound.

$$Tr(\mathbf{K}_{inner}) = \sum_{i=1}^n \left\| \frac{\partial f(\mathbf{x}_i)}{\partial \theta_{inner}} \right\|_2^2 = \sum_{i=1}^n \|\mathbf{Y}_i \otimes \mathbf{x}_i\|_2^2 = \sum_{i=1}^n \|\mathbf{Y}_i\|_2^2 \|\mathbf{x}_i\|_2^2$$

$$\geq \min_{j \in [n]} \|\mathbf{Y}_j\|_2^2 \sum_{i=1}^n \|\mathbf{x}_i\|_2^2 = \min_{j \in [n]} \|\mathbf{Y}_j\|_2^2 \, Tr(\mathbf{X}\mathbf{X}^T)$$

Let $\mathbf{Y} \in \mathbb{R}^{n \times m}$ be the matrix whose $i$th row is equal to $\mathbf{Y}_i$. Then observe that

$$\mathbf{K}_{inner} = [\mathbf{Y}\mathbf{Y}^T] \odot [\mathbf{X}\mathbf{X}^T]$$

where $\odot$ denotes the entry-wise Hadamard product of two matrices. We now recall that if $\mathbf{A}$ and $\mathbf{B}$ are two positive semidefinite matrices we have (Oymak & Soltanolkotabi, 2020, Lemma 2)

$$\lambda_1(\mathbf{A} \odot \mathbf{B}) \leq \max_{i \in [n]} \mathbf{A}_{i,i} \lambda_1(\mathbf{B}).$$

Applying this to $\mathbf{K}_{inner}$ we get that

$$\lambda_1(\mathbf{K}_{inner}) \leq \max_{i \in [n]} \|\mathbf{Y}_i\|_2^2 \lambda_1(\mathbf{X}\mathbf{X}^T)$$

Combining this with our previous result we get

$$\frac{\min_{i \in [n]} \|\mathbf{Y}_i\|_2^2 \, Tr(\mathbf{X}\mathbf{X}^T)}{\max_{i \in [n]} \|\mathbf{Y}_i\|_2^2 \, \lambda_1(\mathbf{X}\mathbf{X}^T)} \leq \frac{Tr(\mathbf{K}_{inner})}{\lambda_1(\mathbf{K}_{inner})}$$

$\square$

We can immediately get a useful corollary that applies to the ReLU activation function

**Corollary C.2.** *Set* $\alpha := \sup_{\|\mathbf{b}\|=1} \left[ \min_{j \in [n]} |\langle \mathbf{Y}_j, \mathbf{b} \rangle| \right]$ *and* $\gamma_{max} := \sup_{x \in \mathbb{R}} |\phi'(x)|$. *Assume* $\alpha > 0$ *and* $\gamma_{max} < \infty$. *Then*

$$\frac{\alpha^2}{\gamma_{max}^2 \|\mathbf{a}\|_2^2} \frac{Tr(\mathbf{X}\mathbf{X}^T)}{\lambda_1(\mathbf{X}\mathbf{X}^T)} \leq \frac{Tr(\mathbf{K}_{inner})}{\lambda_1(\mathbf{K}_{inner})} \leq \frac{\gamma_{max}^2 \|\mathbf{a}\|_2^2}{\alpha^2} \frac{Tr(\mathbf{X}\mathbf{X}^T)}{\lambda_1(\mathbf{X}\mathbf{X}^T)}$$

*Proof.* Note that the hypothesis on $|\phi'|$ gives $\|\mathbf{Y}_i\|_2^2 \leq \gamma_{max}^2 \|\mathbf{a}\|_2^2$ for all $i \in [n]$. Moreover by Cauchy-Schwarz we have that $\min_{i \in [n]} \|\mathbf{Y}_i\|_2 \geq \alpha$. Thus by theorem C.1 we get the desired result. $\square$

If $\phi$ is a leaky ReLU type activation (say like those used in Nguyen & Mondelli (2020)) Theorem C.1 translates into an even simpler bound

**Corollary C.3.** *Suppose* $\phi'(x) \in [\gamma_{min}, \gamma_{max}]$ *for all* $x \in \mathbb{R}$ *where* $\gamma_{min} > 0$. *Then*

$$\frac{\gamma_{min}^2 Tr(\mathbf{X}\mathbf{X}^T)}{\gamma_{max}^2 \lambda_1(\mathbf{X}\mathbf{X}^T)} \leq \frac{Tr(\mathbf{K}_{inner})}{\lambda_1(\mathbf{K}_{inner})} \leq \frac{\gamma_{max}^2}{\gamma_{min}^2} \frac{Tr(\mathbf{X}\mathbf{X}^T)}{\lambda_1(\mathbf{X}\mathbf{X}^T)}$$

*Proof.* We will lower bound

$$\alpha := \sup_{\|\mathbf{b}\|=1} \left[ \min_{j \in [n]} |\langle \mathbf{Y}_j, \mathbf{b} \rangle| \right]$$

so that we can apply Corollary C.2. Set $\mathbf{b} = \mathbf{a} / \|\mathbf{a}\|_2$. Then we have that

$$\langle \mathbf{Y}_j, \mathbf{b} \rangle = \sum_{\ell=1}^m a_\ell \phi'(\langle \mathbf{w}_\ell, \mathbf{x}_j \rangle) a_\ell / \|\mathbf{a}\|_2 \geq \frac{\gamma_{min}}{\|\mathbf{a}\|_2} \sum_{\ell=1}^m a_\ell^2 = \gamma_{min} \|\mathbf{a}\|_2$$

Thus $\alpha \geq \gamma_{min} \|\mathbf{a}\|_2$. The result then follows from Corollary C.2 $\square$

To control $\alpha$ in Theorem C.1 when $\phi$ is the ReLU activation function requires a bit more work. To this end we introduce the following lemma.

**Lemma C.4.** *Assume $\phi(x) = ReLU(x)$. Let $R_{min}, R_{max} > 0$ and define $\tau = \{\ell \in [m] : |a_\ell| \in [R_{min}, R_{max}]\}$. Set $T = \min_{i \in [n]} \sum_{\ell \in \tau} \mathbb{I}[\langle \mathbf{x}_i, \mathbf{w}_\ell \rangle \geq 0]$. Then*

$$\alpha := \sup_{\|\mathbf{b}\|=1} \left[ \min_{i \in [n]} |\langle \mathbf{Y}_i, \mathbf{b} \rangle| \right] \geq \frac{R_{min}^2}{R_{max}} \frac{T}{|\tau|^{1/2}}$$

*Proof.* Let $\mathbf{a}_\tau$ be the vector such that $(\mathbf{a}_\tau)_\ell = a_\ell \mathbb{I}[\ell \in \tau]$. Then note that

$$\langle \mathbf{Y}_j, \mathbf{a}_\tau / \|\mathbf{a}_\tau\|_2 \rangle = \frac{1}{\|\mathbf{a}_\tau\|} \sum_{\ell \in \tau} a_\ell^2 \mathbb{I}[\langle \mathbf{w}_\ell, \mathbf{x}_j \rangle \geq 0] \geq$$

$$\frac{R_{min}^2}{\|\mathbf{a}_\tau\|} \sum_{\ell \in \tau} \mathbb{I}[\langle \mathbf{w}_\ell, \mathbf{x}_j \rangle \geq 0] \geq \frac{R_{min}^2}{\|\mathbf{a}_\tau\|_2} T \geq \frac{R_{min}^2}{R_{max} |\tau|^{1/2}} T.$$

$\square$

Roughly what Lemma C.4 says is that $\alpha$ is controlled when there is a set of inner-layer neurons that are active for each data point whose outer layer weights are similar in magnitude. Note that in Du et al. (2019b), Arora et al. (2019a), Oymak et al. (2019), Li et al. (2020), Xie et al. (2017) and Oymak & Soltanolkotabi (2020) the outer layer weights all have fixed constant magnitude. Thus in that case we can set $R_{min} = R_{max}$ in Lemma C.4 so that $\tau = [m]$. In this setting we have the following result.

**Theorem C.5.** *Assume $\phi(x) = ReLU(x)$. Suppose $|a_\ell| = R > 0$ for all $\ell \in [m]$. Furthermore suppose $\mathbf{w}_1, \ldots, \mathbf{w}_m$ are independent random vectors such that $\mathbf{w}_\ell / \|\mathbf{w}_\ell\|$ has the uniform distribution on the sphere for each $\ell \in [m]$. Also assume $m \geq \frac{4 \log(n/\epsilon)}{\delta^2}$ for some $\delta, \epsilon \in (0, 1)$. Then with probability at least $1 - \epsilon$ we have that*

$$\frac{(1-\delta)^2}{4} \mathrm{eff}(\mathbf{X}\mathbf{X}^T) \leq \mathrm{eff}(\mathbf{K}_{inner}) \leq \frac{4}{(1-\delta)^2} \mathrm{eff}(\mathbf{X}\mathbf{X}^T).$$

*Proof.* Fix $j \in [n]$. Note by the assumption on the $\mathbf{w}_\ell$'s we have that $\mathbb{I}[\langle \mathbf{w}_1, \mathbf{x}_j \rangle \geq 0], \ldots, \mathbb{I}[\langle \mathbf{w}_m, \mathbf{x}_j \rangle \geq 0]$ are i.i.d. Bernouilli random variables taking the values 0 and 1 with probability $1/2$. Thus by the Chernoff bound for Binomial random variables we have that

$$\mathbb{P}\left( \sum_{\ell=1}^m \mathbb{I}[\langle \mathbf{w}_\ell, \mathbf{x}_j \rangle \geq 0] \leq \frac{m}{2}(1-\delta) \right) \leq \exp\left( -\delta^2 \frac{m}{4} \right).$$

Thus taking the union bound over every $j \in [n]$ we get that if $m \geq \frac{4 \log(n/\epsilon)}{\delta^2}$ then

$$\min_{j \in [n]} \sum_{\ell=1}^m \mathbb{I}[\langle \mathbf{w}_\ell, \mathbf{x}_j \rangle \geq 0] \geq \frac{m}{2}(1-\delta)$$

holds with probability at least $1 - \epsilon$. Now note that if we set $R_{min} = R_{max} = R$ we have that $\tau = [m]$ where $\tau$ is defined as it is in Lemma C.4. In this case by our previous bound we have that $T$ as defined in Lemma C.4 satisfies $T \geq \frac{m}{2}(1-\delta)$ with probability at least $1 - \epsilon$. In this case the conclusion of Lemma C.4 gives us

$$\alpha \geq R m^{1/2} \frac{(1-\delta)}{2} = \|\mathbf{a}\|_2 \frac{(1-\delta)}{2}.$$

Thus by Corollary C.2 and the above bound for $\alpha$ we get the desired result. $\square$

We will now use Lemma C.4 to prove a bound in the case of Gaussian initialization.

**Lemma C.6.** *Assume $\phi(x) = ReLU(x)$. Suppose that $a_\ell \sim N(0, \nu^2)$ for each $\ell \in [m]$ i.i.d. Furthermore suppose $\mathbf{w}_1, \ldots, \mathbf{w}_m$ are random vectors independent of each other and $\mathbf{a}$ such that $\mathbf{w}_\ell / \|\mathbf{w}_\ell\|$ has the uniform distribution on the sphere for each $\ell \in [m]$. Set $p = \mathbb{P}_{z \sim N(0,1)}(|z| \in [1/2, 1]) \approx 0.3$. Assume*

$$m \geq \frac{4 \log(n/\epsilon)}{\delta^2 (1 - \delta) p}$$

*for some $\epsilon, \delta \in (0, 1)$. Then with probability at least $(1 - \epsilon)^2$ we have that*

$$\alpha := \sup_{\|\mathbf{b}\|=1} \left[ \min_{i \in [n]} |\langle \mathbf{Y}_i, \mathbf{b} \rangle| \right] \geq \frac{\nu}{8} (1 - \delta)^{3/2} p^{1/2} m^{1/2}$$

*Proof.* Set $R_{min} = \nu/2$ and $R_{max} = \nu$. Now set

$$p = \mathbb{P}_{a \sim N(0, \nu^2)}(|a| \in [R_{min}, R_{max}]) = 2\mathbb{P}_{z \sim N(0,1)} \left( z \in \left[ \frac{R_{min}}{\nu}, \frac{R_{max}}{\nu} \right] \right)$$

$$= 2\mathbb{P}_{z \sim N(0,1)}(z \in [1/2, 1]) \approx 0.3.$$

Now define $\tau = \{\ell \in [m] : |a_\ell| \in [R_{min}, R_{max}]\}$. We have by the Chernoff bound for binomial random variables

$$\mathbb{P}(|\tau| \leq (1 - \delta)mp) \leq \exp\left( -\delta^2 \frac{mp}{2} \right).$$

Thus if $m \geq \log\left(\frac{1}{\epsilon}\right) \frac{2}{p\delta^2}$ (a weaker condition than the hypothesis on $m$) then we have that $|\tau| \geq (1 - \delta)mp$ with probability at least $1 - \epsilon$. From now on assume such a $\tau$ has been observed and view it as fixed so that the only remaining randomness is over the $\mathbf{w}_\ell$'s. Now set $T = \min_{i \in [n]} \sum_{\ell \in \tau} \mathbb{I}[\langle \mathbf{x}_i, \mathbf{w}_\ell \rangle \geq 0]$. By the Chernoff bound again we get that for fixed $i \in [n]$

$$\mathbb{P}\left( \sum_{\ell \in \tau} \mathbb{I}[\langle \mathbf{x}_i, \mathbf{w}_\ell \rangle \geq 0] \leq \frac{(1 - \delta)}{2} |\tau| \right) \leq \exp\left( -\delta^2 \frac{|\tau|}{4} \right).$$

Thus by taking the union bound over $i \in [n]$ we get

$$\mathbb{P}\left( T \leq \frac{(1 - \delta)}{2} |\tau| \right) \leq n \exp\left( -\delta^2 \frac{|\tau|}{4} \right)$$

$$\leq n \exp\left( -\delta^2 \frac{(1 - \delta)mp}{4} \right)$$

Thus if we consider $\tau$ as fixed and $m \geq \frac{4 \log(n/\epsilon)}{\delta^2 (1 - \delta) p}$ then with probability at least $1 - \epsilon$ over the sampling of the $\mathbf{w}_\ell$'s we have that

$$T \geq \frac{(1 - \delta)}{2} |\tau|$$

In this case by lemma C.4 we have that

$$\alpha := \sup_{\|\mathbf{b}\|=1} \left[ \min_{i \in [n]} |\langle \mathbf{Y}_i, \mathbf{b} \rangle| \right] \geq \frac{R_{min}^2}{R_{max}} \frac{T}{|\tau|^{1/2}}$$

$$\geq \frac{\nu}{8} (1 - \delta)^{3/2} m^{1/2} p^{1/2}.$$

Thus the above holds with probability at least $(1 - \epsilon)^2$. $\qquad\square$

This lemma now allows us to bound the effective rank of $\mathbf{K}_{inner}$ in the case of Gaussian initialization.

**Theorem C.7.** *Assume $\phi(x) = ReLU(x)$. Suppose that $a_\ell \sim N(0, \nu^2)$ for each $\ell \in [m]$ i.i.d. Furthermore suppose $\mathbf{w}_1, \ldots, \mathbf{w}_m$ are random vectors independent of each other and $\mathbf{a}$ such that $\mathbf{w}_\ell / \|\mathbf{w}_\ell\|$ has the uniform distribution on the sphere for each $\ell \in [m]$. Set $p = \mathbb{P}_{z \sim N(0,1)}(|z| \in [1/2, 1]) \approx 0.3$. Let $\epsilon, \delta \in (0, 1)$. Then there exists absolute constants $c, K > 0$ such that if*

$$m \geq \frac{4 \log(n/\epsilon)}{\delta^2 (1 - \delta) p}$$

*then with probability at least $1 - 3\epsilon$ we have that*

$$\frac{1}{C}\frac{Tr(\mathbf{X}\mathbf{X}^T)}{\lambda_1(\mathbf{X}\mathbf{X}^T)} \leq \frac{Tr(\mathbf{K}_{inner})}{\lambda_1(\mathbf{K}_{inner})} \leq C\frac{Tr(\mathbf{X}\mathbf{X}^T)}{\lambda_1(\mathbf{X}\mathbf{X}^T)}$$

*where*

$$C = \frac{64}{(1-\delta)^3 p}\left[1 + \frac{\max\{c^{-1}K\log(1/\epsilon), mK\}}{m}\right].$$

*Proof.* By Bernstein's inequality

$$\mathbb{P}\left(\|\mathbf{a}/\nu\|_2^2 - m \geq t\right) \leq \exp\left[-c \cdot \min\left(\frac{t^2}{mK^2}, \frac{t}{K}\right)\right]$$

where $c$ is an absolute constant. Set $t = \max\{c^{-1}K\log(1/\epsilon), mK\}$ so that the right hand side of the above inequality is bounded by $\epsilon$. Thus by Lemma C.6 and the union bound we can ensure that with probability at least

$$1 - \epsilon - [1 - (1-\epsilon)^2] = 1 - 3\epsilon + \epsilon^2 \geq 1 - 3\epsilon$$

that $\|\mathbf{a}/\nu\|_2^2 \leq m + t$ and the conclusion of Lemma C.6 hold simultaneously. In that case

$$\frac{\|\mathbf{a}\|_2^2}{\alpha^2} \leq \frac{\nu^2[m+t]}{\frac{\nu^2}{64}(1-\delta)^3 mp} = \frac{64}{(1-\delta)^3 p}\left[1 + \frac{t}{m}\right] = C.$$

Thus by Corollary C.2 we get the desired result. □

By fixing $\delta > 0$ in the previous theorem we get the immediate corollary

**Corollary C.8.** *Assume $\phi(x) = ReLU(x)$. Suppose that $a_\ell \sim N(0, \nu^2)$ for each $\ell \in [m]$ i.i.d. Furthermore suppose $\mathbf{w}_1, \ldots, \mathbf{w}_m$ are random vectors independent of each other and $\mathbf{a}$ such that $\mathbf{w}_\ell / \|\mathbf{w}_\ell\|$ has the uniform distribution on the sphere for each $\ell \in [m]$. Then there exists an absolute constant $C > 0$ such that $m = \Omega(\log(n/\epsilon))$ ensures that with probability at least $1 - \epsilon$*

$$\frac{1}{C}\frac{Tr(\mathbf{X}\mathbf{X}^T)}{\lambda_1(\mathbf{X}\mathbf{X}^T)} \leq \frac{Tr(\mathbf{K}_{inner})}{\lambda_1(\mathbf{K}_{inner})} \leq C\frac{Tr(\mathbf{X}\mathbf{X}^T)}{\lambda_1(\mathbf{X}\mathbf{X}^T)}$$

### C.2.4 EFFECTIVE RANK OF OUTER-LAYER NTK

Throughout this section $\phi(x) = ReLU(x)$. Our goal of this section, similar to before, is to bound the effective rank of $\mathbf{K}_{outer}$ by the effective rank of the input data gram $\mathbf{X}\mathbf{X}^T$. In this section we will use often make use of the basic identities

$$\|\mathbf{A}\mathbf{B}\|_F \leq \|\mathbf{A}\|_2 \|\mathbf{B}\|_F$$

$$\|\mathbf{A}\mathbf{B}\|_F \leq \|\mathbf{A}\|_F \|\mathbf{B}\|_2$$

$$Tr(\mathbf{A}\mathbf{A}^T) = Tr(\mathbf{A}^T\mathbf{A}) = \|\mathbf{A}\|_F^2$$

$$\|\mathbf{A}\|_2 = \|\mathbf{A}^T\|_2$$

$$\lambda_1(\mathbf{A}^T\mathbf{A}) = \lambda_1(\mathbf{A}\mathbf{A}^T) = \|\mathbf{A}\|_2^2.$$

To begin bounding the effective rank of $\mathbf{K}_{outer}$, we prove the following lemma.

**Lemma C.9.** *Assume $\phi(x) = ReLU(x)$ and $\mathbf{W}$ is full rank with $m \geq d$. Then*

$$\frac{\|\phi(\mathbf{W}\mathbf{X}^T)\|_F^2}{[\|\phi(\mathbf{W}\mathbf{X}^T)\|_2 + \|\phi(-\mathbf{W}\mathbf{X}^T)\|_2]^2} \leq \frac{\|\mathbf{W}\|_2^2}{\sigma_{min}(\mathbf{W})^2}\frac{Tr(\mathbf{X}\mathbf{X}^T)}{\lambda_1(\mathbf{X}\mathbf{X}^T)}$$

*Proof.* First note that

$$\left\|\phi(\mathbf{W}\mathbf{X}^T)\right\|_F^2 \le \left\|\mathbf{W}\mathbf{X}^T\right\|_F^2 \le \left\|\mathbf{W}\right\|_2^2 \left\|\mathbf{X}^T\right\|_F^2 = \left\|\mathbf{W}\right\|_2^2 Tr(\mathbf{X}\mathbf{X}^T).$$

Pick $\mathbf{b} \in \mathbb{R}^d$ such that $\|\mathbf{b}\|_2 = 1$ and $\|\mathbf{X}\mathbf{b}\|_2 = \|\mathbf{X}\|_2$. Since $\mathbf{W}^T$ is full rank we may set $\mathbf{u} = (\mathbf{W}^T)^\dagger \mathbf{b}$ so that $\mathbf{W}^T \mathbf{u} = \mathbf{b}$ where $\|\mathbf{u}\|_2 \le \sigma_{min}(\mathbf{W}^T)^{-1}$ where $\sigma_{min}(\mathbf{W}^T)$ is the smallest *nonzero* singular value of $\mathbf{W}^T$. Well then

$$\|\mathbf{X}\|_2 = \|\mathbf{X}\mathbf{b}\|_2 = \left\|\mathbf{X}\mathbf{W}^T \mathbf{u}\right\|_2 \le \left\|\mathbf{X}\mathbf{W}^T\right\|_2 \|\mathbf{u}\|_2 \le \left\|\mathbf{X}\mathbf{W}^T\right\|_2 \sigma_{min}(\mathbf{W}^T)^{-1}$$
$$= \left\|\mathbf{W}\mathbf{X}^T\right\|_2 \sigma_{min}(\mathbf{W})^{-1}$$

Now using the fact that $x = \phi(x) - \phi(-x)$ we have that

$$\left\|\mathbf{W}\mathbf{X}^T\right\|_2 = \left\|\phi(\mathbf{W}\mathbf{X}^T) - \phi(-\mathbf{W}\mathbf{X}^T)\right\|_2 \le \left\|\phi(\mathbf{W}\mathbf{X}^T)\right\|_2 + \left\|\phi(-\mathbf{W}\mathbf{X}^T)\right\|_2$$

Thus combined with our previous results gives

$$\|\mathbf{X}\|_2 \le \sigma_{min}(\mathbf{W})^{-1} \left[\left\|\phi(\mathbf{W}\mathbf{X}^T)\right\|_2 + \left\|\phi(-\mathbf{W}\mathbf{X}^T)\right\|_2\right]$$

Therefore

$$\frac{\left\|\phi(\mathbf{W}\mathbf{X}^T)\right\|_F^2}{\sigma_{min}(\mathbf{W})^{-2} \left[\left\|\phi(\mathbf{W}\mathbf{X}^T)\right\|_2 + \left\|\phi(-\mathbf{W}\mathbf{X}^T)\right\|_2\right]^2} \le \frac{\left\|\phi(\mathbf{W}\mathbf{X}^T)\right\|_F^2}{\|\mathbf{X}\|_2^2}$$
$$\le \frac{\|\mathbf{W}\|_2^2 Tr(\mathbf{X}\mathbf{X}^T)}{\|\mathbf{X}\|_2^2} = \|\mathbf{W}\|_2^2 \frac{Tr(\mathbf{X}\mathbf{X}^T)}{\lambda_1(\mathbf{X}\mathbf{X}^T)}$$

which gives us the desired result. $\qquad\square$

**Corollary C.10.** *Assume $\phi(x) = ReLU(x)$ and $\mathbf{W}$ is full rank with $m \ge d$. Then*

$$\frac{\max\left(\left\|\phi(\mathbf{W}\mathbf{X}^T)\right\|_F^2, \left\|\phi(-\mathbf{W}\mathbf{X}^T)\right\|_F^2\right)}{\max\left(\|\phi(\mathbf{W}\mathbf{X}^T)\|_2^2, \|\phi(-\mathbf{W}\mathbf{X}^T)\|_2^2\right)} \le 4\frac{\|\mathbf{W}\|_2^2}{\sigma_{min}(\mathbf{W})^2} \frac{Tr(\mathbf{X}\mathbf{X}^T)}{\lambda_1(\mathbf{X}\mathbf{X}^T)}.$$

*Proof.* Using the fact that

$$\left\|\phi(\mathbf{W}\mathbf{X}^T)\right\|_2 + \left\|\phi(-\mathbf{W}\mathbf{X}^T)\right\|_2 \le 2\max\left(\left\|\phi(\mathbf{W}\mathbf{X}^T)\right\|_2, \left\|\phi(-\mathbf{W}\mathbf{X}^T)\right\|_2\right)$$

and lemma C.9 we have that

$$\frac{\left\|\phi(\mathbf{W}\mathbf{X}^T)\right\|_F^2}{4\max\left(\|\phi(\mathbf{W}\mathbf{X}^T)\|_2^2, \|\phi(-\mathbf{W}\mathbf{X}^T)\|_2^2\right)} \le \frac{\|\mathbf{W}\|_2^2}{\sigma_{min}(\mathbf{W})^2} \frac{Tr(\mathbf{X}\mathbf{X}^T)}{\lambda_1(\mathbf{X}\mathbf{X}^T)}$$

Note that the right hand side and the denominator of the left hand side do not change when you replace $\mathbf{W}$ with $-\mathbf{W}$. Therefore by using the above bound for both $\mathbf{W}$ and $-\mathbf{W}$ as the weight matrix separately we can conclude

$$\frac{\max\left(\left\|\phi(\mathbf{W}\mathbf{X}^T)\right\|_F^2, \left\|\phi(-\mathbf{W}\mathbf{X}^T)\right\|_F^2\right)}{4\max\left(\|\phi(\mathbf{W}\mathbf{X}^T)\|_2^2, \|\phi(-\mathbf{W}\mathbf{X}^T)\|_2^2\right)} \le \frac{\|\mathbf{W}\|_2^2}{\sigma_{min}(\mathbf{W})^2} \frac{Tr(\mathbf{X}\mathbf{X}^T)}{\lambda_1(\mathbf{X}\mathbf{X}^T)}.$$

$\qquad\square$

**Corollary C.11.** *Assume $\phi(x) = ReLU(x)$ and $m \ge d$. Suppose $\mathbf{W}$ and $-\mathbf{W}$ have the same distribution. Then conditioned on $\mathbf{W}$ being full rank we have that with probability at least $1/2$*

$$\frac{Tr(\mathbf{K}_{outer})}{\lambda_1(\mathbf{K}_{outer})} \le 4\frac{\|\mathbf{W}\|_2^2}{\sigma_{min}(\mathbf{W})^2} \frac{Tr(\mathbf{X}\mathbf{X}^T)}{\lambda_1(\mathbf{X}\mathbf{X}^T)}.$$

*Proof.* Fix $\mathbf{W}$ where $\mathbf{W}$ is full rank. We have by corollary C.10 that either

$$\frac{\left\|\phi(\mathbf{W}\mathbf{X}^T)\right\|_F^2}{\left\|\phi(\mathbf{W}\mathbf{X}^T)\right\|_2^2} \leq 4 \frac{\|\mathbf{W}\|_2^2}{\sigma_{min}(\mathbf{W})^2} \frac{Tr(\mathbf{X}\mathbf{X}^T)}{\lambda_1(\mathbf{X}\mathbf{X}^T)}.$$

holds or

$$\frac{\left\|\phi(-\mathbf{W}\mathbf{X}^T)\right\|_F^2}{\left\|\phi(-\mathbf{W}\mathbf{X}^T)\right\|_2^2} \leq 4 \frac{\|\mathbf{W}\|_2^2}{\sigma_{min}(\mathbf{W})^2} \frac{Tr(\mathbf{X}\mathbf{X}^T)}{\lambda_1(\mathbf{X}\mathbf{X}^T)}$$

(the first holds in the case where $\left\|\phi(\mathbf{W}\mathbf{X}^T)\right\|_2^2 \geq \left\|\phi(-\mathbf{W}\mathbf{X}^T)\right\|_2^2$ and the second in the case $\left\|\phi(\mathbf{W}\mathbf{X}^T)\right\|_2^2 < \left\|\phi(-\mathbf{W}\mathbf{X}^T)\right\|_2^2$). Since $\mathbf{W}$ and $-\mathbf{W}$ have the same distribution, it follows that the first inequality must hold at least $1/2$ of the time. From

$$\frac{Tr(\mathbf{K}_{outer})}{\lambda_1(\mathbf{K}_{outer})} = \frac{\left\|\mathbf{J}_{outer}^T\right\|_F^2}{\left\|\mathbf{J}_{outer}^T\right\|_2^2} = \frac{\left\|\phi(\mathbf{W}\mathbf{X}^T)\right\|_F^2}{\left\|\phi(\mathbf{W}\mathbf{X}^T)\right\|_2^2}$$

we get the desired result. □

We now note that when $\mathbf{W}$ is rectangular shaped and the entries of $\mathbf{W}$ are i.i.d. Gaussians that $\mathbf{W}$ is full rank with high probability and $\sigma_{min}(\mathbf{W})^{-2} \|\mathbf{W}\|_2^2$ is well behaved. We recall the result from Vershynin (2012)

**Theorem C.12.** *Let $\mathbf{A}$ be a $N \times n$ matrix whose entries are independent standard normal random variables. Then for every $t \geq 0$, with probability at least $1 - 2\exp(-t^2/2)$ one has*

$$\sqrt{N} - \sqrt{n} - t \leq \sigma_{min}(\mathbf{A}) \leq \sigma_1(\mathbf{A}) \leq \sqrt{N} + \sqrt{n} + t$$

Corollary C.11 gives us a bound that works at least half the time. However, we would like to derive a bound that holds with high probability. We will have that when $m \gtrsim n$ we have sufficient concentration of the largest singular value of $\phi(\mathbf{W}\mathbf{X}^T)$ to prove such a bound. We recall the result from Vershynin (2012) (Remark 5.40)

**Theorem C.13.** *Assume that $\mathbf{A}$ is an $N \times n$ matrix whose rows $\mathbf{A}_i$ are independent sub-gaussian random vectors in $\mathbb{R}^n$ with second moment matrix $\Sigma$. Then for every $t \geq 0$, the following inequality holds with probability at least $1 - 2\exp(-ct^2)$*

$$\left\|\frac{1}{N}\mathbf{A}^*\mathbf{A} - \Sigma\right\|_2 \leq \max(\delta, \delta^2) \quad where \quad \delta = C\sqrt{\frac{n}{N}} + \frac{t}{\sqrt{N}}$$

*where $C = C_K, c = c_K > 0$ depend only on $K := \max_i \|\mathbf{A}_i\|_{\psi_2}$.*

We will use theorem C.13 in the following lemma.

**Lemma C.14.** *Assume $\phi(x) = ReLU(x)$. Let $\mathbf{A} = \phi(\mathbf{W}\mathbf{X}^T)$ and $M = \max_{i \in [n]} \|\mathbf{x}_i\|_2$. Suppose that $\mathbf{w}_1, \ldots, \mathbf{w}_m \sim N(0, \nu^2 I_d)$ i.i.d. Set $K = M\nu\sqrt{n}$ and define*

$$\Sigma := \mathbb{E}_{\mathbf{w} \sim N(0, \nu^2 I)}[\phi(\mathbf{X}\mathbf{w})\phi(\mathbf{w}^T\mathbf{X}^T)]$$

*Then for every $t \geq 0$ the following inequality holds with probability at least $1 - 2\exp(-c_K t^2)$*

$$\left\|\frac{1}{m}\mathbf{A}^T\mathbf{A} - \Sigma\right\|_2 \leq \max(\delta, \delta^2) \quad where \quad \delta = C_K\sqrt{\frac{n}{m}} + \frac{t}{\sqrt{m}},$$

*where $c_K, C_K > 0$ are absolute constants that depend only on $K$.*

*Proof.* We will let $\mathbf{A}_{\ell:}$ denote the $\ell$th row of $\mathbf{A}$ (considered as a column vector). Note that

$$\mathbf{A}_{\ell:} = \phi(\mathbf{X}\mathbf{w}_\ell).$$

We immediately get that the rows of $\mathbf{A}$ are i.i.d. We will now bound $\|\mathbf{A}_{\ell:}\|_{\psi_2}$. Let $\mathbf{b} \in \mathbb{R}^n$ such that $\|\mathbf{b}\|_2 = 1$. Then

$$
\begin{aligned}
\|\langle \phi(\mathbf{X}\mathbf{w}_\ell), \mathbf{b}\rangle\|_{\psi_2} &= \left\|\sum_{i=1}^n \phi(\langle \mathbf{x}_i, \mathbf{w}_\ell\rangle) b_i\right\|_{\psi_2} \\
&\leq \sum_{i=1}^n |b_i| \, \|\phi(\langle \mathbf{x}_i, \mathbf{w}_\ell\rangle)\|_{\psi_2} \leq \sum_{i=1}^n |b_i| \, \|\langle \mathbf{x}_i, \mathbf{w}_\ell\rangle\|_{\psi_2} \\
&\leq \sum_{i=1}^n |b_i| C \, \|\mathbf{x}_i\|_2 \, \nu \leq CM\nu \, \|\mathbf{b}\|_1 \leq CM\nu\sqrt{n}
\end{aligned}
$$

where $C > 0$ is an absolute constant. Set $K := M\nu\sqrt{n}$. Well then by theorem C.13 we have the following. For every $t \geq 0$ the following inequality holds with probability at least $1 - 2\exp(-c_K t^2)$

$$
\left\|\frac{1}{m}\mathbf{A}^T\mathbf{A} - \Sigma\right\|_2 \leq \max(\delta, \delta^2) \quad \text{where} \quad \delta = C_K\sqrt{\frac{n}{m}} + \frac{t}{\sqrt{m}}
$$

$\square$

We are now ready to prove a high probability bound for the effective rank of $\mathbf{K}_{outer}$.

**Theorem C.15.** *Assume $\phi(x) = ReLU(x)$ and $m \geq d$. Let $M = \max_{i \in [n]} \|\mathbf{x}_i\|_2$. Suppose that $\mathbf{w}_1, \ldots, \mathbf{w}_m \sim N(0, \nu^2 I_d)$ i.i.d. Set $K = M\nu\sqrt{n}$*

$$
\Sigma := \mathbb{E}_{\mathbf{w} \sim N(0, \nu^2 I)}[\phi(\mathbf{X}\mathbf{w})\phi(\mathbf{w}^T\mathbf{X}^T)]
$$

$$
\delta = C_K\left[\sqrt{\frac{n}{m}} + \sqrt{\frac{\log(2/\epsilon)}{m}}\right]
$$

*where $\epsilon > 0$ is small. Now assume*

$$
\sqrt{m} > \sqrt{d} + \sqrt{2\log(2/\epsilon)}
$$

*and*

$$
\max(\delta, \delta^2) \leq \frac{1}{2}\lambda_1(\Sigma)
$$

*Then with probability at least $1 - 3\epsilon$*

$$
\frac{Tr(\mathbf{K}_{outer})}{\lambda_1(\mathbf{K}_{outer})} \leq 12\left(\frac{\sqrt{m} + \sqrt{d} + t_1}{\sqrt{m} - \sqrt{d} - t_1}\right)^2 \frac{Tr(\mathbf{X}^T\mathbf{X})}{\lambda_1(\mathbf{X}^T\mathbf{X})}
$$

*Proof.* By theorem C.12 with $t_1 = \sqrt{2\log(2/\epsilon)}$ we have that with probability at least $1 - \epsilon$ that

$$
\sqrt{m} - \sqrt{d} - t_1 \leq \sigma_{min}(\mathbf{W}/\nu) \leq \sigma_1(\mathbf{W}/\nu) \leq \sqrt{m} + \sqrt{d} + t_1 \tag{51}
$$

The above inequalities and the hypothesis on $m$ imply that $\mathbf{W}$ is full rank.

Let $\mathbf{A} = \phi(\mathbf{W}\mathbf{X}^T)$ and $\tilde{\mathbf{A}} = \phi(-\mathbf{W}\mathbf{X}^T)$. Set $t_2 = \sqrt{\frac{\log(2/\epsilon)}{c_K}}$ where $c_K$ is defined as in theorem C.14. Note that $\mathbf{A}$ and $\tilde{\mathbf{A}}$ are identical in distribution. Thus by theorem C.14 and the union bound we get that with probability at least $1 - 2\epsilon$

$$
\left\|\frac{1}{m}\mathbf{A}^T\mathbf{A} - \Sigma\right\|_2, \left\|\frac{1}{m}\tilde{\mathbf{A}}^T\tilde{\mathbf{A}} - \Sigma\right\|_2 \leq \max(\delta, \delta^2) =: \rho \tag{52}
$$

where

$$
\delta = C_K\sqrt{\frac{n}{m}} + \frac{t_2}{\sqrt{m}}.
$$

By our previous results and the union bound we can ensure with probability at least $1 - 3\epsilon$ that the bounds (51) and (52) all hold simultaneously. In this case we have

$$\left\| \frac{1}{m} \tilde{\mathbf{A}}^T \tilde{\mathbf{A}} \right\|_2 \leq \left\| \frac{1}{m} \mathbf{A}^T \mathbf{A} \right\|_2 + 2\rho$$

$$= \left\| \frac{1}{m} \mathbf{A}^T \mathbf{A} \right\|_2 \left[ 1 + \frac{2\rho}{\left\| \frac{1}{m} \mathbf{A}^T \mathbf{A} \right\|_2} \right] \leq \left\| \frac{1}{m} \mathbf{A}^T \mathbf{A} \right\|_2 \left[ 1 + \frac{2\rho}{\lambda_1(\Sigma) - \rho} \right]$$

Assuming $\rho \leq \lambda_1(\Sigma)/2$ we have by the above bound

$$\left\| \frac{1}{m} \tilde{\mathbf{A}}^T \tilde{\mathbf{A}} \right\|_2 \leq 3 \left\| \frac{1}{m} \mathbf{A}^T \mathbf{A} \right\|_2 .$$

Now note that

$$\left\| \mathbf{A}^T \mathbf{A} \right\|_2 = \left\| \phi(\mathbf{W}\mathbf{X}^T) \right\|_2^2 \quad \left\| \tilde{\mathbf{A}}^T \tilde{\mathbf{A}} \right\|_2 = \left\| \phi(-\mathbf{W}\mathbf{X}^T) \right\|_2^2$$

so that our previous bound implies

$$\left\| \phi(-\mathbf{W}\mathbf{X}^T) \right\|_2^2 \leq 3 \left\| \phi(\mathbf{W}\mathbf{X}^T) \right\|_2^2$$

then we have by corollary C.10 that

$$\frac{Tr(\mathbf{K}_{outer})}{\lambda_1(\mathbf{K}_{outer})} = \frac{\left\| \phi(\mathbf{W}\mathbf{X}^T) \right\|_F^2}{\left\| \phi(\mathbf{W}\mathbf{X}^T) \right\|_2^2} \leq 12 \frac{\left\| \mathbf{W} \right\|_2^2}{\sigma_{min}(\mathbf{W})^2} \frac{Tr(\mathbf{X}\mathbf{X}^T)}{\lambda_1(\mathbf{X}\mathbf{X}^T)}$$

$$\leq 12 \left( \frac{\sqrt{m} + \sqrt{d} + t_1}{\sqrt{m} - \sqrt{d} - t_1} \right)^2 \frac{Tr(\mathbf{X}\mathbf{X}^T)}{\lambda_1(\mathbf{X}\mathbf{X}^T)}.$$

$\square$

From the above theorem we get the following corollary.

**Corollary C.16.** *Assume $\phi(x) = ReLU(x)$ and $n \geq d$. Suppose that $\mathbf{w}_1, \ldots, \mathbf{w}_m \sim N(0, \nu^2 I_d)$ i.i.d. Fix $\epsilon > 0$ small. Set $M = \max_{i \in [n]} \|\mathbf{x}_i\|_2$. Then*

$$m = \Omega \left( \max(\lambda_1(\Sigma)^{-2}, 1) \max(n, \log(1/\epsilon)) \right)$$

*and*

$$\nu = O(1/M\sqrt{m})$$

*suffices to ensure that with probability at least $1 - \epsilon$*

$$\frac{Tr(\mathbf{K}_{outer})}{\lambda_1(\mathbf{K}_{outer})} \leq C \frac{Tr(\mathbf{X}\mathbf{X}^T)}{\lambda_1(\mathbf{X}\mathbf{X}^T)}$$

*where $C > 0$ is an absolute constant.*

### C.2.5 BOUND FOR THE COMBINED NTK

Based on the results in the previous two sections, we can now bound the effective rank of the combined NTK gram matrix $\mathbf{K} = \mathbf{K}_{inner} + \mathbf{K}_{outer}$.

**Theorem 4.5.** *Assume $\phi(x) = ReLU(x)$ and $n \geq d$. Fix $\epsilon > 0$ small. Suppose that $\mathbf{w}_1, \ldots, \mathbf{w}_m \sim N(0, \nu_1^2 I_d)$ i.i.d. and $a_1, \ldots, a_m \sim N(0, \nu_2^2)$. Set $M = \max_{i \in [n]} \|\mathbf{x}_i\|_2$, and let*

$$\Sigma := \mathbb{E}_{\mathbf{w} \sim N(0, \nu_1^2 I)}[\phi(\mathbf{X}\mathbf{w})\phi(\mathbf{w}^T \mathbf{X}^T)].$$

*Then*

$$m = \Omega \left( \max(\lambda_1(\Sigma)^{-2}, 1) \max(n, \log(1/\epsilon)) \right), \quad \nu_1 = O(1/M\sqrt{m})$$

*suffices to ensure that, with probability at least $1 - \epsilon$ over the sampling of the parameter initialization,*

$$\text{eff}(\mathbf{K}) \leq C \cdot \text{eff}(\mathbf{X}\mathbf{X}^T),$$

*where $C > 0$ is an absolute constant.*

*Proof.* This follows from the union bound and Corollaries C.8 and C.16. $\square$

### C.2.6 MAGNITUDE OF THE SPECTRUM

By our results in sections C.2.3 and C.2.4 we have that $m \gtrsim n$ suffices to ensure that

$$\frac{Tr(\mathbf{K})}{\lambda_1(\mathbf{K})} \lesssim \frac{Tr(\mathbf{X}\mathbf{X}^T)}{\lambda_1(\mathbf{X}\mathbf{X}^T)} \leq d$$

Well note that

$$i\frac{\lambda_i(\mathbf{K})}{\lambda_1(\mathbf{K})} \leq \frac{Tr(\mathbf{K})}{\lambda_1(\mathbf{K})} \lesssim d$$

If $i \gg d$ then $\lambda_i(\mathbf{K})/\lambda_1(\mathbf{K})$ is small. Thus the NTK only has $O(d)$ large eigenvalues. The smallest eigenvalue $\lambda_n(\mathbf{K})$ of the NTK has been of interest in proving convergence guarantees (Du et al., 2019a;b; Oymak & Soltanolkotabi, 2020). By our previous inequality

$$\frac{\lambda_n(\mathbf{K})}{\lambda_1(\mathbf{K})} \lesssim \frac{d}{n}$$

Thus in the setting where $m \gtrsim n \gg d$ we have that the smallest eigenvalue will be driven to zero relative to the largest eigenvalue. Alternatively we can view the above inequality as a lower bound on the condition number

$$\frac{\lambda_1(\mathbf{K})}{\lambda_n(\mathbf{K})} \gtrsim \frac{n}{d}$$

We will first bound the analytical NTK in the setting when the outer layer weights have fixed constant magnitude. This is the setting considered by Xie et al. (2017), Arora et al. (2019a), Du et al. (2019b), Oymak et al. (2019), Li et al. (2020), and Oymak & Soltanolkotabi (2020).

**Theorem C.17.** *Let* $\phi(x) = ReLU(x)$ *and assume* $\mathbf{X} \neq 0$. *Let* $\mathbf{K}_{inner}^{\infty} \in \mathbb{R}^{n \times n}$ *be the analytical NTK, i.e.*

$$(\mathbf{K}_{inner}^{\infty})_{i,j} := \langle \mathbf{x}_i, \mathbf{x}_j \rangle \mathbb{E}_{\mathbf{w} \sim N(0, I_d)} \left[ \phi'(\langle \mathbf{x}_i, \mathbf{w} \rangle) \phi'(\langle \mathbf{x}_j, \mathbf{w} \rangle) \right].$$

*Then*

$$\frac{1}{4} \frac{Tr(\mathbf{X}\mathbf{X}^T)}{\lambda_1(\mathbf{X}\mathbf{X}^T)} \leq \frac{Tr(\mathbf{K}_{inner}^{\infty})}{\lambda_1(\mathbf{K}_{inner}^{\infty})} \leq 4 \frac{Tr(\mathbf{X}\mathbf{X}^T)}{\lambda_1(\mathbf{X}\mathbf{X}^T)}.$$

*Proof.* We consider the setting where $|a_\ell| = 1/\sqrt{m}$ for all $\ell \in [m]$ and $\mathbf{w}_\ell \sim N(0, I_d)$ i.i.d.. As was shown by Jacot et al. (2018), Du et al. (2019b) in this setting we have that if we fix the training data $\mathbf{X}$ and send $m \to \infty$ we have that

$$\|\mathbf{K}_{inner} - \mathbf{K}_{inner}^{\infty}\|_2 \to 0$$

in probability. Therefore by continuity of the effective rank we have that

$$\frac{Tr(\mathbf{K}_{inner})}{\lambda_1(\mathbf{K}_{inner})} \to \frac{Tr(\mathbf{K}_{inner}^{\infty})}{\lambda_1(\mathbf{K}_{inner}^{\infty})}$$

in probability. Let $\eta > 0$. Then there exists an $M \in \mathbb{N}$ such that $m \geq M$ implies that

$$\left| \frac{Tr(\mathbf{K}_{inner})}{\lambda_1(\mathbf{K}_{inner})} - \frac{Tr(\mathbf{K}_{inner}^{\infty})}{\lambda_1(\mathbf{K}_{inner}^{\infty})} \right| \leq \eta \tag{53}$$

with probability greater than $1/2$. Now fix $\delta \in (0, 1)$. On the other hand by Theorem C.5 with $\epsilon = 1/4$ we have that if $m \geq \frac{4}{\delta^2} \log(4n)$ then with probability at least $3/4$ that

$$\frac{(1-\delta)^2}{4} \frac{Tr(\mathbf{X}\mathbf{X}^T)}{\lambda_1(\mathbf{X}\mathbf{X}^T)} \leq \frac{Tr(\mathbf{K}_{inner})}{\lambda_1(\mathbf{K}_{inner})} \leq \frac{4}{(1-\delta)^2} \frac{Tr(\mathbf{X}\mathbf{X}^T)}{\lambda_1(\mathbf{X}\mathbf{X}^T)}. \tag{54}$$

Thus if we set $m = \max(\frac{4}{\delta^2} \log(4n), M)$ we have with probability at least $3/4 - 1/2 = 1/4$ that (53) and (54) hold simultaneously. In this case we have that

$$\frac{(1-\delta)^2}{4} \frac{Tr(\mathbf{X}\mathbf{X}^T)}{\lambda_1(\mathbf{X}\mathbf{X}^T)} - \eta \leq \frac{Tr(\mathbf{K}_{inner}^{\infty})}{\lambda_1(\mathbf{K}_{inner}^{\infty})} \leq \frac{4}{(1-\delta)^2} \frac{Tr(\mathbf{X}\mathbf{X}^T)}{\lambda_1(\mathbf{X}\mathbf{X}^T)} + \eta$$

Note that the above argument runs through for any $\eta > 0$ and $\delta \in (0, 1)$. Thus we may send $\eta \to 0^+$ and $\delta \to 0^+$ in the above inequality to get

$$\frac{1}{4} \frac{Tr(\mathbf{X}\mathbf{X}^T)}{\lambda_1(\mathbf{X}\mathbf{X}^T)} \leq \frac{Tr(\mathbf{K}_{inner}^{\infty})}{\lambda_1(\mathbf{K}_{inner}^{\infty})} \leq 4 \frac{Tr(\mathbf{X}\mathbf{X}^T)}{\lambda_1(\mathbf{X}\mathbf{X}^T)}$$

$\square$

We thus have the following corollary about the conditioning of the analytical NTK.

**Corollary C.18.** *Let $\phi(x) = ReLU(x)$ and assume $\mathbf{X} \neq 0$. Let $\mathbf{K}_{inner}^{\infty} \in \mathbb{R}^{n \times n}$ be the analytical NTK, i.e.*

$$(\mathbf{K}_{inner}^{\infty})_{i,j} := \langle \mathbf{x}_i, \mathbf{x}_j \rangle \mathbb{E}_{\mathbf{w} \sim N(0, I_d)} \left[ \phi'(\langle \mathbf{x}_i, \mathbf{w} \rangle) \phi'(\langle \mathbf{x}_j, \mathbf{w} \rangle) \right].$$

*Then*

$$\frac{\lambda_n(\mathbf{K}_{inner}^{\infty})}{\lambda_1(\mathbf{K}_{inner}^{\infty})} \leq 4\frac{d}{n}.$$

## C.3  EXPERIMENTAL VALIDATION OF RESULTS ON THE NTK SPECTRUM

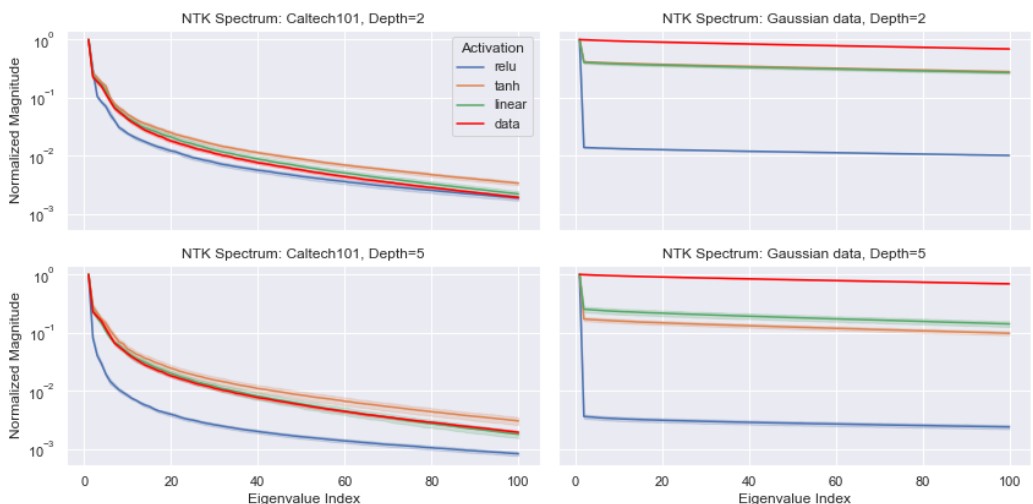

Figure 3: **(NTK Spectrum for CNNs)** We plot the normalized eigenvalues $\lambda_p/\lambda_1$ of the NTK Gram matrix $\mathbf{K}$ and the data Gram matrix $\mathbf{X}\mathbf{X}^T$ for Caltech101 and isotropic Gaussian datasets. To compute the NTK, we randomly initialize convolutional neural networks of depth 2 and 5 with 100 channels per layer. We use the standard parameterization and Pytorch's default Kaiming uniform initialization in order to better connect our results with what is used in practice. We consider a batch size of $n = 200$ and plot the first 100 eigenvalues. The thick part of each curve corresponds to the mean across 10 trials while the transparent part corresponds to the 95% confidence interval.

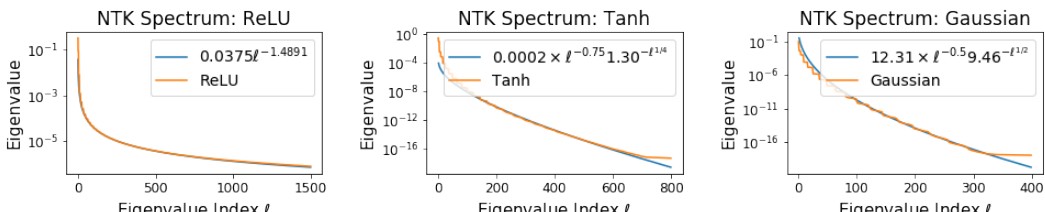

Figure 4: **(Asymptotic NTK Spectrum)** NTK spectrum of two-layer fully connected networks with ReLU, Tanh and Gaussian activations under the NTK parameterization. The orange curve is the experimental eigenvalue. The blue curves in the left shows the regression fit for the experimental eigenvalues as a function of eigenvalue index $\ell$ in the form of $\lambda_\ell = a\ell^{-b}$ where $a$ and $b$ are unknown parameters determined by regression. The blue curves in the middle shows the regression fit for the experimental eigenvalues in the form of $\lambda_\ell = a\ell^{-0.75}b^{-l^{1/4}}$. The blue curves in the right shows the regression fit for the experimental eigenvalues in the form of $\lambda_\ell = a\ell^{-0.5}b^{-l^{1/2}}$.

We experimentally test the theory developed in Section 4.1 and its implications by analyzing the spectrum of the NTK for both fully connected neural network architectures (FCNNs), the results of which are displayed in Figure 1, and also convolutional neural network architectures (CNNs), shown

in Figure 3. For the feedforward architectures we consider networks of depth 2 and 5 with the width of all layers being set at 500. With regard to the activation function we test linear, ReLU and Tanh, and in terms of initialization we use Kaiming uniform (He et al., 2015), which is very common in practice and is the default in PyTorch (Paszke et al., 2019). For the convolutional architectures we again consider depths 2 and 5, with each layer consisting of 100 channels with the filter size set to 5x5. In terms of data, we consider 40x40 patches from both real world images, generated by applying Pytorch's `RandomResizedCrop` transform to a random batch of Caltech101 images (Li et al., 2022), as well as synthetic images corresponding to isotropic Gaussian vectors. The batch sized is fixed at 200 and we plot only the first 100 normalized eigenvalues. Each experiment was repeated 10 times. Finally, to compute the NTK we use the `functorch`[4] module in PyTorch using an algorithmic approach inspired by Novak et al. (2022).

The results for convolutional neural networks show the same trends as observed in feedforward neural networks, which we discussed in Section 4.1. In particular, we again observe the dominant outlier eigenvalue, which increases with both depth and the size of the Gaussian mean of the activation. We also again see that the NTK spectrum inherits its structure from the data, i.e., is skewed for skewed data or relatively flat for isotropic Gaussian data. Finally, we also see that the spectrum for Tanh is closer to the spectrum for the linear activation when compared with the ReLU spectrum. In terms of differences between the CNN and FCNN experiments, we observe that the spread of the 95% confidence interval is slightly larger for convolutional nets, implying a slightly larger variance between trials. We remark that this is likely attributable to the fact that there are only 100 channels in each layer and by increasing this quantity we would expect the variance to reduce. In summary, despite the fact that our analysis is concerned with FCNNs, it appears that the broad implications and trends also hold for CNNs. We leave a thorough study of the NTK spectrum for CNNs and other network architectures to future work.

To test our theory in Section 4.2, we numerically plot the spectrum of NTK of two-layer feedforward networks with ReLU, Tanh, and Gaussian activations in Figure 4. The input data are uniformly drawn from $\mathbb{S}^2$. Notice that when $d = 2$, $k = \Theta(\ell^{1/2})$. Then Corollary 4.7 shows that for the ReLU activation $\lambda_\ell = \Theta(\ell^{-3/2})$, for the Tanh activation $\lambda_\ell = O\left(\ell^{-3/4} \exp(-\frac{\pi}{2}\ell^{1/4})\right)$, and for the Gaussian activation $\lambda_\ell = O(\ell^{-1/2} 2^{-\ell^{1/2}})$. These theoretical decay rates for the NTK spectrum are verified by the experimental results in Figure 4.

## C.4   ANALYSIS OF THE LOWER SPECTRUM: UNIFORM DATA

**Theorem 4.6.** *[Azevedo & Menegatto (2015)] Let $\Gamma$ denote the gamma function. Suppose that the training data are uniformly sampled from the unit hypersphere $\mathbb{S}^d$, $d \geq 2$. If the dot-product kernel function has the expansion $K(x_1, x_2) = \sum_{p=0}^{\infty} c_p \langle x_1, x_2 \rangle^p$ where $c_p \geq 0$, then the eigenvalue of every spherical harmonic of frequency $k$ is given by*

$$\overline{\lambda_k} = \frac{\pi^{d/2}}{2^{k-1}} \sum_{\substack{p \geq k \\ p-k \text{ is even}}} c_p \frac{\Gamma(p+1)\Gamma(\frac{p-k+1}{2})}{\Gamma(p-k+1)\Gamma(\frac{p-k+1}{2}+k+d/2)}.$$

*Proof.* Let $\theta(t) = \sum_{p=0}^{\infty} c_p t^p$, then $K(x_1, x_2) = \theta(\langle x_1, x_2 \rangle)$ According to Funk Hecke theorem (Basri et al., 2019, Section 4.2), we have

$$\overline{\lambda_k} = \text{Vol}(\mathbb{S}^{d-1}) \int_{-1}^{1} \theta(t) P_{k,d}(t)(1-t^2)^{\frac{d-2}{2}} \, dt, \tag{55}$$

where $\text{Vol}(\mathbb{S}^{d-1}) = \frac{2\pi^{d/2}}{\Gamma(d/2)}$ is the volume of the hypersphere $\mathbb{S}^{d-1}$, and $P_{k,d}(t)$ is the Gegenbauer polynomial, given by

$$P_{k,d}(t) = \frac{(-1)^k}{2^k} \frac{\Gamma(d/2)}{\Gamma(k+d/2)} \frac{1}{(1-t^2)^{(d-2)/2}} \frac{d^k}{dt^k} (1-t^2)^{k+(d-2)/2},$$

and $\Gamma$ is the gamma function.

---

[4]https://pytorch.org/functorch/stable/notebooks/neural_tangent_kernels.html

From (55) we have

$$
\begin{aligned}
\overline{\lambda_k} &= \text{Vol}(\mathbb{S}^{d-1}) \int_{-1}^{1} \theta(t) P_{k,d}(t)(1-t^2)^{\frac{d-2}{2}} \mathrm{d}t \\
&= \frac{2\pi^{d/2}}{\Gamma(d/2)} \int_{-1}^{1} \theta(t) \frac{(-1)^k}{2^k} \frac{\Gamma(d/2)}{\Gamma(k+d/2)} \frac{d^k}{dt^k}(1-t^2)^{k+(d-2)/2} \mathrm{d}t \\
&= \frac{2\pi^{d/2}}{\Gamma(d/2)} \frac{(-1)^k}{2^k} \frac{\Gamma(d/2)}{\Gamma(k+d/2)} \sum_{p=0}^{\infty} c_p \int_{-1}^{1} t^p \frac{d^k}{dt^k}(1-t^2)^{k+(d-2)/2} \mathrm{d}t.
\end{aligned}
\tag{56}
$$

Using integration by parts, we have

$$
\begin{aligned}
&\int_{-1}^{1} t^p \frac{d^k}{dt^k}(1-t^2)^{k+(d-2)/2} \mathrm{d}t \\
&= t^p \frac{d^{k-1}}{dt^{k-1}}(1-t^2)^{k+(d-2)/2} \Big|_{-1}^{1} - p \int_{-1}^{1} t^{p-1} \frac{d^{k-1}}{dt^{k-1}}(1-t^2)^{k+(d-2)/2} \mathrm{d}t \\
&= -p \int_{-1}^{1} t^{p-1} \frac{d^{k-1}}{dt^{k-1}}(1-t^2)^{k+(d-2)/2} \mathrm{d}t,
\end{aligned}
\tag{57}
$$

where the last line in (57) holds because $\frac{d^{k-1}}{dt^{k-1}}(1-t^2)^{k+(d-2)/2} = 0$ when $t=1$ or $t=-1$.

When $p < k$, repeat the above procedure (57) $p$ times, we get

$$
\begin{aligned}
\int_{-1}^{1} t^p \frac{d^k}{dt^k}(1-t^2)^{k+(d-2)/2} \mathrm{d}t &= (-1)^p p! \int_{-1}^{1} \frac{d^{k-p}}{dt^{k-p}}(1-t^2)^{k+(d-2)/2} \mathrm{d}t \\
&= (-1)^p p! \frac{d^{k-p-1}}{dt^{k-p-1}}(1-t^2)^{k+(d-2)/2} \Big|_{-1}^{1} \\
&= 0.
\end{aligned}
\tag{58}
$$

When $p \geq k$, repeat the above procedure (57) $k$ times, we get

$$
\int_{-1}^{1} t^p \frac{d^k}{dt^k}(1-t^2)^{k+(d-2)/2} \mathrm{d}t = (-1)^k p(p-1) \cdots (p-k+1) \int_{-1}^{1} t^{p-k}(1-t^2)^{k+(d-2)/2} \mathrm{d}t.
\tag{59}
$$

When $p - k$ is odd, $t^{p-k}(1-t^2)^{k+(d-2)/2}$ is an odd function, then

$$
\int_{-1}^{1} t^{p-k}(1-t^2)^{k+(d-2)/2} \mathrm{d}t = 0.
\tag{60}
$$

When $p - k$ is even,

$$
\begin{aligned}
\int_{-1}^{1} t^{p-k}(1-t^2)^{k+(d-2)/2} \mathrm{d}t &= 2 \int_{0}^{1} t^{p-k}(1-t^2)^{k+(d-2)/2} \mathrm{d}t \\
&= \int_{0}^{1} (t^2)^{(p-k-1)/2}(1-t^2)^{k+(d-2)/2} \mathrm{d}t^2 \\
&= B\left(\frac{p-k+1}{2}, k+d/2\right) \\
&= \frac{\Gamma(\frac{p-k+1}{2})\Gamma(k+d/2)}{\Gamma(\frac{p-k+1}{2}+k+d/2)},
\end{aligned}
\tag{61}
$$

where $B$ is the beta function.

Plugging (61), (58) and (60) into (59), we get

$$
\begin{aligned}
&\int_{-1}^{1} t^p \frac{d^k}{dt^k}(1-t^2)^{k+(d-2)/2} \mathrm{d}t \\
&= \begin{cases} (-1)^k p(p-1) \ldots (p-k+1) \frac{\Gamma(\frac{p-k+1}{2})\Gamma(k+d/2)}{\Gamma(\frac{p-k+1}{2}+k+d/2)}, & p-k \text{ is even and } p \geq k, \\ 0, & \text{otherwise.} \end{cases}
\end{aligned}
\tag{62}
$$

Plugging (62) into (56), we get

$$
\begin{aligned}
\overline{\lambda_k} &= \frac{2\pi^{d/2}}{\Gamma(d/2)} \frac{(-1)^k}{2^k} \frac{\Gamma(d/2)}{\Gamma(k+d/2)} \sum_{\substack{p \geq k \\ p-k \text{ is even}}} c_p (-1)^k p(p-1)\ldots(p-k+1) \frac{\Gamma(\frac{p-k+1}{2})\Gamma(k+d/2)}{\Gamma(\frac{p-k+1}{2}+k+d/2)} \\
&= \frac{\pi^{d/2}}{2^{k-1}} \sum_{\substack{p \geq k \\ p-k \text{ is even}}} c_p \frac{p(p-1)\ldots(p-k+1)\Gamma(\frac{p-k+1}{2})}{\Gamma(\frac{p-k+1}{2}+k+d/2)} \\
&= \frac{\pi^{d/2}}{2^{k-1}} \sum_{\substack{p \geq k \\ p-k \text{ is even}}} c_p \frac{\Gamma(p+1)\Gamma(\frac{p-k+1}{2})}{\Gamma(p-k+1)\Gamma(\frac{p-k+1}{2}+k+d/2)}.
\end{aligned}
$$

$\square$

**Corollary C.19.** *Under the same setting as in Theorem 4.6,*

1. *if $c_p = \Theta(p^{-a})$ where $a \geq 1$, then $\overline{\lambda_k} = \Theta(k^{-d-2a+2})$,*

2. *if $c_p = \delta_{(p \text{ even})}\Theta(p^{-a})$, then $\overline{\lambda_k} = \delta_{(k \text{ even})}\Theta(k^{-d-2a+2})$,*

3. *if $c_p = \mathcal{O}\left(\exp\left(-a\sqrt{p}\right)\right)$, then $\overline{\lambda_k} = \mathcal{O}\left(k^{-d+1/2}\exp\left(-a\sqrt{k}\right)\right)$,*

4. *if $c_p = \Theta(p^{1/2}a^{-p})$, then $\overline{\lambda_k} = \mathcal{O}\left(k^{-d+1}a^{-k}\right)$ and $\overline{\lambda_k} = \Omega\left(k^{-d/2+1}2^{-k}a^{-k}\right)$.*

*Proof of Corollary C.4, part 1.* We first prove $\overline{\lambda_k} = O(k^{-d-2a+2})$. Suppose that $c_p \leq Cp^{-a}$ for some constant $C$, then according to Theorem 4.6 we have

$$
\overline{\lambda_k} \leq \frac{\pi^{d/2}}{2^{k-1}} \sum_{\substack{p \geq k \\ p-k \text{ is even}}} Cp^{-a} \frac{\Gamma(p+1)\Gamma(\frac{p-k+1}{2})}{\Gamma(p-k+1)\Gamma(\frac{p-k+1}{2}+k+d/2)}.
$$

According to Stirling's formula, we have

$$
\Gamma(z) = \sqrt{\frac{2\pi}{z}}\left(\frac{z}{e}\right)^z \left(1+O\left(\frac{1}{z}\right)\right). \tag{63}
$$

Then for any $z \geq \frac{1}{2}$, we can find constants $C_1$ and $C_2$ such that

$$
C_1\sqrt{\frac{2\pi}{z}}\left(\frac{z}{e}\right)^z \leq \Gamma(z) \leq C_2\sqrt{\frac{2\pi}{z}}\left(\frac{z}{e}\right)^z. \tag{64}
$$

Then

$$
\begin{aligned}
\overline{\lambda_k} &\leq \frac{\pi^{d/2}}{2^{k-1}} \frac{C_2^2}{C_1^2} \sum_{\substack{p \geq k \\ p-k \text{ is even}}} Cp^{-a} \frac{\sqrt{\frac{2\pi}{p+1}}\left(\frac{p+1}{e}\right)^{p+1}\sqrt{\frac{2\pi}{\frac{p-k+1}{2}}}\left(\frac{\frac{p-k+1}{2}}{e}\right)^{\frac{p-k+1}{2}}}{\sqrt{\frac{2\pi}{p-k+1}}\left(\frac{p-k+1}{e}\right)^{p-k+1}\sqrt{\frac{2\pi}{\frac{p-k+1}{2}+k+d/2}}\left(\frac{\frac{p-k+1}{2}+k+d/2}{e}\right)^{\frac{p-k+1}{2}+k+d/2}} \\
&= \frac{\pi^{d/2}}{2^{k-1}} \frac{C_2^2 C}{C_1^2} \sum_{\substack{p \geq k \\ p-k \text{ is even}}} p^{-a} \frac{e^{\frac{d}{2}}\sqrt{\frac{2}{p+1}}(p+1)^{p+1}\left(\frac{p-k+1}{2}\right)^{\frac{p-k+1}{2}}}{(p-k+1)^{p-k+1}\sqrt{\frac{1}{\frac{p-k+1}{2}+k+d/2}}\left(\frac{p-k+1}{2}+k+d/2\right)^{\frac{p-k+1}{2}+k+d/2}} \\
&= \frac{\pi^{d/2}}{2^{k-1}} \frac{C_2^2 C}{C_1^2} \sum_{\substack{p \geq k \\ p-k \text{ is even}}} p^{-a} \frac{e^{\frac{d}{2}}2^{\frac{-p+k}{2}}(p+1)^{p+\frac{1}{2}}}{(p-k+1)^{\frac{p-k+1}{2}}\left(\frac{p-k+1}{2}+k+d/2\right)^{\frac{p-k}{2}+k+d/2}} \\
&= 2\pi^{d/2} \frac{2^{\frac{d}{2}}e^{\frac{d}{2}}C_2^2 C}{C_1^2} \sum_{\substack{p \geq k \\ p-k \text{ is even}}} \frac{p^{-a}(p+1)^{p+\frac{1}{2}}}{(p-k+1)^{\frac{p-k+1}{2}}(p+k+1+d)^{\frac{p+k+d}{2}}}. \tag{65}
\end{aligned}
$$

We define

$$f_a(p) = \frac{p^{-a}\,(p+1)^{p+\frac{1}{2}}}{(p-k+1)^{\frac{p-k+1}{2}}\,(p+k+1+d)^{\frac{p+k+d}{2}}}. \tag{66}$$

By applying the chain rule to $e^{\log f_a(p)}$, we have that the derivative of $f_a$ is

$$f_a'(p) = \frac{(p+1)^{p+\frac{1}{2}}p^{-a}}{2(p-k+1)^{\frac{p-k+1}{2}}(p+k+d+1)^{\frac{p+k+d}{2}}}$$
$$\cdot \left( -\frac{2a}{p} - \frac{k+d}{(p+1)(p+k+d+1)} + \log(1 + \frac{k^2 - d(p-k+1)}{(p-k+1)(p+k+d+1)}) \right). \tag{67}$$

Let $g_a(p) = -\frac{2a}{p} - \frac{k+d}{(p+1)(p+k+d+1)} + \log(1 + \frac{k^2-d(p-k+1)}{(p-k+1)(p+k+d+1)})$. Then $g_a(p)$ and $f_a'(p)$ have the same sign. Next we will show that $g_a(p) \geq 0$ for $k \leq p \leq \frac{k^2}{d+24a}$ when $k$ is large enough.

First when $p \geq k$ and $\frac{k^2-d(p-k+1)}{(p-k+1)(p+k+d+1)} \geq 1$, we have

$$g_a(p) \geq -\frac{2a}{k} - \frac{k+d}{(k+1)(k+k+d+1)} + \log(2) \geq 0, \tag{68}$$

when $k$ is sufficiently large.

Second when $p \geq k$ and $0 \leq \frac{k^2-d(p-k+1)}{(p-k+1)(p+k+d+1)} \leq 1$, since $\log(1+x) \geq \frac{x}{2}$ for $0 \leq x \leq 1$, we have

$$g_a(p) \geq -\frac{2a}{p} - \frac{k+d}{(p+1)(p+k+d+1)} + \frac{k^2-d(p-k+1)}{2(p-k+1)(p+k+d+1)}$$
$$\geq -\frac{2a}{p} - \frac{k+d}{(p+1)(p+k+d+1)} + \frac{k^2-dp}{2p(p+k+d+1)}.$$

When $p \leq \frac{k^2}{d+24a}$, we have $k^2 - dp \geq 24ap$. Then

$$\frac{k^2-dp}{4p(p+k+d+1)} \geq \frac{24ap}{4p(p+k+d+1)} \geq \frac{6ap}{(p+1)(p+k+d+1)} \geq \frac{k+d}{(p+1)(p+k+d+1)}$$

when $k$ is sufficiently large. Also we have

$$\frac{k^2-dp}{4r(p+k+d+1)} \geq \frac{24ap}{4r(p+k+d+1)} \geq \frac{6a}{p+k+d+1} \geq \frac{2a}{p}$$

when $k$ is sufficiently large.

Combining all the arguments above, we conclude that $g_a(p) \geq 0$ and $f_a'(p) \geq 0$ when $k \leq p \leq \frac{k^2}{d+24a}$. Then when $k \leq p \leq \frac{k^2}{d+24a}$, we have

$$f_a(p) \leq f_a\left(\frac{k^2}{d+24a}\right). \tag{69}$$

When $p \geq \frac{k^2}{d+24a}$, we have

$$f_a(p) = \frac{p^{-a}\,(p+1)^{p+\frac{1}{2}}}{(p-k+1)^{\frac{p-k+1}{2}}\,(p+k+1+d)^{\frac{p+k+d}{2}}}$$
$$= \frac{p^{-a}\,(p+1)^{p+\frac{1}{2}}}{((p+1)^2 - k^2 + d(p-k+1))^{\frac{p-k+1}{2}}\,(p+k+1+d)^{\frac{2k+d-1}{2}}}$$
$$= \frac{p^{-a}\,(p+1)^{-\frac{d}{2}}}{\left(1 - \frac{k^2-d(p-k+1)}{(p+1)^2}\right)^{\frac{p-k+1}{2}} \left(1 + \frac{k+d}{p+1}\right)^{\frac{2k+d-1}{2}}}$$
$$\leq \frac{p^{-a-\frac{d}{2}}}{\left(1 - \frac{k^2-d(p-k+1)}{(p+1)^2}\right)^{\frac{p-k+1}{2}}}.$$

If $k^2 - d(p-k+1) < 0$, $\left(1 - \frac{k^2 - d(p-k+1)}{(p+1)^2}\right)^{\frac{p-k+1}{2}} \geq 1$. If $k^2 - d(p-k+1) \geq 0$, i.e., $p \leq \frac{k^2+dk-d}{d}$, for sufficiently large $k$, we have

$$
\left(1 - \frac{k^2 - d(p-k+1)}{(p+1)^2}\right)^{\frac{p-k+1}{2}} \geq \left(1 - \frac{k^2 - d(\frac{k^2}{d+24a} - k + 1)}{(\frac{k^2}{d+24a} + 1)^2}\right)^{\frac{\frac{k^2+dk-d}{d} - k + 1}{2}}
$$

$$
\geq \left(1 - \frac{48a(d+24a)}{k^2}\right)^{\frac{k^2}{2d}}
$$

$$
\geq e^{-\frac{k^2}{2d}\frac{48a(d+24a)}{k^2}} = e^{-\frac{48a(d+24a)}{2d}},
$$

which is a constant independent of $k$. Then for $p \geq \frac{k^2}{d+24a}$, we have

$$
f_a(p) \leq e^{\frac{48a(d+24a)}{2d}} p^{-a-\frac{d}{2}}. \tag{70}
$$

Finally we have

$$
\overline{\lambda_k} = 2\pi^{d/2} \frac{2^{\frac{d}{2}} e^{\frac{d}{2}} C_2^2 C}{C_1^2} \sum_{\substack{p \geq k \\ p-k \text{ is even}}} f_a(p)
$$

$$
\leq O\left(\sum_{\substack{k \leq p \leq \frac{k^2}{d+24a} \\ p-k \text{ is even}}} f_a(p) + \sum_{\substack{p \geq \frac{k^2}{d+24a} \\ p-k \text{ is even}}} f_a(p)\right)
$$

$$
\leq O\left(\left(\frac{k^2}{d+24a} - k + 1\right) f_a\left(\frac{k^2}{d+24a}\right) + \sum_{\substack{p \geq \frac{k^2}{d+24a} \\ p-k \text{ is even}}} e^{\frac{48a(d+24a)}{2d}} p^{-a-\frac{d}{2}}\right)
$$

$$
\leq O\left(\left(\frac{k^2}{d+24a} - k + 1\right) e^{\frac{48a(d+24a)}{2d}} \left(\frac{k^2}{d+24a}\right)^{-a-\frac{d}{2}} + e^{\frac{48a(d+24a)}{2d}} \frac{1}{a+\frac{d}{2}-1} \left(\frac{k^2}{d+24a} - 1\right)^{1-a-\frac{d}{2}}\right)
$$

$$
= O(k^{-d-2a+2}).
$$

Next we prove $\overline{\lambda_k} = \Omega(k^{-d-2a+2})$. Since $c_p$ are nonnegative and $c_p = \Theta(p^{-a})$, we have that $c_p \geq C' p^{-a}$ for some constant $C'$. Then we have

$$
\overline{\lambda_k} \geq \frac{\pi^{d/2}}{2^{k-1}} \sum_{\substack{p \geq k \\ p-k \text{ is even}}} C' p^{-a} \frac{\Gamma(p+1)\Gamma(\frac{p-k+1}{2})}{\Gamma(p-k+1)\Gamma(\frac{p-k+1}{2} + k + d/2)}. \tag{71}
$$

According to Stirling's formula (63) and (64), using the similar argument as (65) we have

$$
\overline{\lambda_k} \geq \frac{\pi^{d/2}}{2^{k-1}} \frac{C_1^2}{C_2^2} \sum_{\substack{p \geq k \\ p-k \text{ is even}}} C' p^{-a} \frac{\sqrt{\frac{2\pi}{p+1}}\left(\frac{p+1}{e}\right)^{p+1} \sqrt{\frac{2\pi}{\frac{p-k+1}{2}}}\left(\frac{\frac{p-k+1}{2}}{e}\right)^{\frac{p-k+1}{2}}}{\sqrt{\frac{2\pi}{p-k+1}}\left(\frac{p-k+1}{e}\right)^{p-k+1}\sqrt{\frac{2\pi}{\frac{p-k+1}{2}+k+d/2}}\left(\frac{\frac{p-k+1}{2}+k+d/2}{e}\right)^{\frac{p-k+1}{2}+k+d/2}}
$$

$$
\tag{72}
$$

$$
= 2\pi^{d/2} \frac{2^{\frac{d}{2}} e^{\frac{d}{2}} C_1^2 C'}{C_2^2} \sum_{\substack{p \geq k \\ p-k \text{ is even}}} \frac{p^{-a}(p+1)^{p+\frac{1}{2}}}{(p-k+1)^{\frac{p-k+1}{2}}(p+k+1+d)^{\frac{p+k+d}{2}}} \tag{73}
$$

$$
\geq 2\pi^{d/2} \frac{2^{\frac{d}{2}} e^{\frac{d}{2}} C_1^2 C'}{C_2^2} \sum_{\substack{p \geq k^2 \\ p-k \text{ is even}}} f_a(p), \tag{74}
$$

where $f_a(p)$ is defined in (66). When $p \geq k^2$, we have

$$
\begin{aligned}
f_a(p) &= \frac{p^{-a} (p+1)^{p+\frac{1}{2}}}{(p-k+1)^{\frac{p-k+1}{2}} (p+k+1+d)^{\frac{p+k+d}{2}}} \\
&= \frac{p^{-a} (p+1)^{p+\frac{1}{2}}}{((p+1)^2 - k^2 + d(p-k+1))^{\frac{p-k+1}{2}} (p+k+1+d)^{\frac{2k+d-1}{2}}} \\
&\geq \frac{(p+1)^{-a-\frac{d}{2}}}{\left(1 - \frac{k^2 - d(p-k+1)}{(p+1)^2}\right)^{\frac{p-k+1}{2}} \left(1 + \frac{k+d}{p+1}\right)^{\frac{2k+d-1}{2}}}
\end{aligned}
$$

For sufficiently large $k$, $k^2 - d(p-k+1) < 0$. Then we have

$$
\begin{aligned}
\left(1 - \frac{k^2 - d(p-k+1)}{(p+1)^2}\right)^{\frac{p-k+1}{2}} &= \left(1 - \frac{k^2 - d(p-k+1)}{(p+1)^2}\right)^{\frac{-(p+1)^2}{k^2 - d(p-k+1)} \cdot \frac{-k^2 + d(p-k+1)}{(p+1)^2} \cdot \frac{p-k+1}{2}} \\
&\leq e^{\frac{-k^2 + d(p-k+1)}{(p+1)^2} \cdot \frac{p-k+1}{2}} \\
&\leq e^{\frac{dp^2}{2p^2}} = e^{\frac{d}{2}}
\end{aligned}
$$

which is a constant independent of $k$. Also, for sufficiently large $k$, we have

$$
\begin{aligned}
\left(1 + \frac{k+d}{p+1}\right)^{\frac{2k+d-1}{2}} &= \left(1 + \frac{k+d}{p+1}\right)^{\frac{p+1}{k+d} \frac{k+d}{p+1} \frac{2k+d-1}{2}} \\
&\leq e^{\frac{k+d}{p+1} \frac{2k+d-1}{2}} \\
&\leq e^{\frac{3k^2}{2r}} = e^{\frac{3}{2}}
\end{aligned}
$$

Then for $p \geq k^2$, we have $f_a(p) \geq e^{-\frac{d}{2} - \frac{3}{2}} (p+1)^{-a-\frac{d}{2}}$.

Finally we have

$$
\overline{\lambda_k} \geq 2\pi^{d/2} \frac{2^{\frac{d}{2}} e^{\frac{d}{2}} C_1^2 C'}{C_2^2} \sum_{\substack{p \geq k^2 \\ p-k \text{ is even}}} f_a(p) \tag{75}
$$

$$
\geq 2\pi^{d/2} \frac{2^{\frac{d}{2}} e^{\frac{d}{2}} C_1^2 C'}{C_2^2} \sum_{\substack{p \geq k^2 \\ p-k \text{ is even}}} e^{-\frac{d}{2} - \frac{3}{2}} (p+1)^{-a-\frac{d}{2}} \tag{76}
$$

$$
\geq 2\pi^{d/2} \frac{2^{\frac{d}{2}} e^{\frac{d}{2}} C_1^2 C'}{C_2^2} e^{-\frac{d}{2} - \frac{3}{2}} \frac{1}{2(a + \frac{d}{2} - 1)} (k^2 + 2)^{1-a-\frac{d}{2}} \tag{77}
$$

$$
= \Omega(k^{-d-2a+2}). \tag{78}
$$

Overall, we have $\overline{\lambda_k} = \Theta(k^{-d-2a+2})$. □

*Proof of Corollary C.4, part 2.* It is easy to verify that $\overline{\lambda_k} = 0$ when $k$ is even because $c_p = 0$ when $p \geq k$ and $p - k$ is even. When $k$ is odd, the proof of Theorem 4.6 still applies. □

*Proof of Corollary C.4, part 3.* Since $c_p = \mathcal{O}\left(\exp\left(-a\sqrt{p}\right)\right)$, we have that $c_p \leq Ce^{-a\sqrt{p}}$ for some constant $C$. Similar to (65), we have

$$
\overline{\lambda_k} \leq 2\pi^{d/2} \frac{2^{\frac{d}{2}} e^{\frac{d}{2}} C_2^2 C}{C_1^2} \sum_{\substack{p \geq k \\ p-k \text{ is even}}} \frac{e^{-a\sqrt{p}} (p+1)^{p+\frac{1}{2}}}{(p-k+1)^{\frac{p-k+1}{2}} (p+k+1+d)^{\frac{p+k+d}{2}}}. \tag{79}
$$

Use the definition in (66) and let $a = 0$, we have

$$f_0(p) = \frac{(p+1)^{p+\frac{1}{2}}}{(p-k+1)^{\frac{p-k+1}{2}} (p+k+1+d)^{\frac{p+k+d}{2}}}. \tag{80}$$

Then according to (69) and (70), for sufficiently large $k$, we have $f_0(p) \le f_0\left(\frac{k^2}{d}\right)$ when $k \le p \le \frac{k^2}{d}$ and $f_0(p) \le C_3 p^{-\frac{d}{2}}$ for some constant $C_3$ when $p \ge \frac{k^2}{d}$. Then when $k \le p \le \frac{k^2}{d}$, we have $f_0(p) \le f_0\left(\frac{k^2}{d}\right) \le C_3 \left(\frac{k^2}{d}\right)^{-\frac{d}{2}}$. When $p \ge \frac{k^2}{d}$, we have $f_0(p) \le C_3 p^{-\frac{d}{2}} \le C_3 \left(\frac{k^2}{d}\right)^{-\frac{d}{2}}$. Overall, for all $p \ge k$, we have

$$f_0(p) \le C_3 \left(\frac{k^2}{d}\right)^{-\frac{d}{2}}. \tag{81}$$

Then we have

$$\overline{\lambda_k} \le 2\pi^{d/2} \frac{2^{\frac{d}{2}} e^{\frac{d}{2}} C_2^2 C}{C_1^2} \sum_{\substack{p \ge k \\ p-k \text{ is even}}} e^{-a\sqrt{p}} f_0(p) \tag{82}$$

$$\le 2\pi^{d/2} \frac{2^{\frac{d}{2}} e^{\frac{d}{2}} C_2^2 C_3 C}{C_1^2} \left(\frac{k^2}{d}\right)^{-\frac{d}{2}} \sum_{\substack{p \ge k \\ p-k \text{ is even}}} e^{-a\sqrt{p}} \tag{83}$$

$$\le 2\pi^{d/2} \frac{2^{\frac{d}{2}} e^{\frac{d}{2}} C_2^2 C_3 C}{C_1^2} \left(\frac{k^2}{d}\right)^{-\frac{d}{2}} \frac{2e^{-a\sqrt{k-1}}(a\sqrt{k-1}+1)}{a^2} \tag{84}$$

$$= \mathcal{O}\left(k^{-d+1/2} \exp\left(-a\sqrt{k}\right)\right) \tag{85}$$

$\square$

*Proof of Corollary C.4, part 4.* Since $c_p = \Theta(p^{1/2} a^{-p})$, we have that $c_p \le C p^{1/2} a^{-p}$ for some constant $C$. Similar to (65), we have

$$\overline{\lambda_k} \le 2\pi^{d/2} \frac{2^{\frac{d}{2}} e^{\frac{d}{2}} C_2^2 C}{C_1^2} \sum_{\substack{p \ge k \\ p-k \text{ is even}}} \frac{p^{1/2} a^{-p} (p+1)^{p+\frac{1}{2}}}{(p-k+1)^{\frac{p-k+1}{2}} (p+k+1+d)^{\frac{p+k+d}{2}}}. \tag{86}$$

Use the definition in (66) and let $a = 0$, we have

$$f_0(p) = \frac{(p+1)^{p+\frac{1}{2}}}{(p-k+1)^{\frac{p-k+1}{2}} (p+k+1+d)^{\frac{p+k+d}{2}}}. \tag{87}$$

Then according to (69) and (70), for sufficiently large $k$, we have $f_0(p) \le f_0\left(\frac{k^2}{d}\right)$ when $k \le p \le \frac{k^2}{d}$ and $f_0(p) \le C_3 p^{-\frac{d}{2}}$ for some constant $C_3$ when $p \ge \frac{k^2}{d}$. Then when $k \le p \le \frac{k^2}{d}$, we have $p^{1/2} f_0(p) \le p^{1/2} f_0\left(\frac{k^2}{d}\right) \le C_3 \left(\frac{k^2}{d}\right)^{1/2} \left(\frac{k^2}{d}\right)^{-\frac{d}{2}}$. When $p \ge \frac{k^2}{d}$, we have $p^{1/2} f_0(p) \le C_3 p^{1/2} p^{-\frac{d}{2}} \le C_3 \left(\frac{k^2}{d}\right)^{-\frac{d}{2}+\frac{1}{2}}$. Overall, for all $p \ge k$, we have

$$p^{1/2} f_0(p) \le C_3 \left(\frac{k^2}{d}\right)^{-\frac{d}{2}+\frac{1}{2}}. \tag{88}$$

Then we have

$$\overline{\lambda_k} \le 2\pi^{d/2} \frac{2^{\frac{d}{2}} e^{\frac{d}{2}} C_2^2 C}{C_1^2} \sum_{\substack{p \ge k \\ p-k \text{ is even}}} p^{1/2} a^{-p} f_0(p) \tag{89}$$

$$\le 2\pi^{d/2} \frac{2^{\frac{d}{2}} e^{\frac{d}{2}} C_2^2 C_3 C}{C_1^2} \left(\frac{k^2}{d}\right)^{-\frac{d}{2}+\frac{1}{2}} \sum_{\substack{p \ge k \\ p-k \text{ is even}}} a^{-p} \tag{90}$$

$$\le 2\pi^{d/2} \frac{2^{\frac{d}{2}} e^{\frac{d}{2}} C_2^2 C_3 C}{C_1^2} \left(\frac{k^2}{d}\right)^{-\frac{d}{2}+\frac{1}{2}} \frac{1}{\log a} a^{-(k-1)} \tag{91}$$

$$= \mathcal{O}\left(k^{-d+1} a^{-k}\right). \tag{92}$$

On the other hand, since $c_p = \Theta(p^{1/2} a^{-p})$, we have that $c_p \ge C' p^{1/2} a^{-p}$ for some constant $C'$. Similar to (73), we have

$$\overline{\lambda_k} \ge 2\pi^{d/2} \frac{2^{\frac{d}{2}} e^{\frac{d}{2}} C_1^2 C'}{C_2^2} \sum_{\substack{p \ge k \\ p-k \text{ is even}}} \frac{p^{1/2} a^{-p} (p+1)^{p+\frac{1}{2}}}{(p-k+1)^{\frac{p-k+1}{2}} (p+k+1+d)^{\frac{p+k+d}{2}}} \tag{93}$$

$$\ge 2\pi^{d/2} \frac{2^{\frac{d}{2}} e^{\frac{d}{2}} C_1^2 C'}{C_2^2} \frac{k^{1/2} a^{-k} (k+1)^{k+\frac{1}{2}}}{(k-k+1)^{\frac{k-k+1}{2}} (k+k+1+d)^{\frac{k+k+d}{2}}} \tag{94}$$

$$= \Omega\left(\frac{k^{-d/2+1} a^{-k} (k+1)^k}{(k+k+1+d)^k}\right). \tag{95}$$

Since $(k+1)^k = k^k (1+1/k)^k = \Theta(k^k)$. Similarly, $(k+k+1+d)^k = \Theta((2k)^k)$. Then we have

$$\overline{\lambda_k} = \Omega\left(\frac{k^{-d/2+1} a^{-k} (k+1)^k}{(k+k+1+d)^k}\right) \tag{96}$$

$$= \Omega\left(\frac{k^{-d/2+1} a^{-k} k^k}{(2k)^k}\right) \tag{97}$$

$$= \Omega\left(k^{-d/2+1} 2^{-k} a^{-k}\right). \tag{98}$$

$\square$

For the NTK of a two-layer ReLU network with $\gamma_b > 0$, then according to Lemma 3.2 we have $c_p = \kappa_{p,2} = \Theta(p^{-3/2})$. Therefore using Corollary 4.7 $\overline{\lambda_k} = \Theta(k^{-d-1})$. Notice here that $k$ refers to the frequency, and the number of spherical harmonics of frequency at most $k$ is $\Theta(k^d)$. Therefore, for the $\ell$th largest eigenvalue $\lambda_\ell$ we have $\lambda_\ell = \Theta(\ell^{-(d+1)/d})$. This rate agrees with Basri et al. (2019) and Velikanov & Yarotsky (2021). For the NTK of a two-layer ReLU network with $\gamma_b = 0$, the eigenvalues corresponding to the even frequencies are 0, which also agrees with Basri et al. (2019). Corollary 4.7 also shows the decay rates of eigenvalues for the NTK of two-layer networks with Tanh activation and Gaussian activation. We observe that when the coefficients of the kernel power series decay quickly then the eigenvalues of the kernel also decay quickly. As a faster decay of the eigenvalues of the kernel implies a smaller RKHS, Corollary 4.7 demonstrates that using ReLU results in a larger RKHS relative to using either Tanh or Gaussian activations. We numerically illustrate Corollary 4.7 in Figure 4, Appendix C.3.

## C.5    ANALYSIS OF THE LOWER SPECTRUM: NON-UNIFORM DATA

The purpose of this section is to prove a formal version of Theorem 4.8. In order to prove this result we first need the following lemma.

**Lemma C.20.** *Let the coefficients $(c_j)_{j=0}^\infty$ with $c_j \in \mathbb{R}_{\ge 0}$ for all $j \in \mathbb{Z}_{\ge 0}$ be such that the series $\sum_{j=0}^\infty c_j \rho^j$ converges for all $\rho \in [-1, 1]$. Given a data matrix $\mathbf{X} \in \mathbb{R}^{n \times d}$ with $\|\mathbf{x}_i\| = 1$ for all*

$i \in [n]$, *define* $r := \text{rank}(\mathbf{X}) \geq 2$ *and the gram matrix* $\mathbf{G} := \mathbf{X}\mathbf{X}^T$. *Consider the kernel matrix*

$$n\mathbf{K} = \sum_{j=0}^{\infty} c_j \mathbf{G}^{\odot j}.$$

*For arbitrary* $m \in \mathbb{Z}_{\geq 1}$, *let the eigenvalue index* $k$ *satisfy* $n \geq k > \text{rank}(\mathbf{H}_m)$, *where* $\mathbf{H}_m := \sum_{j=0}^{m-1} c_j \mathbf{G}^{\odot j}$. *Then*

$$\lambda_k(\mathbf{K}) \leq \frac{\|\mathbf{G}^{\odot m}\|}{n} \sum_{j=m}^{\infty} c_j. \tag{99}$$

*Proof.* We start our analysis by considering $\lambda_k(n\mathbf{K})$ for some arbitrary $k \in \mathbb{N}_{\leq n}$. Let

$$\mathbf{H}_m := \sum_{j=0}^{m-1} c_j \mathbf{G}^{\odot j},$$

$$\mathbf{T}_m := \sum_{j=m}^{\infty} c_j \mathbf{G}^{\odot j}$$

be the $m$-head and $m$-tail of the Hermite expansion of $n\mathbf{K}$: clearly $n\mathbf{K} = \mathbf{H}_m + \mathbf{T}_m$ for any $m \in \mathbb{N}$. Recall that a constant matrix is symmetric and positive semi-definite, furthermore, by the Schur product theorem, the Hadamard product of two positive semi-definite matrices is positive semi-definite. As a result, $\mathbf{G}^{\odot j}$ is symmetric and positive semi-definite for all $j \in \mathbb{Z}_{\geq 0}$ and therefore $\mathbf{H}_m$ and $\mathbf{T}_m$ are also symmetric positive semi-definite matrices. From Weyl's inequality (Weyl, 1912, Satz 1) it follows that

$$n\lambda_k(\mathbf{K}) \leq \lambda_k(\mathbf{H}_m) + \lambda_1(\mathbf{T}_m). \tag{100}$$

In order to upper bound $\lambda_1(\mathbf{T}_m)$, observe, as $\mathbf{T}_m$ is square, symmetric and positive semi-definite, that $\lambda_1(\mathbf{T}_m) = \|\mathbf{T}_m\|$. Using the non-negativity of the coefficients $(c_j)_{j=0}^{\infty}$ and the triangle inequality we have

$$\lambda_1(\mathbf{T}_m) = \left\| \sum_{j=m}^{\infty} c_j \mathbf{G}^{\odot j} \right\| \leq \sum_{j=m}^{\infty} c_j \|\mathbf{G}^{\odot j}\|$$

By the assumptions of the lemma $[\mathbf{G}]_{ii} = 1$ and therefore $[\mathbf{G}]_{ii}^j = 1$ for all $j \in \mathbb{Z}_{\geq 0}$. Furthermore, for any pair of positive semi-definite matrices $\mathbf{A}, \mathbf{B} \in \mathbb{R}^{n \times n}$ and $k \in [n]$

$$\lambda_1(\mathbf{A} \odot \mathbf{B}) \leq \max_{i \in [n]} [\mathbf{A}]_{ii} \lambda_1(\mathbf{B}), \tag{101}$$

Schur (1911). Therefore, as $\max_{i \in [n]} [\mathbf{G}]_{ii} = 1$,

$$\|\mathbf{G}^{\odot j}\| = \lambda_1(\mathbf{G}^{\odot j}) = \lambda_1(\mathbf{G} \odot \mathbf{G}^{\odot(j-1)}) \leq \lambda_1(\mathbf{G}^{\odot(j-1)}) = \|\mathbf{G}^{\odot(j-1)}\|$$

for all $j \in \mathbb{N}$. As a result

$$\lambda_1(\mathbf{T}_m) \leq \|\mathbf{G}^{\odot m}\| \sum_{j=m}^{\infty} c_j.$$

Finally, we now turn our attention to the analysis of $\lambda_k(\mathbf{H}_m)$. Upper bounding a small eigenvalue is typically challenging, however, the problem simplifies when and $k$ exceeds the rank of $\mathbf{H}_m$, as is assumed here, as this trivially implies $\lambda_k(\mathbf{H}_m) = 0$. Therefore, for $k > \text{rank}(\mathbf{H}_m)$

$$\lambda_k(\mathbf{K}) \leq \frac{\|\mathbf{G}^m\|}{n} \sum_{j=m}^{\infty} c_j$$

as claimed. $\qquad\square$

In order to use Lemma C.20 we require an upper bound on the rank of $\mathbf{H}_m$. To this end we provide Lemma C.21.

**Lemma C.21.** *Let* $\mathbf{G} \in \mathbb{R}^{n \times n}$ *be a symmetric, positive semi-definite matrix of rank* $2 \leq r \leq d$. *Define* $\mathbf{H}_m \in \mathbb{R}^{n \times n}$ *as*

$$\mathbf{H}_m = \sum_{j=0}^{m-1} c_j \mathbf{G}^{\odot j} \tag{102}$$

*where* $(c_j)_{j=0}^{m-1}$ *is a sequence of real coefficients. Then*

$$\begin{aligned}
rank\,(\mathbf{H}_m) \leq &1 + \min\{r - 1, m - 1\}(2e)^{r-1} \\
&+ \max\{0, m - r\} \left(\frac{2e}{r-1}\right)^{r-1} (m-1)^{r-1}.
\end{aligned} \tag{103}$$

*Proof.* As $\mathbf{G}$ is a symmetric and positive semi-definite matrix, its eigenvalues are real and non-negative and its eigenvectors are orthogonal. Let $\{\mathbf{v}_i\}_{i=1}^r$ be a set of orthogonal eigenvectors for $\mathbf{G}$ and $\gamma_i$ the eigenvalue associated with $\mathbf{v}_i \in \mathbb{R}^n$. Then $\mathbf{G}$ may be written as a sum of rank one matrices as follows,

$$\mathbf{G} = \sum_{i=1}^r \gamma_i \mathbf{v}_i \mathbf{v}_i^T.$$

As the Hadamard product is commutative, associative and distributive over addition, for any $j \in \mathbb{Z}_{\geq 0}$ $\mathbf{G}^{\odot j}$ can also be expressed as a sum of rank 1 matrices,

$$\begin{aligned}
\mathbf{G}^{\odot j} &= \left(\sum_{i=1}^r \gamma_i \mathbf{v}_i \mathbf{v}_i^T\right)^{\odot j} \\
&= \left(\sum_{i_1=1}^r \gamma_{i_1} \mathbf{v}_{i_1} \mathbf{v}_{i_1}^T\right) \odot \left(\sum_{i_2=1}^r \gamma_{i_2} \mathbf{v}_{i_2} \mathbf{v}_{i_2}^T\right) \odot \cdots \odot \left(\sum_{i_j=1}^r \gamma_{i_j} \mathbf{v}_{i_j} \mathbf{v}_{i_j}^T\right) \\
&= \sum_{i_1,i_2\ldots i_j=1}^r \gamma_{i_1} \gamma_{i_2} \cdots \gamma_{i_r} \left(\mathbf{v}_{i_1} \mathbf{v}_{i_1}^T\right) \odot \left(\mathbf{v}_{i_2} \mathbf{v}_{i_2}^T\right) \odot \cdots \odot \left(\mathbf{v}_{i_j} \mathbf{v}_{i_j}^T\right) \\
&= \sum_{i_1,i_2,\ldots,i_j=1}^r \gamma_{i_1} \gamma_{i_2} \cdots \gamma_{i_j} \left(\mathbf{v}_{i_1} \odot \mathbf{v}_{i_2} \odot \cdots \odot \mathbf{v}_{i_j}\right) \left(\mathbf{v}_{i_1} \odot \mathbf{v}_{i_2} \odot \cdots \odot \mathbf{v}_{i_j}\right)^T.
\end{aligned}$$

Note the fourth equality in the above follows from $\mathbf{v}_i \mathbf{v}_i^T = \mathbf{v}_i \otimes \mathbf{v}_i$ and an application of the mixed-product property of the Hadamard product. As matrix rank is sub-additive, the rank of $\mathbf{G}^{\odot j}$ is less than or equal to the number of distinct rank-one matrix summands. This quantity in turn is equal to the number of vectors of the form $\left(\mathbf{v}_{i_1} \odot \mathbf{v}_{i_2} \odot \cdots \odot \mathbf{v}_{i_j}\right)$, where $i_1, i_2, \ldots, i_j \in [r]$. This in turn is equivalent to computing the number of $j$-combinations with repetition from $r$ objects. Via a stars and bars argument this is equal to $\binom{r+j-1}{j} = \binom{r+j-1}{r(n)-1}$. It therefore follows that

$$\begin{aligned}
\text{rank}(\mathbf{G}^{\odot j}) &\leq \binom{r+j-1}{r-1} \\
&\leq \left(\frac{e(r+j-1)}{r-1}\right)^{r-1} \\
&\leq e^{r-1} \left(1 + \frac{j}{r-1}\right)^{r-1} \\
&\leq (2e)^{r-1} \left(\delta_{j \leq r-1} + \delta_{j > r-1} \left(\frac{j}{r-1}\right)^{r-1}\right).
\end{aligned}$$

The rank of $\mathbf{H}_m$ can therefore be bounded via subadditivity of the rank as

$$
\begin{aligned}
\mathrm{rank}(\mathbf{H}_m) =& \mathrm{rank}\left(a_0 \mathbf{1}_{n \times n} + \sum_{j=1}^{m-1} c_j \mathbf{G}^{\odot j}\right) \\
\leq& 1 + \sum_{j=1}^{m-1} \mathrm{rank}\left(\mathbf{G}^{\odot j}\right) \\
\leq& 1 + \sum_{j=1}^{m-1} (2e)^{r-1}\left(\delta_{j \leq r-1} + \delta_{j > r-1}\left(\frac{j}{r-1}\right)^{r-1}\right) \\
\leq& 1 + \min\{r-1, m-1\}(2e)^{r-1} \\
&+ \max\{0, m-r\}\left(\frac{2e}{r-1}\right)^{r-1}(m-1)^{r-1}.
\end{aligned}
\tag{104}
$$

$\square$

As our goal here is to characterize the small eigenvalues, then as $n$ grows we need both $k$ and therefore $m$ to grow as well. As a result we will therefore be operating in the regime where $m > r$. To this end we provide the following corollary.

**Corollary C.22.** *Under the same conditions and setup as Lemma C.21 with $m \geq r \geq 7$ then*

$$
\mathrm{rank}(\mathbf{H}_m) < 2m^r.
$$

*Proof.* If $r \geq 7 > 2e + 1$ then $r - 1 > 2e$. As a result from Lemma C.21

$$
\begin{aligned}
\mathrm{rank}(\mathbf{H}_m) &\leq 1 + (r-1)(2e)^{r-1} + (m-r)\left(\frac{2e}{r-1}\right)^{r-1}(m-1)^{r-1} \\
&< r(2e)^{r-1} + (m-1)^r \\
&< 2m^r
\end{aligned}
$$

as claimed. $\square$

Corollary C.22 implies for any $k \geq 2m^r$, $k \leq n$ that we can apply Lemma C.20 to upper bound the size of the $k$th eigenvalue. Our goal is to upper bound the decay of the smallest eigenvalue. To this end, and in order to make our bounds as tight as possible, we therefore choose the truncation point $m(n) = \lfloor (n/2)^{1/r} \rfloor$, note this is the largest truncation which still satisfies $2m(n)^r \leq n$. In order to state the next lemma, we introduce the following pieces of notation: with $\mathcal{L} := \{\ell : \mathbb{R}_{\geq 0} \to \mathbb{R}_{\geq 0}\}$ define $U : \mathcal{L} \times \mathbb{Z}_{\geq 1} \to \mathbb{R}_{\geq 0}$ as

$$
U(\ell, m) = \int_{m-1}^{\infty} \ell(x)dx.
$$

**Lemma C.23.** *Given a sequence of data points $(\mathbf{x}_i)_{i \in \mathbb{Z}_{\geq 1}}$ with $\mathbf{x}_i \in \mathbb{S}^d$ for all $i \in \mathbb{Z}_{\geq 1}$, construct a sequence of row-wise data matrices $(\mathbf{X}_n)_{n \in \mathbb{Z}_{\geq 1}}$, $\mathbf{X}_n \in \mathbb{R}^{n \times d}$, with $\mathbf{x}_i$ corresponding to the ith row of $\mathbf{X}_n$. The corresponding sequence of gram matrices we denote $\mathbf{G}_n := \mathbf{X}_n \mathbf{X}_n^T$. Let $m(n) := \lfloor (n/2)^{1/r(n)} \rfloor$ where $r(n) := \mathrm{rank}(\mathbf{X}_n)$ and suppose for all sufficiently large $n$ that $m(n) \geq r(n) \geq 7$. Let the coefficients $(c_j)_{j=0}^{\infty}$ with $c_j \in \mathbb{R}_{\geq 0}$ for all $j \in \mathbb{Z}_{\geq 0}$ be such that 1) the series $\sum_{j=0}^{\infty} c_j \rho^j$ converges for all $\rho \in [-1, 1]$ and 2) $(c_j)_{j=0}^{\infty} = \mathcal{O}(\ell(j))$, where $\ell \in \mathcal{L}$ satisfies $U(\ell, m(n)) < \infty$ for all $n$ and is monotonically decreasing. Consider the sequence of kernel matrices indexed by $n$ and defined as*

$$
n\mathbf{K}_n = \sum_{j=0}^{\infty} c_j \mathbf{G}_n^{\odot j}.
$$

*With $\nu : \mathbb{Z}_{\geq 1} \to \mathbb{Z}_{\geq 1}$ suppose $\left\| \mathbf{G}_n^{\odot m(n)} \right\| = \mathcal{O}(n^{-\nu(n)+1})$, then*

$$
\lambda_n(\mathbf{K}_n) = \mathcal{O}(n^{-\nu(n)}U(\ell, m(n))).
\tag{105}
$$

*Proof.* By the assumptions of the Lemma we may apply Lemma C.20 and Corollary C.22, which results in

$$\lambda_n(\mathbf{K}_n) \leq \frac{\left\| \mathbf{G}_n^{\odot m(n)} \right\|}{n} \sum_{j=m(n)}^{\infty} c_j = \mathcal{O}(n^{-\nu(n)}) \sum_{j=m(n)}^{\infty} c_j.$$

Additionally, as $(c_j)_{j=0}^{\infty} = \mathcal{O}(\ell(j))$ then

$$\lambda_n(\mathbf{K}_n) = \mathcal{O}\left( n^{-\nu(n)} \sum_{j=m(n)}^{\infty} \ell(j) \right)$$

$$= \mathcal{O}\left( n^{-\nu(n)} \int_{m(n)-1}^{\infty} \ell(x)dx \right)$$

$$= \mathcal{O}\left( n^{-\nu(n)} U(\ell, m(n)) \right)$$

as claimed. □

Based on Lemma C.20 we provide Theorem C.24, which considers three specific scenarios for the decay of the power series coefficients inspired by Lemma 3.2.

**Theorem C.24.** *In the same setting, and also under the same assumptions as in Lemma C.23, then*

1. *if $c_p = \mathcal{O}(p^{-\alpha})$ with $\alpha > r(n) + 1$ for all $n \in \mathbb{Z}_{\geq 0}$ then $\lambda_n(\mathbf{K}_n) = \mathcal{O}\left( n^{-\frac{\alpha-1}{r(n)}} \right)$,*

2. *if $c_p = \mathcal{O}(e^{-\alpha\sqrt{p}})$, then $\lambda_n(\mathbf{K}_n) = \mathcal{O}\left( n^{\frac{1}{2r(n)}} \exp\left( -\alpha' n^{\frac{1}{2r(n)}} \right) \right)$ for any $\alpha' < \alpha 2^{-1/2r(n)}$,*

3. *if $c_p = \mathcal{O}(e^{-\alpha p})$, then $\lambda_n(\mathbf{K}_n) = \mathcal{O}\left( \exp\left( -\alpha' n^{\frac{1}{r(n)}} \right) \right)$ for any $\alpha' < \alpha 2^{-1/2r(n)}$.*

*Proof.* First, as $[\mathbf{G}_n]_{ij} \leq 1$ then

$$\frac{\left\| \mathbf{G}^{\odot m(n)} \right\|}{n} \leq \frac{\text{Trace}(\mathbf{G}^{\odot m(n)})}{n} = 1.$$

Therefore, to recover the three results listed we now apply Lemma C.23 with $\nu(n) = 0$. First, to prove 1., under the assumption $\ell(x) = x^{-\alpha}$ with $\alpha > 0$ then

$$\int_{m(n)-1}^{\infty} x^{-\alpha} dx = \frac{(m(n)-1)^{-\alpha+1}}{\alpha - 1}.$$

As a result

$$\lambda_n(\mathbf{K}_n) = \mathcal{O}\left( n^{-\frac{\alpha-1}{r(n)}} \right).$$

To prove ii), under the assumption $\ell(x) = e^{-\alpha\sqrt{x}}$ with $\alpha > 0$ then

$$\int_{m(n)-1}^{\infty} e^{-\alpha\sqrt{x}} dx = \frac{2\exp(-\alpha(\sqrt{m(n)-1})(\alpha\sqrt{m(n)-1}+1)}{\alpha^2}.$$

As a result

$$\lambda_n(\mathbf{K}_n) = \mathcal{O}\left( n^{\frac{1}{2r(n)}} \exp\left( -\alpha' n^{\frac{1}{2r(n)}} \right) \right)$$

for any $\alpha' < \alpha 2^{-1/2r(n)}$. Finally, to prove iii), under the assumption $\ell(x) = e^{-\alpha x}$ with $\alpha > 0$ then

$$\int_{m(n)-1}^{\infty} e^{-\alpha x} dx = \frac{\exp(-\alpha(m(n)-1))}{\alpha}.$$

Therefore

$$\lambda_n(\mathbf{K}_n) = \mathcal{O}\left( \exp\left( -\alpha' n^{\frac{1}{r(n)}} \right) \right)$$

again for any $\alpha' < \alpha 2^{-1/2r(n)}$. □

Unfortunately, the curse of dimensionality is clearly present in these results due to the $1/r(n)$ factor in the exponents of $n$. However, although perhaps somewhat loose we emphasize that these results are certainly far from trivial. In particular, while trivially we know that $\lambda_n(\mathbf{K}_n) \leq Tr(\mathbf{K}_n)/n = \mathcal{O}(n^{-1})$, in contrast, even the weakest result concerning the power law decay our result is a clear improvement as long as $\alpha > r(n) + 1$. For the other settings, i.e., those specified in 2. and 3., our results are significantly stronger.

