# OpenReview forum: "Characterizing the spectrum of the NTK via a power series expansion"
_ICLR.cc/2023/Conference — ICLR 2023 poster_

### Official Review · Reviewer_VHSv · 2022-10-24

**Confidence:** 4
**Correctness:** 4
**Technical Novelty And Significance:** 2
**Empirical Novelty And Significance:** Not applicable
**Recommendation:** 6

**Clarity, Quality, Novelty And Reproducibility:**

The paper is well-written but some statements of this paper need to be written more formally. For example, the authors argue that the effective rank of the shallow ReLU is roughly 2.5. But, it is unclear how the network depth is small enough. Similar issue is in Observation 4.4.

Novelty is incremental as the main analysis tool (Hermite expansion <-> NTK) was already studied in many previous works.


**Strength And Weaknesses:**

Strength:
- This paper provides a power series of fully-connected NTK (Theorem 3.1) when inputs are normalized. This can be obtained by Hermite expansion of the activation and polynomial composition. Hence, once the coefficients of the series of NTK is computed (which is data-independent), then the NTK is computed by applying some polynomial to each entry of the data Gram matrix.

- The authors make use of the power series of the NTK to analyze its effective rank. They particularly compute the ratio of effective ranks between the NTK and the data Gram matrix, and show that this can be small for common activations (e.g., ReLU).

Weakness:
- The main focus of the spectrum analysis part is the effective rank. The authors mention that it can quantify some counts of eigenvalues of the NTK. But, it needs more comprehensive motivation why the effective rank analysis is meaningful. And what is expected when the effective rank of the NTK is close to that of the data Gram matrix?

  One question is that Figure 2 looks that the spectrum of the NTK with ``tanh’’ activation matches the data Gram matrix more than that of the ReLU. However, in practice, it is well-known that the ReLU activation performs better than the tanh in terms of training and test (for both CIFAR-10 and MNIST). It is uncertain that such spectrum is meaningful to analyze the NTK.

- The paper should add/edit more related works. The relationship to the Hermite expansion was originally studied by [1], which actually characterizes the NTK for shallow networks. There is also rich literature studying the Hermite polynomials for the NTK, e.g., [2,3,4].

- The paper entirely assumes that inputs are normalized, which can be fairly unrealistic because practical datasets are not. Under this assumption, I think the contributions are quite incremental, which can be derived from the simple composition of power series (coefficients are obtained from the Hermite expansion). For homogeneous activation (e.g., ReLU), the results can be easily extended to the entire R^d (also studied in [3]). But, generally, it seems that the results can be more complicated and different.

- Moreover, the results of this paper are limited only to the ``fully-connected’’ neural network. Practical usage of the NTK is also related to the convolutional neural networks or more results on modern architectures would improve the results.

[1] Amit Daniely, Roy Frostig, and Yoram Singer. Toward deeper understanding of neural networks: The power of initialization and a dual view on expressivity. In NeurIPS, 2016.

[2] James B Simon, Sajant Anand, and Michael R DeWeese. Reverse Engineering the Neural Tangent Kernel. In ICML, 2022.

[3] Insu Han, Amir Zandieh, Jaehoon Lee, Roman Novak, Lechao Xiao and Amin Karbasi.Fast Neural Kernel Embeddings for General Activations. In NeurIPS, 2022.

[4] Lokhande, Vishnu Suresh, Songwong Tasneeyapant, Abhay Venkatesh, Sathya N. Ravi, and Vikas Singh. Generating accurate pseudo-labels in semi-supervised learning and avoiding overconfident predictions via Hermite polynomial activations. In CVPR, 2020.


**Summary Of The Paper:**

This paper studies a power series of the neural tangent kernel (NTK) for an infinitely wide fully-connected neural network when inputs are on the unit sphere. Using the power series, the authors analyze the spectrum of the NTK. In particular, they focus on the effective rank of the NTK matrix, defined as the ratio of the trace to the leading eigenvalue, and compare it to that of the data Gram matrix. For shallow ReLU network, the effective rank of the NTK is a constant multiple of that of the data Gram matrix. Finally, they provide the lower part of the eigenvalues using the spherical harmonic analysis.

**Summary Of The Review:**

The results of this paper are incremental and the motivation of the effective rank ratio of the NTK is unclear.

---

> ### Author Response · Authors · 2022-11-14
> **Initial response to reviewer VHSv (part 1)**
>
> Thank you for taking the time to review our paper.  We appreciate your comments about the usefulness of the power series expansion with regards to computation and analyzing the effective rank, as well as the comment that the paper is well-written. You raise some important and valid questions which we answer in turn below. Before tackling each of these we would like to highlight however that perhaps **the most significant contribution of our work, the analysis of the asymptotic decay of the spectrum and its relation to the NTK power series coefficients, is absent from your review.** We appreciate that we should have perhaps signposted and highlighted this result more clearly and apologize if it wasn’t clear.  You raise some important points in your review, which we address below.
> * In your summary you state "For shallow ReLU network, the effective rank of the NTK is a constant multiple of that of the data Gram matrix.".  We would like to emphasize that while the finite width result Theorem 4.5 assumes a shallow ReLU network, for infinite width networks we are not limited to the shallow case nor are we limited to the ReLU activation.  Theorem 4.3 holds for any kernel with a power series expansion and Theorem 3.1 provides the power series expansion for the NTK for general activations and depths.
> * In regard to "The main focus of the spectrum analysis part is the effective rank."  We would like to emphasize that while the effective rank results appear first, we do not consider them our main results and are merely one component of the work.  Our asymptotic results provided in Section 4.2, in particular Theorem 4.6, Corollary 4.7, and Theorem 4.8 are perhaps our biggest contribution, and are not discussed in the review.  We chose to put the effective rank results first because they are simpler to explain and derive, and thus serve as a nice warm up to introduce the usefulness of a power series characterization.
> * In regard to "The authors mention that it [the effective rank] can quantify some counts of eigenvalues of the NTK. But, it needs more comprehensive motivation why the effective rank analysis is meaningful. And what is expected when the effective rank of the NTK is close to that of the data Gram matrix?"  As we mention in the introduction, the spectrum of the NTK has many consequences for convergence and generalization.  In regard to convergence, the convergence rates of different eigenvectors of the NTK is given by their eigenvalues [A].  Thus if the NTK has a small number of large outlier eigenvalues (and hence a low effective rank), this implies that only a small number of components can be learned quickly and that the rest may take a much longer time in comparison to converge.  Furthermore, the spectrum of the NTK is relevant for generalization bounds.  For example, in Lemma 22 in [B] it is shown that the Rademacher complexity of a kernel method within a parameter ball is determined by the trace of the kernel and the radius of the ball.  This bound is instantiated by [A] to provide a generalization bound for the NTK.  We note that $\lambda_1(K) = O(1)$, and thus consequently whenever the NTK has low effective rank it will have limited capacity within bounded parameter balls.  Since real world data often has low effective rank, this means that the model has limited capacity until the late stages of training when the parameters have varied significantly.  **We have added a comment about the implications of the effective rank on the model complexity at the end of Section 4.1**

---

> ### Author Response · Authors · 2022-11-14
> **Initial response to reviewer VHSv (part 2)**
>
> * In regard to "One question is that Figure 2 looks that the spectrum of the NTK with ``tanh’’ activation matches the data Gram matrix more than that of the ReLU. However, in practice, it is well-known that the ReLU activation performs better than the tanh in terms of training and test (for both CIFAR-10 and MNIST). It is uncertain that such spectrum is meaningful to analyze the NTK."
>      * As for why the spectrum of the NTK with tanh matches the data more than that of ReLU,  the coefficients for Tanh decay quickly relative to ReLU, and therefore the resulting spectrum more closely follows the spectrum of the input data. Intuitively this should not be surprising as close to the origin Tanh is well approximated by the identity.  We refer the reviewer also to our comments in response to reviewer rRGi.
>      * In regard to your comment "it is well-known that the ReLU activation performs better than the tanh in terms of training and test (for both CIFAR-10 and MNIST). It is uncertain that such spectrum is meaningful to analyze the NTK.",  we emphasize that we never claim that tanh will outperform ReLU, nor do we claim that the NTK having a low effective rank is a good property in general.  Having a low effective rank implies that the capacity of the kernel is more limited, which can lead to worse performance if the target function does not align with the kernel.  Having a low effective rank is good if the target function aligns with the kernel, as it means that you can get stronger generalization bounds.  Thus the effective rank being small is neither good nor bad in general, it depends on the data and target. It is highly informative however for the reasons given in the previous bullet points.
>
> *  In regard to "The paper should add/edit more related works. The relationship to the Hermite expansion was originally studied by [1], which actually characterizes the NTK for shallow networks. There is also rich literature studying the Hermite polynomials for the NTK, e.g., [2,3,4].",  we address each of these references individually below.
>      * In regard to "Amit Daniely, Roy Frostig, and Yoram Singer. Toward deeper understanding of neural networks: The power of initialization and a dual view on expressivity. In NeurIPS, 2016.", before we discuss the paper itself we wish to highlight that we do actually cite it in Lemma A.4 in Appendix A.4. However, we agree it was an oversight to not also cite it in the related work section and have happily amended this.  **We have added a citation for this work in the "Hermite Expansion" segment of the related work section.**  With regard to its contents, the primary goal of this paper is to study the expressive power of networks at initialization, a key takeaway being that a rich class of functions can be approximated by wide, randomly initialized networks in which only the outer layer of the network is trained. Such a model, where only the outer layer is trained, corresponds to what is now frequently referred to as the conjugate kernel: we note that this paper predates and anticipates the NTK, CK and NNGP terminology.  **We note that the Neural Tangent Kernel was introduced in 2018 [E] which came after this work.**  Of particular relevance is the introduction of the concept of a dual activation, equivalent to the expectation in equation 19 of our paper, for which a power series, taking as argument the input correlation and whose coefficients depend on the Hermite coefficients of the activation function used, is derived. Theorem 3.1 builds on this aspect of their work by studying not just the CK but also the NTK, and also provides explicit expressions for the coefficients arising from the composition of kernels. Using this power series expansion for the NTK we proceed to analyze the spectrum of the NTK, which we believe is our primary contribution, and is not something studied in this work.
>      * In regard to "James B Simon, Sajant Anand, and Michael R DeWeese. Reverse Engineering the Neural Tangent Kernel. In ICML, 2022.", this paper demonstrates that any dot product kernel can be realized using an NNGP or NTK corresponding to an infinitely-wide, zero bias, single hidden layer network equipped with a carefully chosen activation function. As such depth is not necessary for expressivity. **This is again an interesting work and we have happily amended the "Hermite Expansion" segment of the related work section section to include it.** However, we wish to highlight that the objectives and focus of this work and ours are different. In particular, we seek to understand the spectrum of the NTK associated with an arbitrary feedforward network, and do so by analyzing its power series coefficients, which in turn depend on both the depth and activation function deployed. The spectrum of the NTK is not discussed or analyzed in their work.

---

> ### Author Response · Authors · 2022-11-14
> **Initial response to reviewer VHSv (part 3)**
>
> * We continue discussing the references listed by the reviewer below:
>      * In regard to "Insu Han, Amir Zandieh, Jaehoon Lee, Roman Novak, Lechao Xiao and Amin Karbasi. Fast Neural Kernel Embeddings for General Activations. In NeurIPS, 2022." This work derives a power series expansion for the dual kernel in terms of the hermite expansion of the activation function.  They also provide an expression for the dual kernel of the activation's derivative that involves differentiating the dual kernel. Consequently the NTK and NNGP can be approximated by polynomials, and the tensor products of inputs can be approximated by sketching.  They use this to derive Algorithm 1 which applies to homogenous kernels, and they offer a theoretical bound on the accuracy of Algorithm 1. They also validate the effectiveness of Algorithm 1 experimentally.  In the Appendix they also discuss the computation of the CNTK.  This is a very interesting work which indeed has some connections to ours, but also has some important differences.  The main connection between our work and theirs is that they also investigate approximating the NTK using a truncated power series expansion. **However, unlike our work, the work [3] does not consider nonzero biases, it does not explicitly derive expressions for the coefficients of deep networks (which is important for the analysis of the spectrum), their result on deep networks applies only for homogenous activations, and, most importantly, there is no analysis of the effective rank nor an asymptotic characterization of the spectrum.**  If we restricted ourselves to homogeneous activations then we could relax our assumption on the input normalization. However, this would greatly reduce the class of activation functions our results apply to.  Nevertheless, this work is very interesting and connects with our work, and thus we have updated the manuscript to properly discuss this reference.  We emphasize also that the manuscript [3] was posted September 9th, which is within 30 days of our ICLR submission September 28. **Therefore this should be considered as a concurrent submission (per the guidelines of NeurIPS, ICML, and ICLR)**, and as a result we should not be penalized on the basis of missing this reference.
>      * In regard to "Lokhande, Vishnu Suresh, Songwong Tasneeyapant, Abhay Venkatesh, Sathya N. Ravi, and Vikas Singh. Generating accurate pseudo-labels in semi-supervised learning and avoiding overconfident predictions via Hermite polynomial activations. In CVPR, 2020."  This paper investigates the usage of hermite polynomial activations where the coefficients are learnable parameters.  They demonstrate that when the polynomial is of moderate degree there can be boosts in convergence speed, with an "early riser" property where the accuracy increases quickly at the beginning of training.  They utilize this property for the pseudo labeling problem using the speed as a supervisor (SaaS) technique.  They also demonstrate that the hermite activations can be more robust to label noise.  While this is an interesting work we do not see how it overlaps enough with our manuscript to warrant discussion.  Our work seeks to characterize the spectrum of the NTK using a power series expansion.  The work referenced above has no discussion of the NTK nor its spectrum.
> * In regard to "The paper entirely assumes that inputs are normalized, which can be fairly unrealistic because practical datasets are not. Under this assumption, I think the contributions are quite incremental, which can be derived from the simple composition of power series (coefficients are obtained from the Hermite expansion). For homogeneous activation (e.g., ReLU), the results can be easily extended to the entire R^d (also studied in [3]). But, generally, it seems that the results can be more complicated and different.”:
>      * We first point out that **the finite width result given by Theorem 4.5 does not require the inputs to be normalized.**  However, it is correct that the infinite width results do assume the data is normalized.  This assumption allows us to handle non-homogenous activations, however if one restricts to homogenous activations it is possible to relax this assumption.  **We would also like to emphasize that this assumption is quite standard (see e.g. [A], [C], [D]).** We do not expect the results for normalized vs bounded data to be in any way fundamentally different.

---

> ### Author Response · Authors · 2022-11-14
> **Initial response to reviewer VHSv (part 4)**
>
> * In regard to whether or not Theorem 3.1 in our paper is incremental, although we agree with the reviewer that the application of the Hermite expansion for understanding neural network kernels is not novel and has a rich recent history, we argue that Theorem 3.1 is nontrivial. In particular, we are unaware of any work other than the concurrent work [3] which you highlighted to us in which a power series for the deep case with arbitrary activations is derived, and, as we have already discussed, there are significant differences. Repeating our previous comment, **unlike our analysis of the NTK power series, in [3] they do not consider nonzero biases, they do not explicitly derive expressions for the coefficients of deep networks and their results on deep networks apply only for homogenous activations. Moreover, we obtain the decay rate of the coefficients of the NTK power series for shallow networks with different activations in Lemma 3.2, which is important to the derivation of asymptotic characterization of the spectrum. In particular Theorem 4.6 and Corollary 4.7, the proofs of which are given in Appendix C.3, are highly technical and required significant work.**
> * In regard to your comment "Moreover, the results of this paper are limited only to the ``fully-connected’’ neural network. Practical usage of the NTK is also related to the convolutional neural networks or more results on modern architectures would improve the results."  **We have added plots of the spectrum for convolutional nets, displayed in the new Appendix C.3, which demonstrate our results on the effective rank also hold for convolutional networks in practice.** We agree that extending our analysis in order to understand the spectrum of the CNTK is a natural and logical next step for future work.
> * In regard to your comment "some statements of this paper need to be written more formally. For example, the authors argue that the effective rank of the shallow ReLU is roughly 2.5. But, it is unclear how the network depth is small enough. Similar issue is in Observation 4.4" The goal of the two statements you have highlighted was to provide some interpretation and intuition as to the formal results Theorem 4.1 and Theorem 4.3. Indeed, they are supposed to compliment these formal results, rather than standalone, we apologize if this wasn’t clear. To explain these observations, in Table 2 we have that the zeroth hermite coefficient takes up about 44% of the sum of the coefficients, and thus we have that the effective rank is upper bounded by approximately 1/.44 which is a bit less than 2.3.  As for Observation 4.4 we are referring to Theorem 4.3 which demonstrates that if the effective rank of the data is $O(1)$ then the effective rank of the NTK is also $O(1)$.
>
> To summarize, we greatly appreciate your feedback and have made appropriate changes to the manuscript.  Again we would also like to emphasize that the results on the effective rank are just one part of the work, and the important contribution of the asymptotic results given in Theorem 4.6, Corollary 4.7, and Theorem 4.8 were not addressed in your review.  We hope that these asymptotic results will be incorporated into your final evaluation and that they may persuade you to consider raising your recommendation.  Regardless, thank you for your feedback and time.
>
> [A] Sanjeev Arora, Simon Du, Wei Hu, Zhiyuan Li, and Ruosong Wang. Fine-grained analysis of optimization and generalization for overparameterized two-layer neural networks. In Proceedings of the 36th International Conference on Machine Learning, volume 97 of Proceedings of Machine Learning Research, pp. 322–3
>
> [B] Peter L. Bartlett and Shahar Mendelson. Rademacher and Gaussian complexities: Risk bounds and structural results. J. Mach. Learn. Res., 3(null):463–482, mar 2003. ISSN 1532-4435.
>
> [C] Simon S. Du, Xiyu Zhai, Barnabas Poczos, and Aarti Singh. Gradient descent provably optimizes over-parameterized neural networks. In International Conference on Learning Representations, 2019b. URL https://openreview.net/forum?id=S1eK3i09YQ.
>
> [D] Samet Oymak and Mahdi Soltanolkotabi. Toward moderate overparameterization: Global convergence guarantees for training shallow neural networks. IEEE Journal on Selected Areas in Information Theory, 1(1), 2020. URL https://par.nsf.gov/biblio/10200049.
>
> [E] Arthur Jacot, Franck Gabriel, and Clement Hongler. Neural tangent kernel: Convergence and generalization in neural networks. In Advances in Neural Information Processing Systems, volume 31. Curran Associates, Inc., 2018. URL https://proceedings.neurips.cc/ paper/2018/file/5a4be1fa34e62bb8a6ec6b91d2462f5a-Paper.pdf.

---

> ### Author Response · Authors · 2022-11-17
> **Any further discussion before the end of rebuttal?**
>
> Dear reviewer,
>
> We made thorough efforts to address all comments from your initial review. We hope our responses and updated manuscript facilitate your assessment and motivate you to update your recommendation. Kindly do let us know if there is anything else you would like to see improved or clarified before the end of the rebuttal period. We remain attentive to your feedback!

---

### Official Review · Reviewer_rRGi · 2022-10-24

**Confidence:** 3
**Correctness:** 4
**Technical Novelty And Significance:** 4
**Empirical Novelty And Significance:** Not applicable
**Recommendation:** 8

**Clarity, Quality, Novelty And Reproducibility:**

As I have already written, I generally like the theoretical results of the paper, but I didn't check their proofs, which seem rather involved. Unfortunately the main body of the paper does not include any sketches of these proofs.

The results are mostly theoretical, but the authors provide the code for their experiments.

**Strength And Weaknesses:**

Strengths
-------------
I think that the paper develops a new and interesting general analytic approach to the study of NTK. The idea of a power expansion of the NTK appearing in main theorem 3.1 seems new and important to me. Importantly, this approach is applicable to a very large class of activations and generic data sets. As the authors show, the coefficients of this power expansion can be explicitly computed to some extent, and can be related to the spectral properties of the NTK, in particular the eigenvalue convergence rate. The obtained spectral results agree with some earlier results and provide a different perspective on them. All this opens up potential new ways for NTK analysis.

Weaknesses
-----------------
1. The experimental part of the paper seems very weak to me. There are only a few experiments with real data in section 4.1, and I don't quite see what these experiments are supposed to demonstrate. The authors observe that there is a long tail of small eigenvalues, but such long tails are common, and it is not clear how this observation confirms the developed theoretical picture. On the other hand, there are some obvious interesting features in Figure 2 that the authors don't comment on, despite there apparent relevance to the theoretical results. In particular, the curves for the activation Tanh almost match the curves for Data - is it because Tanh has a very fast decaying power expansion? Would this also be so if Tanh was replaced by the trivial linear activation? Why is there a constant difference between the Tanh and Data curves in the case of Depth-5 MNIST? Is the slope of the ReLU curve smaller than the slope of Data/Tanh because the coefficient decay of ReLU dominates the data eigenvalue decay? I think that the experimental part and its discussion could have been much more prominent.

2. I didn't understand some statements. In particular, it's not clear to me under which conditions Observation 4.2 holds and why. The notation \Omega(1) and O(1) here is confusing (what is the respective parameter?); I would suggest to write this in a more explicit form. I also don't understand the statement of Theorem 4.8. Is r(n) here an arbitrary function?  Is K_n some n x n submatrix of K and \lambda_n(K_n) its lowest eigenvalue?

**Summary Of The Paper:**

The paper studies the infinite-width NTK in the setting of general activation functions and datasets. It is shown that, under the standard Gaussian initialization, the NTK can be expressed as a power series in the data Gram matrix elements. The paper then discusses the coefficients of this series and its convergence for a few common activations. After that the authors discuss spectral properties of kernels that can be represented by such series. In particular, the paper give bounds on the efficient rank of the NTK and the convergence rates of its eigenvalues in several scenarios.

**Summary Of The Review:**

An interesting work developing a new and potentially important analytic method. However, the experimental part is weak, and there are some issues with the exposition.

---

> ### Author Response · Authors · 2022-11-14
> **Initial response to reviewer rRGi (part 1)**
>
> First, thank you for giving our paper an honest read and providing a helpful review.  We appreciate your comment on the applicability of our results to a range of data distributions and activation functions, and the insight it can provide.  You raise some important points in the "Weaknesses" section which we address below.
>
> * In regard to "The experimental part of the paper seems very weak to me. There are only a few experiments with real data in section 4.1, and I don't quite see what these experiments are supposed to demonstrate."  What these experiments are supposed to illustrate is that the NTK has a very skewed spectrum with a small number of outlier eigenvalues (as predicted by Theorem 4.1 and Observation 4.2), and that the NTK to a large extent inherits this structure from the input data Gram matrix (as predicted by Theorem 4.3, Observation 4.4, and Theorem 4.5).   **We have improved our experiments in Section 4 by computing and comparing the spectrums of the NTK for Caltech101 data with Gaussian data.  We refer the reviewer to the new Figure 1 and preceding paragraph where we provide detailed commentary.** In particular, we see the outlier eigenvalue as predicted by our theory and also observe that skewed, real world data leads to a skewed NTK spectrum, while flatter, Gaussian data leads to a flatter NTK spectrum.  **Furthermore, in the new Appendix C.3 we first give a detailed description of our experimental setup, and second provide Figure 3, which provides an analogue of Figure 1 but for CNN architectures instead of feedforward networks.** Despite CNNs lying outside the regime of our theory, we still observe the same trends as we did for feedforward networks. **In Appendix C.3 Figure 4, we also added the experiments for the asymptotic NTK spectrum and verified our asymptotic results (Corollary 4.7).**
>
> * In regard to "The authors observe that there is a long tail of small eigenvalues, but such long tails are common, and it is not clear how this observation confirms the developed theoretical picture."  We do agree that long tails commonly appear and in many different settings;  however, the reasons for such a small bulk of outliers in the spectrum of the NTK is not clear a priori.  In Section 4.1 we derive a simple theoretical explanation for this phenomenon which was lacking in the literature to date.  First, the network will always have a dominant outlier eigenvalue due to the nonzero constant coefficient in the power series (note that the network has a nonzero constant coefficient whenever the network has nonzero bias terms or if the activation has nonzero Gaussian mean, which is typical).  This is formalized by Theorem 4.1 and Observation 4.2.  Second, real-world data often has a skewed spectrum (and hence a low effective rank), and whenever this is the case the NTK will also exhibit a skewed spectrum (and consequently a low effective rank). This is formalized by Theorem 4.3, Observation 4.4, and Theorem 4.5. **We have added detailed comments to the paper (above the new Figure 1) to better convey these points and to highlight the connection between what we observe empirically and our theory.**
>
> * In regard to "On the other hand, there are some obvious interesting features in Figure 2 that the authors don't comment on, despite there apparent relevance to the theoretical results. In particular, the curves for the activation Tanh almost match the curves for Data - is it because Tanh has a very fast decaying power expansion? Would this also be so if Tanh was replaced by the trivial linear activation?" You are correct that the decay of the coefficients for Tanh is fast, relative to ReLU, and therefore the resulting spectrum more closely follows the spectrum of the input data. Furthermore, if Tanh was replaced by the identity activation the curve would even more closely match the input data spectrum, as in this case the power series is completely concentrated on the first Hermite coefficient which is simply the identity. In addition, for the plots in Figure 2 to remain as close to practice as possible we use the standard parameterization in PyTorch which is a bit different from the NTK parameterization.  In the standard parameterization the conjugate kernel given by the last layer features has a more significant contribution to the NTK.  Since tanh looks like the identity near the origin (and hence will be dominant in the first hermite coefficient), the last layer features for a depth 2 tanh network will look very similar to the input features.
>
> * In regard to "Why is there a constant difference between the Tanh and Data curves in the case of Depth-5 MNIST?" As for the constant difference between the data and tanh curves in the case of a depth-5 MNIST network we believe this is due to the deeper network having a larger constant term in the power series and thus a larger outlier eigenvalue, which is predicted by our theory. **We have added a comment to this effect in above the new Figure 1.**

---

> ### Author Response · Authors · 2022-11-14
> **Initial response to reviewer rRGi (part 2)**
>
> * In regard to "Is the slope of the ReLU curve smaller than the slope of Data/Tanh because the coefficient decay of ReLU dominates the data eigenvalue decay? I think that the experimental part and its discussion could have been much more prominent."
> Yes, asymptotically, ReLU's hermite coefficients decay slower than tanh (as demonstrated in Lemma 3.2) and thus the NTK spectrum of ReLU has slower asymptotic decay (as demonstrated in Corollary 4.7 and Theorem 4.8).  This explains why the slope of ReLU is flatter than that of tanh at the end.  **We agree with the reviewer that these observations connect well with the theory and have updated the manuscript accordingly, in particular, see the paragraph above Figure 1 and the new Appendix C.3.**
>
> * In regard to "I didn't understand some statements. In particular, it's not clear to me under which conditions Observation 4.2 holds and why."  Observation 4.2 holds whenever Theorem 4.1 holds.  Theorem 4.1 states that the Trace of the kernel divided by the largest eigenvalue is upper bounded by a constant.  Equivalently, this means that the largest eigenvalue divided by the trace is lower bounded by a constant.  Furthermore, by the Markov-like inequality (10), this implies for any $\epsilon > 0$ that there are $O(\mathrm{eff}(K) / \epsilon)$ eigenvalues which are greater than or equal to $\epsilon \lambda_1(K)$.  Thus if one sets $\epsilon = 0.1$ we have that there are $O(10 \cdot \mathrm{eff}(K)) = O(1)$ eigenvalues on the same order of magnitude as $\lambda_1(K)$.
> * "The notation $\Omega(1)$ and $O(1)$ here is confusing (what is the respective parameter?); I would suggest to write this in a more explicit form."  The respective parameter is $n$.  A well conditioned matrix will have an effective rank that is $\Omega(n)$ and consequently the largest eigenvalue takes up an $O(1/n)$ proportion of the trace.  **Following your suggestion, we have updated the manuscript to make this clear.**
> * In regard to "I also don't understand the statement of Theorem 4.8. Is $r(n)$ here an arbitrary function? Is $\mathbf{K}_n$ some $n \times n$ submatrix of $\mathbf{K}$ and $\lambda_n(\mathbf{K}_n)$ its lowest eigenvalue?"  We apologize for the confusion, these were a result of editing errors in the original submission.  $r(n)$ is the rank of the matrix $\mathbf{X} \mathbf{X}^T$.  The subscript on $\mathbf{K}$ is a typo, we have replaced $\mathbf{K}_n$ with $\mathbf{K}$.  $\lambda_n(\mathbf{K}_n)$ should simply be $\lambda_n(\mathbf{K})$ which is indeed the smallest eigenvalue.  **We have updated the manuscript to make the notation clear.**
>
> In summary, we thank you for your helpful review and comments.  **We have made a systematic effort to address all your comments and concerns, and have updated the manuscript accordingly.**  We hope that our additional experimental data and improved exposition will motivate you to consider an increase in your recommendation.  Again we also would like to highlight that one of our most important contributions is the asymptotic characterization of the spectrum, given by Theorem 4.6, Corollary 4.7, and Theorem 4.8, which was not much discussed in your review.  In any case, we greatly appreciate your valuable time and for providing helpful feedback.

---

> > ### Comment · Reviewer_rRGi · 2022-11-19
> > **Thank you**
> >
> > Thank you for your response.
> > 1. The revision has indeed improved the paper in multiple ways.
> > 2. You seem to have removed experiments with MNIST and CIFAR10 - why?
> > 3. In Figure 1, the caption says: "The thick part of each curve corresponds to the mean across 10 trials, while the transparent part corresponds to the 95% confidence interval", but I don't see any transparent parts there. Moreover, some of the curves are not visible (presumably because they are behind other curves) and should be shown somehow (e.g. by extra thickness).
> > 4. (minor) My understanding of Observation 4.2 now it that $O(1), \Omega(1)$ mean some bounds independent of the data set $\mathbf H$ - if so, it might be clearer to just say it this way.

---

> > > ### Author Response · Authors · 2022-11-19
> > > **Thank you for your response**
> > >
> > > Dear reviewer,
> > >
> > > Thank you for your careful evaluation of our revision and your positive feedback. We address your latest questions below.
> > >
> > > * In regard to "You seem to have removed experiments with MNIST and CIFAR10 - why?”. We replaced the MNIST and CIFAR10 experiments with those on Caltech101 data and Gaussian data in order to better illustrate the influence of the data spectrum on the spectrum of the NTK. In particular, the Caltech101 data is skewed, which results in a skewed NTK spectrum, while the spectrum of the Gaussian data is fairly flat, resulting in a flatter NTK spectrum. By contrast, the spectra of the MNIST and CIFAR10 data are very similar and therefore did not illustrate this point as well. Indeed, each of the real datasets we have experimented with, i.e., MNIST, CIFAR10 and Caltech101, exhibited a similar skewed spectrum. We would happily re-add our plots for MNIST and CIFAR10 to the Appendix for the camera ready version of the manuscript.
> > >
> > > * In regard to "I don't see any transparent parts there. Moreover, some of the curves are not visible (presumably because they are behind other curves) and should be shown somehow (e.g. by extra thickness).” This is actually due to the low variance between trials resulting in a very tight 95% confidence interval. This means the spread is very tightly concentrated around the mean and thus the transparent region is barely visible. We note that if one zooms in on these images a blur around each line becomes more noticeable but we agree with the reviewer that this might be confusing to the reader. We will add comments to both Figure 1 and 3 emphasizing that the variance between trials is very small and therefore the confidence interval visually coincides with the mean. We will also try to draw a clearer figure in the camera ready version. In regard to overlapping curves, we will try to plot with greater thickness in the camera ready version as you suggest, and also add a comment when in particular the linear and data curves overlap.
> > >
> > > * In regard to “My understanding of Observation 4.2 now is that $O(1), \Omega(1)$ mean some bounds independent of the data set $\mathbf{H}$ - if so, it might be clearer to just say it this way.” Your understanding is correct. What we mean is that these quantities are upper bounded/ lower bounded by a constant and these bounds do not increase or decrease as the number of training data goes to infinity. We will take action on your suggestion and make this clearer in the camera ready version.
> > >
> > > Let us know if you have any questions about the revised proof sketch and our latest response. Thanks again for taking the time to help improve this manuscript and we remain attentive to your comments and feedback!

---

> > > > ### Comment · Reviewer_rRGi · 2022-11-22
> > > > **Thank you**
> > > >
> > > > Thank you for your clarification. I believe that the revision has improved the paper, so I'm increasing my score.

---

> ### Author Response · Authors · 2022-11-17
> **Any further discussion before the end of rebuttal?**
>
> Dear reviewer,
>
> We made thorough efforts to address all comments from your initial review. We hope our responses and updated manuscript facilitate your assessment and motivate you to update your recommendation. Kindly do let us know if there is anything else you would like to see improved or clarified before the end of the rebuttal period. We remain attentive to your feedback!

---

### Official Review · Reviewer_jD66 · 2022-10-25

**Confidence:** 3
**Correctness:** 4
**Technical Novelty And Significance:** 3
**Empirical Novelty And Significance:** Not applicable
**Recommendation:** 8

**Clarity, Quality, Novelty And Reproducibility:**

The text is mostly well written, and the results seem novel compared to previous literature, though I haven't checked the proofs in the appendix. Code is provided to reproduce the empirical results

**Strength And Weaknesses:**

### Strengths
* The analysis is mathematically rigorous.
* A discussion on their practical interpretation follows each of the main results.
* The approach taken in the analysis is relatively simpler than pre-existing attempts on deriving general results for the NTK spectrum.
* The power series formulation (Thr. 3.1) and in special the eigenvalue results (Lem. 3.2, Cor. 4.7, Thr. 4.8) have the potential to impact many applications of NTK-based convergence analysis of deep learning frameworks.
* Empirical results practically validating theoretical insights are also presented (Fig. 1 & 2)

### Weaknesses
* In Assumption 2, "The hyperparameters of the network satisfy $\gamma_w^2 + \gamma_b^2 = 1$ ... and $\sigma_b^2 = 1 - \sigma_w^2 \mathbb{E}_{Z\sim\mathcal{N}(0,1)}[\phi(Z)]$" seems somewhat restrictive to me, and this part of the assumption hasn't been discussed in the paragraph following it.
* Some notation definitions are missing in the main text, e.g.: $r(n)$, $h_k$

**Summary Of The Paper:**

This paper presents theoretical results on the eigenvalue spectrum of the neural tangent kernel (NTK) based on Hermite polynomial expansions. The results enable expressing the NTK matrices for neural networks of arbitrary depth via power series expansions based on the data points matrix. Bounds on the eigenvalues of the NTK for popular activation functions are also provided.

**Summary Of The Review:**

The paper presents an interesting theoretical analysis deriving bounds for the NTK spectrum decay and relating it to the data eigenspectrum. The results practical interpretation is provided and empirical results to demonstrate theoretical insights in practice are also presented. The approach is novel compared to previous work, and the results are more broadly applicable.

---

> ### Author Response · Authors · 2022-11-14
> **Initial response to reviewer jD66**
>
> We thank the reviewer for their thoughtful and positive comments. In the text below we believe we have fully addressed the two areas highlighted for improvement.
>
> * In regard to the comment “In Assumption 2, The hyperparameters of the network satisfy $\gamma_w^2 + \gamma_b^2 = 1$  and  $\sigma_b^2 = 1 - \sigma_w^2 \mathbb{E}_{Z \sim N(0, 1)}[\phi(Z)^2]$ seems somewhat restrictive to me, and this part of the assumption hasn't been discussed in the paragraph following it.”, we would like to highlight Appendix A.3, in which we argue that Assumption 2 aligns with the principles of many initialization techniques used in practice, in particular the LeCun, Xavier, and He initializations. To avoid the problems of exploding and vanishing gradients, the goal of these particular initializations is for the mean and variance of the activations to be 0 and constant respectively, across all layers of the network. While Assumption 1 ensures the mean of the preactivations is 0, Assumption 2 ensures that in the infinite width limit the variance of the network preactivations are fixed at one across all units and layers. We remark that variance one was chosen for convenience and our results could be extended to any fixed constant. From a theoretical perspective, Assumption 2 also ensures the Gaussian kernel associated with the NTK is a univariate function of the correlation between the two input arguments, which is highly convenient to work with. Due to space constraints it is hard to discuss these aspects in detail in Section 3, however, **we have amended the paragraph following Assumption 2 to better motivate it as well as highlight these connections and motivations.**
>
> * In regard to “Some notation definitions are missing in the main text, e.g.: $r(n)$, $h_k$” we apologize for the lack of clarity concerning both of these. The quantity $r(n)$ refers to the rank of $XX^T$ which is an $n \times n$ matrix. **We have amended Theorem 4.8 to make this clearer.** With regards to $h_k$, this is the notation we use for the $k$th normalized probabilists Hermite polynomial, **we have corrected Section 2.1 to make this clear. In addition, we have made changes to improve the clarity of Appendix C.4.**
>
> To summarize, we have amended the paper in line with this feedback and would like to thank the reviewer for helping us improve the exposition of the manuscript.  Please let us know if you have any further questions or comments.  Again, thank you for taking the time to give our paper a careful read, and a thoughtful and helpful review.

---

> ### Author Response · Authors · 2022-11-17
> **Any further discussion before the end of rebuttal?**
>
> Dear reviewer,
>
> We made thorough efforts to address all comments from your initial review. We hope our responses and updated manuscript facilitate your assessment. Kindly do let us know if there is anything else you would like to see improved or clarified before the end of the rebuttal period. We remain attentive to your feedback!

---

### Decision · Program_Chairs · 2023-01-20

**Decision:**

Accept: poster

**Justification For Why Not Higher Score:**

See meta review that clearly explains the paper is incremental in comparison to prior work.

**Justification For Why Not Lower Score:**

Some results are new and worth publishing.

**Metareview: Summary, Strengths And Weaknesses:**

This paper studies a power series of the NTK for infinitely wide neural networks assuming the input is distributed on the unit sphere. Similar power series have been considered in various prior works such as Oymak et al. (2019) or Oymak & Soltanolkotabi (2020). The paper heavily builds on this prior work. The authors here focus more on the effective rank of the NTK matrix which they compare to that of the data Gram matrix. For instance, the bounds derived in the paper reveal some insights about several popular activation functions (see e.g. Lemma 3.2), although I note here again that some of these results are already known by Panigrahi et al. (2020).

In general, the reviewers are mostly positive about the paper, although they find the paper incremental in comparison to prior work. The post-rebuttal discussion led some of the reviewers to increase their scores. I will recommend acceptance as a poster as the paper has some interesting theoretical results on the effective rank, but I would also recommend the authors to better highlight the novel insights in comparison to the many prior works that have used similar power series expansion (for instance by expanding the conclusion/discussion section of the paper that is currently very short).

**Note From Pc:**

if the above contains the word "oral" or "spotlight" please see: "oral" presentation means -> notable-top-5% and "spotlight" means -> notable-top-25%. As stated in our emails, we are disassociating presentation type from AC recommendations

**Summary Of Ac-Reviewer Meeting:**

N/A